# SurrogateSHAP: Training-Free Contributor Attribution for Text-to-Image (T2I) Models

**Mingyu Lu** [1]  **Soham Gadgil** [1]  **Chris Lin** [1]  **Chanwoo Kim** [1]  **Su-In Lee** [1]

## Abstract

As Text-to-Image (T2I) diffusion models are increasingly used in real-world creative workflows, a principled framework for valuing contributors who provide a collection of data is essential for fair compensation and sustainable data marketplaces. While the Shapley value offers a theoretically grounded approach to attribution, it faces a dual computational bottleneck: (i) the prohibitive cost of exhaustive model training for each sampled subset of players (i.e., data contributors) and (ii) the combinatorial number of subsets needed to estimate marginal contributions due to contributor interactions. To this end, we propose SURROGATESHAP, a training-free framework that approximates the expensive training game through inference from a pretrained model. To further improve efficiency, we employ a gradient-boosted tree to approximate the utility function and derive Shapley values analytically from the tree-based model. We evaluate SurrogateSHAP across three diverse attribution tasks: (i) image quality for DDPM-CFG on CIFAR-20, (ii) aesthetics for Stable Diffusion on Post-Impressionist artworks, and (iii) product diversity for FLUX.1 on Fashion-Product data. Across settings, SurrogateSHAP outperforms prior methods while substantially reducing computational overhead, consistently identifying influential contributors across multiple utility metrics. Finally, we demonstrate that SurrogateSHAP effectively localizes data sources responsible for spurious correlations in clinical images, providing a scalable path toward auditing safety-critical generative models. Code is available at https://github.com/suinleelab/SurrogateSHAP.

[1] Paul G. Allen School of Computer Science & Engineering, University of Washington. Correspondence to: Mingyu Lu <{mingyulu, sgadgil, clin25, chanwookim, suinlee}@cs.washington.edu>.

*Proceedings of the 43rd International Conference on Machine Learning*, Seoul, South Korea. PMLR 306, 2026. Copyright 2026 by the author(s).

## 1. Introduction

Modern conditional generative models enable controlled generation from structured signals such as class labels, text prompts, or learned embeddings. In particular, Text-to-image (T2I) diffusion models such as Stable Diffusion (Rombach et al., 2022) and FLUX (Labs et al., 2025), have driven rapid progress in controllable image synthesis. At the same time, their widespread deployment has intensified concerns about fair credits for training data contributors. In the context of data marketplaces (Zhang et al., 2024) and revenue sharing for generative AI (Deng et al., 2023; Wang et al., 2024a), a contributor typically provides a group of samples rather than a single sample. This motivates *contributor* attribution: quantifying the marginal value of each contributor's group of samples as a unit.

A common approach for group-level attribution is to aggregate individual data attributions, which measure the impact of training data on the model. Existing methods for diffusion models, such as D-TRAK (Zheng et al., 2023) and DAS (Lin et al., 2024), approximate the leave-one-out (LOO) influence (Koh & Liang, 2017) using first-order gradients. To obtain group-level scores, one typically sums these individual influences (Koh et al., 2019; Deng et al., 2023) or computes group-wise influence directly (Ley et al., 2024). While efficient, this paradigm typically relies on loss-based gradients, which may fail to capture downstream properties such as aesthetics. Furthermore, prior work has shown that such additive approximation often underestimates a group's true influence (Koh et al., 2019; Basu et al., 2020).

To address these concerns, researchers have turned to the Shapley value (Shapley et al., 1953), a principled valuation framework that treats each data provider as a "player" (Wang et al., 2024a; Lu et al., 2025; Lee et al., 2025). It quantifies importance by averaging a contributor's marginal contribution, the change in utility when their data is added, across all possible subsets of players. Despite its theoretical appeal, applying Shapley value to modern T2I diffusion faces a dual scalability bottleneck. First, its computation requires evaluating a utility function across an exponential number of subsets, which ideally entails model training for every subset. While recent work proposes alternatives to bypass training (Lu et al., 2025), these alternatives still incur

high computational costs. Other approximations, such as the gradient-based methods (Wang et al., 2024b), are designed for individual data points and do not naturally scale to group-level attribution. Second, even with sample-efficient estimators (Lundberg & Lee, 2017; Mitchell et al., 2022), obtaining stable, low-variance estimates can require a prohibitive number of subset queries when the utility function is highly non-linear or involves complex interactions—a common effect in dataset valuation, even for regression tasks (Garrido Lucero et al., 2024).

To enable efficient and accurate Shapley computation at the scale of modern text-to-image diffusion models, we introduce SURROGATESHAP, which has two complementary ideas. First, we define a training-free proxy game that reinterprets the target model as a mixture of conditional generators, enabling subset evaluation without training. We propose a theoretical bound and empirically demonstrate that this proxy closely tracks the subset utility of trained models. Second, to improve Shapley estimation for complex utility functions, we propose learning a tree-ensemble surrogate of the proxy game and computing Shapley values analytically via TreeSHAP. This surrogate accurately captures the underlying utility function, allowing for high-fidelity subset approximations that significantly enhance the sample efficiency of Shapley estimation. We benchmark SurrogateSHAP against existing attribution methods and find that it significantly outperforms prior baselines in predicting held-out model behaviors. Furthermore, our approach demonstrates superior computational and sampling efficiency compared to existing methods based on the Shapley value. Finally, we show that SurrogateSHAP reliably identifies key contributors across diverse use cases, ranging from perceptual quality to auditing biased data sources in a clinical setting, thereby offering a principled framework for contributor valuation.

## 2. Related Work

**Data Attribution in Diffusion Models** Existing attribution methods for diffusion models primarily focus on individual data attribution and utilize the first-order gradient to approximate leave-one-out (LOO) influence. For example, Journey-TRAK (Georgiev et al., 2023) and D-TRAK (Zheng et al., 2023) adapt the TRAK framework (Park et al., 2023) to diffusion training by computing TRAK-style scores from gradients of diffusion losses, whereas DAS (Lin et al., 2024) utilizes the KL divergence between noise distributions as a proxy for sample impact.[1] However, these methods are anchored to the diffusion training objective (e.g., noise-prediction losses); their attributions can be misaligned with downstream utilities that are not directly optimized during training, such as aesthetics or diversity.

**Scaling Individual Data to Group Influence** Group-level attribution seeks to evaluate entity-based subsets, such as data owners or curated collections. While a natural extension of individual data attribution is summation (Koh et al., 2019) or computing group influence via batch gradients (Ley et al., 2024), this linear approximation often diverges from the actual change in loss as the group size increases (Koh et al., 2019; Basu et al., 2020), suggesting that simple aggregation overlooks the non-linear interaction effects present in large-scale data removals.

**Shapley-Based Contributor Attribution** Recent work has leveraged the *Shapley value* (Shapley et al., 1953) to provide principled frameworks for revenue sharing and copyright assignment for data contributors in T2I models (Wang et al., 2024a; Lee et al., 2025; Lu et al., 2025). Wang et al. (2024a); Lee et al. (2025) rely on exact retraining, which is only feasible when the number of contributors is small, an assumption that rarely holds for T2I models. While Lu et al. (2025) proposed replacing full training with sparsified fine-tuning, the fine-tuning still incurs a significant computational cost considering the total number of subsets. Furthermore, existing works often rely on Shapley estimators (Lundberg & Lee, 2017; Mitchell et al., 2022) that do not account for the high-variance challenges inherent in the non-linear and non-additive utility functions often found in dataset valuation (Garrido Lucero et al., 2024). These challenges are especially pronounced in T2I diffusion models, motivating methods that can both (i) reduce the cost of per-subset evaluation and (ii) improve the sampling efficiency for Shapley estimation.

## 3. Preliminaries and Problem Setup

This section establishes the formal framework for Text-to-Image (T2I) diffusion models and defines the problem of attributing global model properties to data contributors.

### 3.1. Text-to-Image (T2I) Diffusion Models

We consider T2I models that approximate a conditional data distribution $q(\mathbf{x}_0|\mathbf{c})$ for a given conditioning signal $\mathbf{c}$ (e.g., a text embedding). This framework encompasses Denoising Diffusion Probabilistic Models (DDPMs) (Ho et al., 2020), Latent Diffusion Models (LDMs) (Rombach et al., 2022), and Flow Matching (FM) models (Lipman et al., 2022).

**Training and Inference.** Each sample $(\mathbf{x}, y) \in \mathcal{D}$ consists of an image $\mathbf{x}$ and a prompt $y$, where $y$ is mapped to $\mathbf{c} = \psi(y)$ via a text encoder $\psi$.[2] For DDPM/LDM variants, training optimizes a conditional denoising objective:

$$\mathcal{L}(\mathbf{x}, y; \theta) = \mathbb{E}_{t,\epsilon} \left[ \lambda_t \left\| \epsilon - \epsilon_\theta(\mathbf{x}_t, t, \mathbf{c}) \right\|_2^2 \right], \quad (1)$$

---

[1]Please refer to Section A for details about these methods.

[2]Many T2I models, e.g., Stable Diffusion, keep the text encoder frozen during training.

where $\epsilon_\theta$ predicts the noise added to image $\mathbf{x}$ at timestep $t$, and $\lambda_t$ is a timestep-dependent loss weight. FM models generalize this by regressing a conditional vector field $v_\theta(\mathbf{x}_t, t, \mathbf{c})$. We denote $\theta$ as the target model optimized on the full dataset $\mathcal{D}$, and $p_\theta(\mathbf{x}_0|\mathbf{c})$ as the distribution induced by the model during inference via classifier-free guidance (CFG) (Ho & Salimans, 2022).

## 3.2. Attribute Contributor via Shapley value

We study attribution settings where data provenance is tied to explicit prompt metadata (e.g., artist name, brand tokens), so that contributors correspond to subsets of prompt labels.

**Definition 3.1** (Contributor Attribution). An attribution method $\tau$ assigns importance scores to contributors based on a utility function $v : 2^N \to \mathbb{R}$ that maps a subset of contributors to a scalar utility.

**Contributor–label correspondence.** Let $N = \{1, \ldots, n\}$ denote the set of contributors. Each contributor $i \in N$ is associated with a set of identifying tags or aliases, $\mathcal{T}_i$ (e.g., "Van Gogh" or "vanGogh"). Let $\mathcal{Y}$ denote the set of all text prompts in the dataset. We define $\mathcal{Y}_i \subseteq \mathcal{Y}$ as the subset of prompts that contain at least one tag $t \in \mathcal{T}_i$. For any coalition of contributors $S \subseteq N$, we define the collective prompt set as $\mathcal{Y}_S := \bigcup_{i \in S} \mathcal{Y}_i$. The training data induced by coalition $S$ is then

$$\mathcal{D}_S := \{(\mathbf{x}, y) \in \mathcal{D} : y \in \mathcal{Y}_S\}. \tag{2}$$

**The Training Game.** Let $\mathcal{A}$ be a training algorithm, ranging from full retraining to various fine-tuning procedures, e.g., LoRA or full-finetuning, and let $\theta_S^* \sim \mathcal{A}(\mathcal{D}_S)$ denote the parameters obtained from a stochastic training run on training set $\mathcal{D}_S$. We define the Training Game via the utility function:

$$v(S) := \mathcal{F}(p_{\theta_S^*}), \tag{3}$$

where $p_{\theta_S^*}$ is the (implicit) model distribution induced by $\theta_S^*$, and $\mathcal{F} : \tilde{\mathcal{P}}(\mathcal{X}) \to \mathbb{R}$ is a function that maps this distribution to a scalar quality metric. Following Lu et al. (2025), we define utility over a collection of outputs because comprehensive model evaluation requires studying global properties (Covert et al., 2020) rather than individual outputs. For example, in supervised learning, $\mathcal{F}$ might manifest as test accuracy; in text-to-image generation, it may encompass metrics ranging from perceptual scores (Zhang et al., 2018) to distributional measures, e.g., FID (Heusel et al., 2017).

**Shapley value computation.** The Shapley value (Shapley et al., 1953) assigns credit to each contributor (player) $i \in N$ via the expected marginal contribution over subsets. Formally, given the training utility function $v(S)$, the Shapley value of contributor $i$ is

$$\phi_i(v) = \frac{1}{n} \sum_{S \subseteq N \setminus \{i\}} \binom{n-1}{|S|}^{-1} [v(S \cup i) - v(S)]. \tag{4}$$

**Computational bottlenecks.** Estimating $\phi_i(v)$ for the training game is computationally challenging. Each coalition query $v(S) = \mathcal{F}(p_{\theta_S^*})$ requires: (i) training a model $\theta_S^*$ on $\mathcal{D}_S$, and (ii) generating a sample set to compute the distributional metric $\mathcal{F}$. For models like FLUX (Labs et al., 2025), the per-query cost is expensive. Beyond this cost, the utility functions are typically non-additive, which inflates the variance of marginal contributions and increases the number of coalitions $M$ needed for accurate estimation (Mitchell et al., 2022). Consequently, even with sampling-based estimators such as KernelSHAP (Lundberg & Lee, 2017), the total cost $O(M \cdot \text{Cost}(\mathcal{A}))$ remains prohibitive. This motivates our training-free proxy framework, which approximates the training-game utility to reduce the cost for subset queries.

## 4. Method - SurrogateSHAP

We now present SurrogateSHAP in detail. First, we define a training-free proxy game to enable efficient subset evaluation in T2I models (Section 4.1). Second, we introduce a surrogate model-based procedure (Section 4.2) that computes Shapley values with improved sample efficiency.

### 4.1. Training-Free Coalition Evaluation

Training $\theta_S^*$ even for a single coalition $S \subseteq N$ is computationally prohibitive for large-scale diffusion models. To bypass this, one might consider leveraging the compositional properties of diffusion scores, reformulating $p_{\theta_S^*}$ via test-time score composition (Liu et al., 2022). However, while this reduces the problem to a standard inference task, the computational complexity scales as $O(T|S|)$ per sample, as it requires aggregating $|S|$ scores at every denoising timestep $t \in T$. Fortunately, as our objective is to evaluate the global utility $\mathcal{F}$ on the induced distribution $p_{\theta_S^*}$, we can reformulate the coalition's output as a distributional mixture. For conditional diffusion models, $p_{\theta_S^*}$ is defined as:

$$p_{\theta_S^*}(\mathbf{x}) = \sum_{y \in \mathcal{Y}_S} \pi_S(y) p_{\theta_S^*}(\mathbf{x} \mid \psi(y)), \tag{5}$$

where $\pi_S(y) = \hat{p}(y) / \sum_{y' \in \mathcal{Y}_S} \hat{p}(y')$ represents the relative label prior based on the empirical frequency $\hat{p}$ in $\mathcal{D}$.

#### 4.1.1. TRAINING-FREE PROXY GAME

Building on this structural observation, we define a training-free proxy game $(\mathcal{N}, \hat{v}_\theta)$ that leverages the conditional generation capabilities of the target model $\theta$. By assuming the

full-data conditional $p_\theta(\mathbf{x} \mid \psi(y))$ serves as a valid proxy for the trained conditional $p_{\theta_S^*}(\mathbf{x} \mid \psi(y))$ for all $y \in \mathcal{Y}_S$, we obtain the proxy utility function:

$$\hat{v}_\theta(S) := \mathcal{F}\left( \sum_{y \in \mathcal{Y}_S} \pi_S(y) p_\theta(\mathbf{x} \mid \psi(y)) \right). \qquad (6)$$

Evaluating equation 6 is highly efficient as it eliminates the need for subset-specific training, reducing subset evaluation to a mixture sampling problem. To obtain $\hat{v}_\theta(S)$ we draw a label $y \sim \pi_S$ and then perform standard conditional generation $\mathbf{x} \sim p_\theta(\cdot \mid \psi(y))$. By bypassing both training and score-level aggregation, this proxy game enables practical coalition evaluation for large-scale generative models.

### 4.1.2. THEORETICAL FIDELITY OF THE PROXY

The fidelity of proxy utility to the training utility hinges on the stability of the class-conditional distributions, which we formalize below:

**Assumption 4.1** (($\varepsilon, \varphi$)-Stability). *For any coalition $S \subseteq N$ and any condition $\mathbf{c} \in \mathcal{C}$, let $\varphi_u : \mathcal{X} \to \mathbb{R}^{m_u}$ be a perceptual embedding function (or utility-aligned feature extractor). We assume that under a shared-noise coupling $\gamma_\mathbf{c}$ induced by a common noise seed $\mathbf{z} \sim \mathcal{N}(0, I)$, the root-mean-squared (RMS) representation drift between the class-conditional output distributions of the target model $\theta$ and the coalition-trained model $\theta_S^*$ is uniformly bounded in the utility-aligned feature space:*

$$\mathrm{d}^{\varphi_u}(p_{\theta_S^*}, p_\theta \mid \mathbf{c}) := \left( \mathbb{E}_{(X_S, X_F) \sim \gamma_\mathbf{c}} \left[ \|\varphi_u(X_S) - \varphi_u(X_F)\|_2^2 \right] \right)^{1/2} \leq \varepsilon^{\varphi_u}. \tag{7}$$

*where $X_S \sim p_{\theta_S^*}(\cdot \mid \mathbf{c})$ and $X_F \sim p_\theta(\cdot \mid \mathbf{c})$ are the outputs of the trained model and target model, respectively, generated from the same noise seed $\mathbf{z}$. This coupling implies*

$$W_p\left( \varphi_{u\#} p_{\theta_S^*}(\cdot \mid \mathbf{c}), \ \varphi_{u\#} p_\theta(\cdot \mid \mathbf{c}) \right) \leq d^{\varphi_u}\left( p_{\theta_S^*}, p_\theta \mid \mathbf{c} \right) \leq \varepsilon^{\varphi_u}. \tag{8}$$

Since $(\varphi_u(X_S), \varphi_u(X_F))$ constitutes a valid coupling of the distributions induced in the representation space (Peyré et al., 2019), and $W_p$ is defined as the infimum over all possible couplings, for $p = 2$, this implies $W_2 \leq d^{\varphi_u} \leq \varepsilon^{\varphi_u}$.

**Proposition 4.2** (Proxy Error Bound). *Under Assumption 4.1, recall that $v(S) := \mathcal{F}(p_{\theta_S^*})$, where $p_{\theta_S^*}$ is the coalition-specific mixture in Equation (5), and $\hat{v}_\theta(S)$ is defined by the coalition-specific proxy mixture in Equation (6). The utility functional $\mathcal{F}$ is induced by $\mathcal{V}$ via*

$$\mathcal{F}(p) := \mathcal{V}\left( \mathbb{E}_{\mathbf{c} \sim \pi}[\varphi_{u\#} p(\cdot \mid \mathbf{c})] \right), \qquad (9)$$

*where $\pi$ denotes the condition prior used in the corresponding coalition mixture, i.e., $\pi = \pi_S$ for coalition $S$. Here, $\mathcal{V}$ is a utility functional on probability measures over $\mathbb{R}^{m_u}$. If $\mathcal{V}$ is $L_u$-Lipschitz with respect to the $W_p$ metric for some*

$p \in \{1, 2\}$ *on $\mathcal{P}(\mathbb{R}^{m_u})$, then for any realization of the trained parameters $\theta_S^*$, the following bound holds:*

$$|v(S) - \hat{v}_\theta(S)| \leq L_u \varepsilon^{\varphi_u}. \qquad (10)$$

*The proof is provided in Section B.1.*

*Remark* 4.3 (Interpretation and Metric Applicability). Proposition 4.2 shows that the proxy error is controlled by two factors: the stability of the model trained on a coalition in a utility-aligned feature space and the Lipschitz continuity of the evaluation functional. The stability assumption means that training on a coalition should only mildly perturb the model's class-conditional output distributions in the representation space relevant to the evaluation metric. Under this condition, $\hat{v}_\theta(S)$ serves as a computationally efficient surrogate for $v(S)$, with approximation error bounded by equation 10.

The Lipschitz requirement covers many standard image evaluation utilities. For expectation-based metrics such as LPIPS-based perceptual scores (Zhang et al., 2018) or CLIP-score (Radford et al., 2021), if the underlying scorer $s(\cdot)$ is $L_s$-Lipschitz and $\mathcal{V}$ is linear in the induced distribution, then the expected score is $L_s$-Lipschitz with respect to $W_1$. For distributional metrics such as FID (Heusel et al., 2017), which compares Gaussian moment matches in feature space, the corresponding functional is locally $W_2$-Lipschitz. Although FID is not globally Lipschitz, a small $W_2$ drift under bounded second moments implies small perturbations of the feature mean and covariance, yielding stability in our evaluation regime.

### 4.2. Approximate Shapley Values via Tree Explainer

Although the training-free proxy $\hat{v}_\theta(S)$ makes individual coalition queries inexpensive, Shapley value estimation can still require many queries due to the utility function's strong nonlinearity in the player set (Garrido Lucero et al., 2024). Butler et al. (2025) addresses this by fitting a gradient-boosting tree (GBT) surrogate to exploit hierarchical interaction priors and subsequently extracting a sparse interaction spectrum. However, such explicit sparsification can compromise faithfulness if the utility function contains even higher-order interactions. In contrast to explicit sparsification, recent work has computed Shapley values directly on tree-based surrogates and corrected residuals with Monte Carlo sampling (Witter et al., 2026). We similarly use the learned GBT surrogate and compute Shapley values directly on the trained ensemble using TreeSHAP (Lundberg et al., 2020), without imposing explicit sparsity or truncation constraints on the utility function. Our implementation is as follows:

**Approximating Utilities with Tree Surrogates** We sample $M$ coalitions $S_m$ via the Shapley kernel, and encode each as $\mathbf{z}_m \in \{0, 1\}^n$, the indicator vector of contribu-

tors $S_m \subseteq \{1, \ldots, n\}$. These samples form a training set $\mathcal{D} = \{(\mathbf{z}_m, \hat{v}_\theta(S_m))\}_{m=1}^M$ used to learn a surrogate model $\hat{f}_\omega$ by solving:

$$\hat{f}_\omega = \arg\min_{f \in \mathcal{H}} \sum_{m=1}^M (f(\mathbf{z}_m) - \hat{v}_\theta(S_m))^2 + \Omega(f), \quad (11)$$

where $\Omega$ is a tree-specific complexity penalty (Section C.3). We optimize hyperparameters using 5-fold cross-validation. This transforms expensive subset queries into a reusable surrogate that can be evaluated on any point within the Boolean hypercube $\{0, 1\}^n$.

**TreeSHAP on the Surrogate Model** We then compute Shapley values $\hat{\phi} = \phi(\hat{f}_\omega)$ via Interventional TreeSHAP (Lundberg et al., 2020) with the all-zeros reference $\mathbf{r} = \mathbf{0}$ as the background. Because TreeSHAP is exact for tree models, $\hat{\phi}$ satisfies the efficiency axiom $\sum_{i=1}^n \hat{\phi}_i = \hat{f}_\omega(\mathbf{1}) - \hat{f}_\omega(\mathbf{0})$. Under this framework, $\hat{\phi}$ are deterministic and exact conditional on $\hat{f}_\omega$; the remaining error is driven by the surrogate's generalization error over the $2^n$ possible coalitions.

**Error decomposition** Our goal is to estimate $\phi(v)$, the Shapley values for the training game $v$ (Equation (3)), using an estimator $\hat{\phi}$. Using linearity of the Shapley operator and the triangle inequality, we decompose the attribution error as

$$\|\hat{\phi} - \phi(v)\|_2 = \|\phi(\hat{f}_\omega) - \phi(v)\|_2$$
$$\leq \underbrace{\|\phi(\hat{f}_\omega - \hat{v}_\theta)\|_2}_{\mathcal{E}_{\mathrm{sur}}} + \underbrace{\|\phi(\hat{v}_\theta - v)\|_2}_{\mathcal{E}_{\mathrm{gap}}}. \quad (12)$$

The term $\mathcal{E}_{\mathrm{sur}}$ captures the surrogate approximation and estimation error, including function-class bias and variance from fitting $\hat{f}_\omega$ on $M$ sampled coalitions. The term $\mathcal{E}_{\mathrm{gap}}$ captures the discrepancy between the proxy game and the retraining game.

In summary, SurrogateSHAP (Algorithm 1) enables scalable attribution by (i) reformulating the training game as a training-free proxy game, (ii) approximating utilities via a GBT, and (iii) computing Shapley values via TreeSHAP.

## 5. Experimental Setup & Results

We evaluate SurrogateSHAP through: (i) the fidelity of the proxy game to the training game, (ii) sample efficiency and convergence for Shapley estimation, (iii) the Linear Datamodel Score (LDS), and (iv) counterfactual evaluation.

### 5.1. Datasets & Models

**CIFAR-20 ($32 \times 32$).** Following Lu et al. (2025), we use CIFAR-20, a 20-class subset of CIFAR-100, where each contributor (labeler) corresponds to a distinct class (Krizhevsky et al., 2009). We evaluate a DDPM-CFG (Ho & Salimans,

---

**Algorithm 1** SurrogateSHAP

**Require:** Contributors $N$, target model $\theta$, utility $\mathcal{F}$, budget $M$, sampler $q_{\mathrm{KS}}$, hypothesis class $\mathcal{H}$.
**Ensure:** Estimated attributions $\hat{\phi} \in \mathbb{R}^n$.
1: Initialize $\mathcal{D}_{\mathrm{surr}} \leftarrow \emptyset$.
2: **for** $m = 1, \ldots, M$ **do**
3:     Sample $S_m \sim q_{\mathrm{KS}}$ and encode it as $\mathbf{z}_m \in \{0, 1\}^n$.
4:     Set $\mathcal{Y}_{S_m} \leftarrow \bigcup_{i \in S_m} \mathcal{Y}_i$.
5:     Define $P_{S_m}^\theta \leftarrow \sum_{y \in \mathcal{Y}_{S_m}} \pi_{S_m}(y) \, p_\theta(\mathbf{x} \mid \psi(y))$.
6:     Set $y_m \leftarrow \mathcal{F}(P_{S_m}^\theta)$.
7:     Add $(\mathbf{z}_m, y_m)$ to $\mathcal{D}_{\mathrm{surr}}$.
8: **end for**
9: Fit $\hat{f}_\omega \in \mathcal{H}$ on $\mathcal{D}_{\mathrm{surr}}$ by minimizing $\mathcal{L} + \Omega$.
10: Set $\hat{\phi} \leftarrow \mathrm{TreeSHAP}(\hat{f}_\omega, \mathbf{1}_n; \mathrm{baseline} = \mathbf{0}_n)$.
11: **return** $\hat{\phi}$.

---

2022), generating samples with a 50-step DDIM sampler. To assess model behavior, we report FID and Inception Score (IS).

**ArtBench (Post-Impressionism) ($256 \times 256$).** We use the post-impressionism subset of ArtBench (Liao et al., 2022), which contains 5,000 images from 258 artists, and treat each artist as a player. Each image has a corresponding prompt template ``a Post-Impressionist painting by {artist}''. We fine-tune Stable Diffusion v1.5 (Rombach et al., 2022). The model property is computed as the average aesthetic score with an aesthetic predictor[3].

**Fashion-Product ($256 \times 256$).** We build a dataset from the Fashion Product Images collection (Aggarwal, 2019), comprising 4,468 images from 100 brands. Each image is paired with a text prompt that includes the product description and brand name. We treat each brand as a player and define a player's data as the subset of images associated with that brand. We fine-tune FLUX-dev1 (Labs et al., 2025) using LoRA (rank 128). To assess fashion-specific behavior, we report LPIPS and diversity computed over generated samples (Moosaei et al., 2022).

As defined in Section 3.2, the concept of "training" encompasses a broad range of algorithms. Therefore, we use CIFAR-20 as a simplified analogue to T2I models for full-parameter retraining. For the ArtBench (Post-Impressionism) and Fashion-Product, we employ full fine-tuning and LoRA fine-tuning, respectively, as computationally efficient alternatives. Consequently, throughout the following sections, the term "training" specifically refers to full-parameter retraining for CIFAR-20, full fine-tuning for ArtBench, and LoRA fine-tuning for Fashion-Product.

---

[3]https://github.com/LAION-AI/aesthetic-predictor

*Table 1.* **Empirical fidelity of the proxy relative to training utility and runtime.** Sample-level fidelity is measured by pixel similarity (MAE, SSIM) and representation alignment (RMS in LPIPS, CLIP distance); utility error is reported via range-normalized NMAE and NRMSE. Time is average wall-clock minutes per subset query. Grey rows indicate the empirical bounds derived from independent training runs of the original game using different random seeds.

| | Pixel similarity | | Representation alignment | | Utility error | | |
| Method | MAE ↓ | SSIM ↑ | RMS CLIP ↓ | RMS LPIPS ↓ | NMAE ↓ | NRMSE ↓ | Time (min) ↓ |
|---|---|---|---|---|---|---|---|
| | | | | **CIFAR-20** | | | |
| Retraining | 0.104 | 0.802 | 0.403 | 0.098 | 0.023 | 0.053 | 234.36 |
| sFT | 0.145 | 0.639 | 0.496 | 0.129 | 0.079 | 0.150 | 27.42 |
| **Ours** | **0.105** | **0.779** | **0.452** | **0.108** | **0.037** | **0.071** | **10.72** |
| | | | | **ArtBench** | | | |
| Full ft | 0.104 | 0.601 | 0.427 | 0.337 | 0.057 | 0.082 | 128.83 |
| sFT | 0.179 | 0.441 | 0.571 | 0.479 | 0.101 | 0.166 | 14.15 |
| **Ours** | **0.147** | **0.531** | **0.529** | **0.458** | **0.068** | **0.124** | **10.92** |
| | | | | **Fashion-Product** | | | |
| LoRA ft | 0.121 | 0.871 | 0.399 | 0.267 | 0.087 | 0.116 | 105.24 |
| sFT | 0.186 | 0.772 | 0.458 | 0.337 | 0.198 | 0.296 | 54.33 |
| **Ours** | **0.177** | **0.786** | **0.429** | **0.301** | **0.192** | **0.265** | **35.23** |

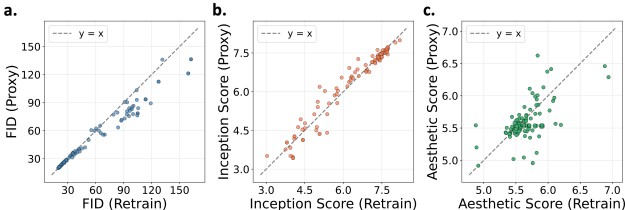

*Figure 1.* Alignment between the proxy utility $\hat{v}_\theta(S)$ (y-axis) and training utility $v(S)$ (x-axis) across subsets, for **(a)** FID, **(b)** Inception Score (IS), and **(c)** mean aesthetic score.

Please refer to Section C for datasets, training, and implementation details.

## 5.2. Proxy Game Fidelity

We start by evaluating the empirical faithfulness of the proxy $\hat{v}_\theta(S)$ relative to training utility $v(S)$ at two granularities. First, we measure paired sample agreement under a shared noise **z** and conditioning **c** using pixel-level metrics (MAE, SSIM) and RMS of feature-space distances (LPIPS, CLIP similarity). Second, at the utility level, we assess using NMAE and NMSE. These metrics are computed over 100 subsets sampled from the Shapley distribution, with utilities defined as described in Section 5.1. For reference, we also report results from independently trained instances of the original game using different random seeds, evaluated on the same subsets, which provide an empirical lower bound for Equation (7).

As shown in Table 1, our proposed proxy game achieves high pixel-level alignment with the training game. Across all benchmarks, it consistently exhibits lower representation drift (cf. Equation (7)) compared to other methods such as sparsified fine-tuning (sFT) (Lu et al., 2025). Specifically, our method achieves lower RMS LPIPS than sFT on CIFAR-20 (0.108 vs. 0.129), ArtBench Post-Impressionism (0.458 vs. 0.479), and Fashion-Product (0.301 vs. 0.337).

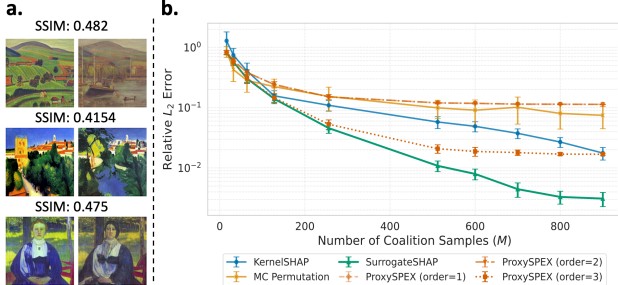

*Figure 2.* **(a)** Local samples and structural similarities between (re)trained models (left) and the target model (right) for three subsets. **(b)** Shapley estimator accuracy: $\ell_2$ error to the oracle versus subset budget on synthetic games (mean ± std over 60 independent trials).

Similar performance gains are observed in RMS CLIP, indicating that our method effectively preserves structural and semantic features relative to training instances (Figure 2a). This local stability translates into a tight global utility approximation. As shown in Figure 1 and the utility-level metrics (Table 1), our proxy tracks the training utility with an NRMSE ranging from 0.071 on CIFAR-20 to 0.265 on Fashion-Product. These findings empirically validate that the bound in equation 10 is effectively controlled in practice.

Computationally, because our proposed game is inference-only, it offers a significant speedup (Table 1). On CIFAR-20, ArtBench (Post-Impressionism), and Fashion-Product, it is $23.4\times$, $12.8\times$, and $2.9\times$ faster than explicit training, respectively. Compared to sFT, we achieve gains of $2.5\times$, $1.4\times$, and $1.5\times$ while bypassing the need for subset fine-tuning. Collectively, these gains in speed and proxy fidelity make large-scale subset queries for T2I attribution practical.

## 5.3. Sample Efficiency on Synthetic Games

To validate the Shapley estimation proposed in Section 4.2, we first evaluate its performance on synthetic coalition games ($n = 10$) characterized by non-linearities and up to three-way interactions (Section D). This controlled setting eliminates the proxy-training gap and enables the computation of exact Shapley values $\phi(v)$ via equation 4. We then measure the $\ell_2$ error relative to $\phi(v)$ across various query budgets $M$ for different Shapley estimators.

As shown in Figure 2b, SurrogateSHAP is substantially more sample-efficient than KernelSHAP (Lundberg & Lee, 2017) and ProxySPEX ($k \in 1, 2$) (Butler et al., 2025) in the interaction-heavy setting. At a budget of $M = 512$, kernelSHAP maintains a $\ell_2$ error of 0.085. In contrast, SurrogateSHAP reaches an error of 0.010 with the same budget. This reduction in error translates directly to better axiomatic fidelity; as $M$ increases, our method more consistently satisfies properties like efficiency and the null-player axiom compared to ProxySPEX (Section D.3). While KernelSHAP

is effective in purely additive regimes (Figure S.1a), its performance degrades if the coalition utility function involves non-linearity and interaction (Figure S.1b, Figure 2). These results indicate that when utility functions involve complex interactions and query budgets are limited, SurrogateSHAP provides superior sample efficiency and estimation accuracy.

## 5.4. Evaluating SurrogateSHAP with Linear Datamodel Score (LDS)

In this section, we evaluate SurrogateSHAP in attribution tasks for T2I diffusion models through Linear Datamodel Score (LDS) (Ilyas et al., 2022), which measures how well an attribution method predicts utilities of unseen coalitions. Formally, for a coalition $S \subseteq [n]$ and an attribution method $\tau$, its additive score is $g(S; \tau) = \sum_{i \in S} \tau_i$, and LDS is defined as:

$$\text{LDS}(\tau; \alpha) \coloneqq \rho\big(\{g(S; \tau)\}_{S \in \mathcal{S}_\alpha}, \{v(S)\}_{S \in \mathcal{S}_\alpha}\big), \quad (13)$$

where $\rho$ is Spearman correlation (Spearman, 1961) and $\mathcal{S}_\alpha$ is a held-out subset sampled uniformly from $\{S \subseteq [n] : |S| = \alpha n\}$. We consider $\alpha \in \{0.25, 0.5, 0.75\}$. For each $\alpha$, we sample 100 subsets, train the model (retrain, full fine-tune, or fine-tune with LoRA as described in Section 5.1), and compute $v(S)$ following Section 5.1. We report the mean LDS with 95% confidence intervals over five seeds.

**Baseline Methods.** We group baseline attribution methods into four categories: (i) similarity-based methods; (ii) leave-one-out and its approximations (e.g., influence functions (Koh & Liang, 2017) and TRAK-based methods (Park et al., 2023)); (iii) application-specific heuristics (e.g., aesthetic scores for ArtBench Post-Impressionism); and (iv) training proxies (Lu et al., 2025).[4] For TRAK-based methods, gradients are computed at the target model $\theta$ following Zheng et al. (2023). For local attribution methods, we averaged sample-level scores over images generated by $\theta$ to obtain global scores. The aggregation strategy at the contributor level depends on the method: we sum datum-level attributions for influence-function and TRAK methods (the principled aggregation (Koh et al., 2019)), while for similarity-based and application-specific baselines, we aggregate by either the mean or the maximum. Additional details for baseline implementations are in Section A.

**Results.** Table 2 summarizes LDS ($\alpha = 0.5$) performance for each method. Our results reveal that TRAK and influence-function (IF) variants frequently underperform LOO estimates. In ArtBench (Post-Impressionism), these methods exhibit negative correlations, most notably D-TRAK ($-20.75\%$ LDS). Similarity-based methods (e.g., CLIP similarity, GMValuator (Yang et al., 2025)) are often inconsistent across datasets and utilities. Such results

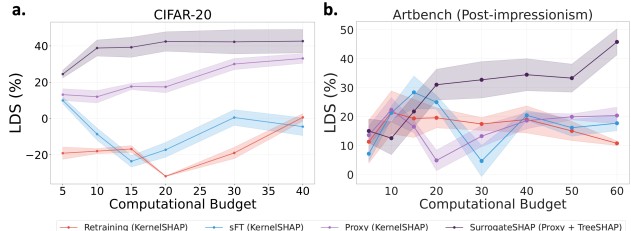

*Figure 3.* LDS (%) at $\alpha = 0.5$ for Shapley-based methods, evaluated on **(a)** FID and **(b)** average aesthetic score. Computational budget is measured in units of training-and-inference runtime.

suggest that these baseline methods can be misleading regarding the impact of data contributors on generative quality. In contrast, SurrogateSHAP shows consistent and superior performance across diverse utility functions, ranging from distributional metrics (FID) to perceptual ones (Avg. Aesthetic, LPIPS). Within the Shapley-based framework, SurrogateSHAP outperforms sFT (Lu et al., 2025) by a substantial margin. On CIFAR-20, it achieves 51.21% LDS for FID, whereas sFT struggles and underperforms other baselines. In ArtBench (Post-Impressionism), SurrogateSHAP reaches 44.14% LDS, more than doubling sFT. SurrogateSHAP consistently outperforms other methods across various datamodel $\alpha \in \{0.25, 0.75\}$ (see Tables S.5 and S.6).

We next evaluate the efficiency-performance trade-off within the Shapley-based framework by measuring LDS across varying computational budgets. For SurrogateSHAP, we omit the GBT training time from the total budget, as it is negligible compared to the total cost of subset queries[5]. Across all tested budgets, SurrogateSHAP achieves the highest LDS and demonstrates better scaling trends on CIFAR-20 and ArtBench (Post-Impressionism) (Figure 3). For CIFAR-20, our method exceeds baseline performance at low budgets ($< 10$ training units) and scales consistently as compute increases; in contrast, training and sFT frequently plateau or exhibit high variance. For FID at budget 10, our method reaches an LDS of $41.2\%$ whereas the training baseline and sFT are below 0% (Figure 3a). We also compare TreeSHAP-based estimates (dark purple) with KernelSHAP (light purple) using the same proxy subsets, and our approach outperforms the KernelSHAP estimate by a significant margin. These results confirm that SurrogateSHAP provides computationally efficient, reliable estimates of held-out set utilities.

## 5.5. Counterfactual Evaluation

To complement the rank-based LDS evaluation, we perform counterfactual evaluation (Hooker et al., 2019) to verify the

---

[4]We exclude TracIn (Pruthi et al., 2020) since intermediate checkpoints are often unavailable in practice.

[5]In Table 2, the average runtime of 5-fold CV is 0.86s for CIFAR-20, 1.65s for ArtBench Post-impressionism, and 26.62s for Fashion-Product.

*Table 2.* LDS (%) results with $\alpha = 0.5$. Means and 95% confidence intervals across five random initializations are reported.

| Method | CIFAR-20 | | ArtBench (Post-Impressionism) | Fashion Product | |
|---|---|---|---|---|---|
| | Inception score | FID | Aesthetic score | Diversity | LPIPS |
| Pixel similarity (avg.) | $-8.88 \pm 2.80$ | $13.74 \pm 7.44$ | $-5.43 \pm 5.57$ | $10.16 \pm 6.09$ | $-14.90 \pm 4.90$ |
| Pixel similarity (max) | $-23.20 \pm 3.16$ | $8.37 \pm 1.76$ | $5.44 \pm 4.49$ | $-21.40 \pm 7.11$ | $22.30 \pm 14.37$ |
| CLIP similarity (avg.) | $-21.49 \pm 5.10$ | $20.13 \pm 6.40$ | $5.60 \pm 9.09$ | $14.84 \pm 7.72$ | $-4.19 \pm 16.49$ |
| CLIP similarity (max) | $20.36 \pm 4.04$ | $-17.79 \pm 1.03$ | $3.62 \pm 6.65$ | $1.94 \pm 10.62$ | $8.97 \pm 2.69$ |
| Gradient similarity (avg.) | $13.29 \pm 2.84$ | $-11.47 \pm 1.88$ | $5.73 \pm 5.73$ | $19.41 \pm 8.38$ | $-23.21 \pm 12.02$ |
| Gradient similarity (max) | $-20.70 \pm 3.64$ | $12.54 \pm 1.85$ | $5.73 \pm 2.79$ | $18.47 \pm 11.87$ | $-23.22 \pm 8.70$ |
| GMValuator (Yang et al., 2025) | $27.85 \pm 3.17$ | $5.37 \pm 6.20$ | $3.49 \pm 4.24$ | $-2.86 \pm 7.86$ | $-12.75 \pm 0.20$ |
| Aesthetic score (avg.) | – | – | $6.91 \pm 3.79$ | – | – |
| Aesthetic score (max) | – | – | $6.71 \pm 4.28$ | – | – |
| Relative IF (Barshan et al., 2020) | $23.88 \pm 1.55$ | $13.29 \pm 2.84$ | $-3.01 \pm 4.07$ | $7.23 \pm 5.97$ | $12.52 \pm 1.65$ |
| Renormalized IF (Hammoudeh & Lowd, 2022) | $22.74 \pm 4.61$ | $12.82 \pm 2.70$ | $-5.84 \pm 3.99$ | $2.45 \pm 5.25$ | $16.02 \pm 2.86$ |
| TRAK (Park et al., 2023) | $24.08 \pm 1.45$ | $12.84 \pm 2.76$ | $-5.34 \pm 3.96$ | $6.27 \pm 5.86$ | $18.52 \pm 2.31$ |
| Journey-TRAK (Georgiev et al., 2023) | $10.80 \pm 3.03$ | $-34.42 \pm 1.49$ | $6.64 \pm 6.36$ | $6.65 \pm 16.34$ | $-0.40 \pm 16.17$ |
| D-TRAK (Zheng et al., 2023) | $12.82 \pm 4.11$ | $9.38 \pm 0.33$ | $-8.75 \pm 2.52$ | $17.61 \pm 5.71$ | $15.82 \pm 6.67$ |
| DAS (Lin et al., 2024) | $24.70 \pm 1.50$ | $13.59 \pm 2.71$ | $-5.66 \pm 4.09$ | $6.35 \pm 6.10$ | $12.31 \pm 2.37$ |
| Leave-one-out (LOO) | $48.31 \pm 1.11$ | $34.82 \pm 0.92$ | $-0.18 \pm 3.19$ | $-3.88 \pm 3.00$ | $-1.50 \pm 0.71$ |
| Sparsified-FT Shapley (Lu et al., 2025) | $40.0 \pm 2.25$ | $-14.1 \pm 3.51$ | $18.85 \pm 1.89$ | $20.34 \pm 12.43$ | $18.57 \pm 6.21$ |
| SurrogateSHAP (**Ours**) | $\mathbf{79.7 \pm 2.20}$ | $\mathbf{51.21 \pm 6.78}$ | $\mathbf{39.0 \pm 5.27}$ | $\mathbf{32.41 \pm 5.31}$ | $\mathbf{42.60 \pm 0.81}$ |

Sparsified-FT and SurrogateSHAP use the same number of sampled coalitions ($M$): 600 for CIFAR-20, 700 for ArtBench, and 800 for Fashion-Product.

*Table 3.* **Counterfactual performance after bottom removal.** We report the relative change (%) in $\Delta\mathcal{F}$ after removing the bottom $k\%$ most influential groups identified by each method. Results are mean $\pm$ std over five random seeds.

| Method | 5% | 10% | 15% | 20% |
|---|---|---|---|---|
| **CIFAR-20 (FID ↓)** | | | | |
| CLIP Score | $19.39 \pm 3.20$ | $17.36 \pm 2.86$ | $17.21 \pm 3.22$ | $16.98 \pm 3.66$ |
| LOO | $23.21 \pm 4.76$ | $27.85 \pm 2.27$ | $32.22 \pm 3.65$ | $43.11 \pm 2.02$ |
| SurrogateSHAP (Ours) | $11.79 \pm 1.47$ | $11.85 \pm 2.55$ | $12.75 \pm 3.34$ | $16.98 \pm 3.67$ |
| **ArtBench (Post-Impressionism) (Aesthetic Score ↑)** | | | | |
| Avg. Aesthetic Score | $-2.59 \pm 1.44$ | $-1.38 \pm 1.16$ | $-2.22 \pm 1.06$ | $-2.40 \pm 0.42$ |
| Avg. Pixel Similarity | $-1.32 \pm 1.05$ | $-1.68 \pm 1.78$ | $-0.80 \pm 1.07$ | $-2.20 \pm 1.13$ |
| Relative Influence | $-2.93 \pm 2.72$ | $-1.25 \pm 2.28$ | $1.08 \pm 0.94$ | $0.17 \pm 0.65$ |
| SurrogateSHAP (Ours) | $0.43 \pm 1.67$ | $2.54 \pm 2.38$ | $3.06 \pm 0.38$ | $3.62 \pm 2.37$ |
| **Fashion Product (Diversity ↑)** | | | | |
| Avg. GradSim | $3.19 \pm 4.98$ | $-6.53 \pm 2.44$ | $4.83 \pm 3.54$ | $6.70 \pm 6.17$ |
| D-TRAK | $-1.12 \pm 4.25$ | $-4.49 \pm 4.98$ | $-0.50 \pm 2.14$ | $-7.24 \pm 5.95$ |
| SurrogateSHAP (Ours) | $9.36 \pm 3.46$ | $5.22 \pm 5.01$ | $12.75 \pm 3.34$ | $7.46 \pm 4.44$ |

influence of identified contributors. We measure the relative change in model property, $\Delta\mathcal{F} := \frac{\mathcal{F}(p_{\theta_K^*}) - \mathcal{F}(p_\theta)}{\mathcal{F}(p_\theta)}$, where $\theta_K^*$ is model retrained after excluding or retaining the top-$K$ contributors. A high-fidelity attribution method should maximize this change (degradation or preservation) for a given $K$. We compare SurrogateSHAP against the best-performing baselines within each category from Table 2.

Upon bottom-removal, Table 3 shows that removing low-ranked groups can enable dataset pruning with minimal impact or even performance gains. SurrogateSHAP yields the most consistent improvements across percentiles. For example, removing low-ranked groups on ArtBench (Post-Impressionism) improves the aesthetic score by 3.62% at 20% removal. On Fashion Product, diversity improves by 12.75% at 15% removal, whereas baselines often degrade the same metrics (e.g., Avg. GradSim: $-6.53\%$). A similar pattern holds for CIFAR-20, where our method yields the

smallest FID increases under bottom removal (11.79% at 5% removal). These findings suggest that SurrogateSHAP is an effective data pruning method that filters out non-essential instances without compromising downstream generative utility.

In contrast, under top-removal counterfactuals (Table S.8), SurrogateSHAP induces the largest degradations in generative quality. On CIFAR-20, removing the top 10% of groups increases FID by 31.70% (vs. CLIP Score 17.53% and LOO 17.48%) and decreases the Inception Score by 26.55% (vs. LOO 22.45% and GMValuator 19.52%) (Table S.10). Similarly, on ArtBench (Post-Impressionism), removing the top 20% results in a 7.71% drop in aesthetic score, exceeding all baselines. As shown in Figure S.17, SurrogateSHAP identifies the most visually salient classes as top contributors to FID on CIFAR-20, and highlights ArtBench (Post-Impressionism) artists whose works are warm-toned and atmospheric (Figure S.21), indicating it captures the most influential contributors.

Overall, these counterfactual results, combined with LDS analysis, confirm that SurrogateSHAP provides accurate attribution effectively and identifies influential contributors.

# 6. Case Study: Dermatology Data Auditing

Finally, we showcase the clinical utility of SurrogateSHAP by auditing a Stable Diffusion model fine-tuned on the ISIC dataset[6], a multi-site collection of dermoscopic images from ten hospitals. As modern generative models are trained on multi-site clinical data (Ktena et al., 2024; Koetzier et al.,

---

[6] https://www.isic-archive.com/

**a.**

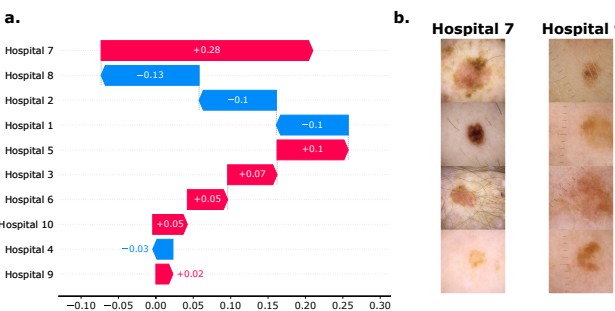

**b.**   **Hospital 7**   **Hospital 9**

*Figure 4.* **(a)** SurrogateSHAP value for ten hospitals (x-axis: magnitude; y-axis: site; red/blue: positive/negative). **(b)** Examples of dermoscopic images with visual artifacts.

2024), especially in scenarios like federated learning (Zhang et al., 2021) where the training data is not directly accessible, they risk inheriting site-specific acquisition protocols that exacerbate undesired behaviors, e.g., non-biological association between sex and specific diseases (Sagers et al., 2023; Ktena et al., 2024). To address this, we define the utility function $v(S)$ as the Spearman correlation between the outputs of two classifiers, one trained to predict biological sex and another for melanoma, on the model's generated images (Section F). This utility quantifies the learned correlation between sex and melanoma diagnosis, which should ideally be near zero. However, our results reveal a significant correlation ($r = 0.21, p = 3.2 \times 10^{-7}$).

To localize the source of this bias, we attribute the correlation to individual hospitals using SurrogateSHAP; as shown in Figure 4a, it identifies Hospital 7 as the primary driver of the spurious correlation, while Hospital 9 contributes the least. We subsequently inspected the source datasets from these hospitals and found that images from Hospital 7 exhibit a high prevalence of hair artifacts, which correlate with male patients (Gadgil et al., 2025) and possess dark pigmentation associated with melanoma (Tsao et al., 2015) (Figure 4b). In contrast, Hospital 9 images frequently feature ruler markings, typically associated with benign lesions, and show no correlation with patient sex. Upon excluding Hospital 7 from the training data, the correlation in the resulting model's generations drops to nearly zero ($r = 5 \times 10^{-4}, p = 3.9 \times 10^{-4}$), effectively reducing the spurious association. In contrast, removing the hospital with the highest correlation between the binary sex and melanoma labels in the training set (Figure S.27) results in only a marginal decrease in the generated image correlation ($r = 0.11, p = 1.4 \times 10^{-3}$). This disparity suggests that there are spurious correlations induced by inter-hospital interactions that are only captured by SurrogateSHAP.

# 7. Conclusion

In this work, we introduce SurrogateSHAP, a framework designed to enable efficient and scalable contributor valuation for T2I diffusion models. By reformulating the expensive training game as a training-free proxy game and approximate subset evaluations via tree-based surrogates, we enable efficient and accurate Shapley estimation without subset-specific training. This approach not only significantly enhances subset query speed but also improves the sample efficiency required for stable Shapley estimation, making contributor valuation practical for large-scale models. Our experiments across CIFAR-20, ArtBench (Post-Impressionism), and Fashion-Product demonstrate that SurrogateSHAP provides accurate estimates of influence, enabling scalable and reliable attribution of data contributors.

# Acknowledgement

We thank members of the Lee lab for providing feedback on this project and the reviewers for their constructive comments. This work was funded by the National Institutes of Health [R01 AG061132, R01 EB035934, RF1 AG088824].

# Impact Statement

This work introduces a training-free group-attribution framework for text-to-image diffusion models, enabling efficient estimation of contributor influence on various model behaviors of interest.

The primary positive impact is the democratization of model auditing. By eliminating the computational burden of retraining-based methods, our framework allows researchers with limited compute resources to perform transparency audits on large-scale models. This facilitates fair credit assignment for data contributors and supports principled dataset curation, which can lead to more efficient and specialized generative systems. Additionally, as demonstrated in our clinical case study, SurrogateSHAP identifies biased data providers with high precision. This provides a critical tool for identifying and mitigating systematic disparities in healthcare AI, ensuring that generative technologies are deployed equitably and safely across diverse populations.

While our method enhances transparency, it carries risks if misused as a sole arbiter for data removal. Because attribution scores are metric-specific, optimizing purely for generative quality (e.g., FID) might lead to the systematic pruning of "outlier" data representing minority groups or marginalized artistic styles, thereby amplifying representation bias. Furthermore, because SurrogateSHAP operates under a one-to-one mapping between contributors and labels, it is most effectively deployed in settings without overlapping prompt spaces. We emphasize that this framework

should be used as a diagnostic tool within a broader multi-objective governance pipeline, supplemented by qualitative safety audits to ensure that data curation decisions do not suppress diverse or underrepresented content.

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

# Appendix.

# A. Data Attribution Methods for Diffusion Models

Here we summarize data attribution methods for diffusion models and provide implementation details for the baselines used in Section 5.4. For local attribution methods, we compute sample-level attributions for images generated by $\theta$ and average them to obtain global scores. Contributor-level aggregation depends on the method: for variants of influence-function (IF) and TRAK, we aggregate by summing datum-level attributions; for similarity-based and application-specific baselines, we aggregate using either the mean or the maximum.

## A.1. Similarity-based Methods

**Raw pixel similarity.** This is a simple baseline that uses each raw image as the representation and then computes the cosine similarity between the generated sample of interest and each training sample as the attribution score.

**CLIP similarity.** This method utilizes CLIP[7] (Radford et al., 2021) to encode each sample into an embedding and then compute the cosine similarity between the generated sample of interest and each training sample as the attribution score.

**GMValuator.** Yang et al. (2025) propose a data valuation method for generative models. Given training data $X = \{x_i\}_{i=1}^n$ and generated samples $\hat{X} = \{\hat{x}_j\}_{j=1}^m$, each $x_i$ is valued by its similarity to $\hat{X}$. Using a distance $d_{ij} = d(x_i, \hat{x}_j)$, the (normalized) contribution to $\hat{x}_j$ is

$$V(x_i, \hat{x}_j) = \frac{\exp(-d_{ij})}{\sum_{i'=1}^n \exp(-d_{i'j})}, \tag{14}$$

and the value of $x_i$ is $\phi_i = \sum_{j=1}^m V(x_i, \hat{x}_j)$. For efficiency, GMValuator considers only the top-$k$ nearest neighbors $P_j$ of each $\hat{x}_j$ and weights by a generation-quality score $q_j \in [0, 1]$:

$$V(x_i, \hat{x}_j) = \begin{cases} q_j \cdot \dfrac{\exp(-d_{ij})}{\sum\limits_{x \in P_j} \exp\big(-d(x, \hat{x}_j)\big)}, & x_i \in P_j, \\ 0, & \text{otherwise.} \end{cases} \tag{15}$$

In our implementation, we use LPIPS for $d(\cdot, \cdot)$ and set $k = 50$.

## A.2. Influence Functions and TRAKs

**Leave-one-out.** Leave-one-out (LOO) evaluates the change in model property due to the removal of a single data contributor $i$ from the training set (Koh & Liang, 2017). Formally, it is defined as:

$$\tau_{\text{LOO}}(\mathcal{F}, \mathcal{D})_i = \mathcal{F}(\theta^*) - \mathcal{F}(\theta^*_{\mathcal{D}\setminus\mathcal{C}_i}), \tag{16}$$

where $\mathcal{F}(\theta^*)$ denotes the global property for the model trained with the full dataset, and $\mathcal{F}(\theta^*_{\mathcal{D}\setminus\mathcal{C}_i})$ represents the global property of the model trained with the data provided by the $i$th contributor.

**Gradient similarity.** This is a gradient-based influence estimator (Charpiat et al., 2019), which computes the cosine similarity using the gradient representations of the generated sample of interest, $\tilde{x}$ and each training sample $\mathbf{x}^{(j)}$ as the attribution score:

$$\frac{\mathcal{P}^\top \nabla_\theta \mathcal{L}_{\text{Simple}}(\mathbf{x}^{(j)}; \theta^*) \cdot \mathcal{P}^\top \nabla_\theta \mathcal{L}_{\text{Simple}}(\tilde{\mathbf{x}}; \theta^*)}{\|\mathcal{P}^\top \nabla_\theta \mathcal{L}_{\text{Simple}}(\mathbf{x}^{(j)}; \theta^*)^\top\| \|\mathcal{P}^\top \nabla_\theta \mathcal{L}_{\text{Simple}}(\tilde{\mathbf{x}}; \theta^*)\|} \tag{17}$$

**TRAK.** Park et al. (2023) propose an approach that aims to enhance the efficiency and scalability of data attribution for classifiers. Zheng et al. (2023) adapt TRAK for attributing images generated by diffusion models, defined as follows:

$$\frac{1}{S} \sum_{s=1}^S \Phi_s \left(\Phi_s^\top \Phi_s + \lambda I\right)^{-1} \mathcal{P}_s^\top \nabla_\theta \mathcal{L}_{\text{Simple}}(\tilde{\mathbf{x}}; \theta_s^*), \text{ and} \tag{18}$$

$$\Phi_s = \left[\phi_s(\mathbf{x}^{(1)}), \ldots, \phi_s(\mathbf{x}^{(N)})\right]^\top, \text{ where } \phi_s(\mathbf{x}) = \mathcal{P}_s^\top \nabla_\theta \mathcal{L}_{\text{Simple}}(\mathbf{x}; \theta_s^*), \tag{19}$$

---

[7]https://github.com/openai/CLIP

where $\mathcal{L}_{\text{Simple}}$ is the diffusion loss used during training, $\mathcal{P}_s$ is a random projection matrix, and $\lambda I$ serves for numerical stability and regularization. For TRAK-based approaches, we follow the implementation details in Zheng et al. (2023)[8].

**D-TRAK.** Zheng et al. (2023) find that, compared to the theoretically motivated setting where the loss function is defined as $\mathcal{L} = \mathcal{L}_{\text{Simple}}$, using alternative functions, such as the squared loss $\mathcal{L}_{\text{Square}}$, to replace both the loss function and local model property result in improved performance:

$$\frac{1}{S} \sum_{s=1}^{S} \Phi_s \left( \Phi_s^\top \Phi_s + \lambda I \right)^{-1} \mathcal{P}_s^\top \nabla_\theta \mathcal{L}_{\text{Square}}(\tilde{\mathbf{x}}; \theta_s^*), \text{ and} \tag{20}$$

$$\Phi_s = \left[ \phi_s(\mathbf{x}^{(1)}), \dots, \phi_s(\mathbf{x}^{(N)}) \right]^\top, \text{ where } \phi_s(\mathbf{x}) = \mathcal{P}_s^\top \nabla_\theta \mathcal{L}_{\text{Square}}(\mathbf{x}; \theta_s^*). \tag{21}$$

**Journey-TRAK.** (Georgiev et al., 2023) focus on attributing noisy images $\mathbf{x}_t$ during the denoising process. It attributes training data not only for the final generated sample but also for the intermediate noisy samples throughout the denoising process. At each time step, the reconstruction loss is treated as a local model property. Following (Zheng et al., 2023), we compute attribution scores by averaging over the generation time steps, as shown below:

$$\frac{1}{T'} \frac{1}{S} \sum_{t=1}^{T'} \sum_{s=1}^{S} \Phi_s \left( \Phi_s^\top \Phi_s + \lambda I \right)^{-1} \mathcal{P}_s^\top \nabla_\theta \mathcal{L}_{\text{Simple}}^t(\tilde{\mathbf{x}}_t; \theta_s^*), \text{ and} \tag{22}$$

$$\Phi_s = \left[ \phi_s(\mathbf{x}^{(1)}), \dots, \phi_s(\mathbf{x}^{(N)}) \right]^\top, \text{ where } \phi_s(\mathbf{x}) = \mathcal{P}_s^\top \nabla_\theta \mathcal{L}_{\text{Simple}}(\mathbf{x}; \theta_s^*). \tag{23}$$

Here, $\mathcal{L}_{\text{Simple}}^t$ is the diffusion loss restricted to the time step $t$ (hence without taking the expectation over the time step).

In practice, we follow the main implementation by Zheng et al. (2023) that considers TRAK, D-TRAK, and Journey-TRAK as retraining-free methods. That is, $S = 1$, and $\theta_1^* = \theta^*$ is the original diffusion model trained on the entire dataset.

**DAS.** Lin et al. (2024) propose the *Diffusion Attribution Score* (DAS), which quantifies the influence of a training sample by comparing the model's predicted distributions before vs. after removing that sample. For a target generated example $\mathbf{x}_{\text{gen}}$, DAS defines the attribution of $\mathbf{x}^{(i)}$ as the KL divergence between the original model and the leave-one-out model. For diffusion models, this is approximated by the expected squared difference between noise-predictor outputs:

$$\tau_{\text{DAS}}(\mathbf{x}_{\text{gen}}, \mathcal{D})^{(i)} \approx D_{\text{KL}}\Big( p_\theta(\mathbf{x}_{\text{gen}}) \,\big\|\, p_{\theta \backslash i}(\mathbf{x}_{\text{gen}}) \Big) \approx \mathbb{E}_{\epsilon, t}\Big[ \big\| \epsilon_\theta(\mathbf{x}_t^{\text{gen}}, t) - \epsilon_{\theta \backslash i}(\mathbf{x}_t^{\text{gen}}, t) \big\|_2^2 \Big]. \tag{24}$$

To avoid retraining, they linearize the noise predictor around $\theta^*$ and approximate the parameter change $\theta^* - \theta^{\backslash i}$ via a one-step Newton update. Using the Sherman–Morrison formula, they derive a closed-form attribution at timestep $t$:

$$\tau_{\text{DAS},t}(\mathbf{x}_{\text{gen}}, \mathcal{D})^{(i)} = \mathbb{E}_\epsilon \left[ \left\| \frac{g_t(\mathbf{x}_{\text{gen}})^\top \left( G_t(\mathcal{D})^\top G_t(\mathcal{D}) \right)^{-1} g_t(\mathbf{x}^{(i)}) \, r_t^{(i)}}{1 - g_t(\mathbf{x}^{(i)})^\top \left( G_t(\mathcal{D})^\top G_t(\mathcal{D}) \right)^{-1} g_t(\mathbf{x}^{(i)})} \right\|_2^2 \right], \tag{25}$$

where $g_t(\mathbf{x}) = \nabla_\theta \epsilon_{\theta^*}(\mathbf{x}_t, t)$, $G_t(\mathcal{D}) = \nabla_\theta \epsilon_{\theta^*}(\mathcal{D}_t, t)$, and $r_t^{(i)}$ is the residual term from the noise prediction. Here, $\mathcal{D} = \{\mathbf{x}^{(j)}\}_{j=1}^N$ denotes the training dataset, and all quantities are evaluated at the final weights $\theta^*$. In practice, we follow the implementation from Lin et al. (2024)[9] which treats DAS as a retraining-free method using the final weights $\theta^*$.

**Relative Influence.** Proposed by Barshan et al. (2020), the $\theta$-relative influence functions estimator normalizes the influence functions estimator of Koh & Liang (2017) by the HVP magnitude. Following Zheng et al. (2023), we combine this estimator with TRAK dimension reduction. The attribution for each training sample $\mathbf{x}^{(j)}$ is as follows:

$$\frac{\mathcal{P}^\top \nabla_\theta \mathcal{L}_{\text{Simple}}(\tilde{\mathbf{x}}; \theta^*) \cdot \left( \Phi_{\text{TRAK}}^\top \cdot \Phi_{\text{TRAK}} + \lambda I \right)^{-1} \cdot \mathcal{P}^\top \nabla_\theta \mathcal{L}_{\text{Simple}}(\mathbf{x}^{(j)}; \theta^*)}{\left\| \left( \Phi_{\text{TRAK}}^\top \cdot \Phi_{\text{TRAK}} + \lambda I \right)^{-1} \cdot \mathcal{P}^\top \nabla_\theta \mathcal{L}_{\text{Simple}}(\mathbf{x}^{(j)}; \theta^*) \right\|}, \tag{26}$$

---

[8]https://github.com/sail-sg/D-TRAK/
[9]https://github.com/Jinxu-Lin/DAS/

and

$$\Phi_{\text{TRAK}} = \left[ \phi(\mathbf{x}^{(1)}), \ldots, \phi(\mathbf{x}^{(N)}) \right]^{\top}, \text{ where } \phi(\mathbf{x}) = \mathcal{P}^{\top} \nabla_{\theta} \mathcal{L}_{\text{Simple}}(\mathbf{x}; \theta^*). \tag{27}$$

**Renormalized Influence.** Introduced by Hammoudeh & Lowd (2022), this method renormalizes the influence functions by the magnitude of the training sample's gradients. Similar to relative influence, we made an adaptation following Zheng et al. (2023). The attribution for each training sample $\mathbf{x}^{(j)}$ is as follows:

$$\frac{\mathcal{P}^{\top} \nabla_{\theta} \mathcal{L}_{\text{Simple}}(\tilde{\mathbf{x}}; \theta^*) \cdot \left( \Phi_{\text{TRAK}}^{\top} \cdot \Phi_{\text{TRAK}} + \lambda I \right)^{-1} \cdot \mathcal{P}^{\top} \nabla_{\theta} \mathcal{L}_{\text{Simple}}(\mathbf{x}^{(j)}; \theta^*)}{\| \mathcal{P}^{\top} \nabla_{\theta} \mathcal{L}_{\text{Simple}}(\mathbf{x}^{(j)}; \theta^*) \|}, \tag{28}$$

and

$$\Phi_{\text{TRAK}} = \left[ \phi(\mathbf{x}^{(1)}), \ldots, \phi(\mathbf{x}^{(N)}) \right]^{\top}, \text{ where } \phi(\mathbf{x}) = \mathcal{P}^{\top} \nabla_{\theta} \mathcal{L}_{\text{Simple}}(\mathbf{x}; \theta^*). \tag{29}$$

For TRAK-based methods and gradient similarity, we compute gradients using the final model checkpoint, meaning that $S = 1$ in Equation (18). We select 100 time steps evenly spaced within the interval $[1, T]$ for computing the diffusion loss. At each time step, we introduce one instance of noise. The projection dimensions are set to $k = 4096$ for CIFAR-20 and $k = 32768$ for ArtBench (Post-Impressionism) and Fashion-Product. For D-TRAK, we use the best-performing output function $\mathcal{L}_{\text{Square}}$ and choose $\lambda = 5e^{-1}$ as in Zheng et al. (2023).

**Generalized Group Data Attribution (GGDA)** Ley et al. (2024) propose GGDA as a unified framework for attributing model behavior to groups of training examples. In its leave-one-out instantiation, GGDA measures a group's attribution as the change in a target property between a model trained on the full dataset and a model retrained with that group removed, i.e., $\mathcal{F}(\theta_D) - \mathcal{F}(\theta_{D \setminus \mathcal{C}_i})$, which is precisely a group leave-one-out effect. Accordingly, in our experiments, we report group leave-one-out estimates. Ley et al. (2024) also considers forming groups via unsupervised methods (e.g., via $k$-means) to improve efficiency; however, such groupings do not generally coincide with our pre-defined contributor groups.

**TracIn.** Pruthi et al. (2020) propose to attribute the training sample's influence based on first-order approximation with saved checkpoints during the training process. However, this is also a major limitation because sometimes it is impossible to obtain checkpoints for models such as Stable Diffusion (Rombach et al., 2022) and FLUX (Labs et al., 2025).

### A.3. Retraining-based Methods

**Sparsified fine-tuning (sFT).** Lu et al. (2025) propose sparsified fine-tuning, which first prunes a trained target model and then fine-tunes the pruned model on sampled data subsets. Their approach relies on the approximation

$$\mathcal{F}(p_{\tilde{\theta}_{S,k}^{\text{ft}}}) \approx \mathcal{F}(p_{\theta_S^*}), \tag{30}$$

where $\tilde{\theta}_{S,k}^{\text{ft}}$ denotes the sparsified-and-fine-tuned model for subset $S$ with fine-tuning steps $k$. Following their protocol[10], we apply magnitude-based pruning (Han et al., 2015) to produce sparse diffusion models: 10.4M→3.0M (CIFAR-20), 5.1M→2.6M (ArtBench, Post-Impressionism), and 358M→107.9M (Fashion-Product). We then fine-tune each sparsified model using the same training configuration (see Section C) to obtain a performant pruned model.

For unlearning on subsets $S$, we adopt the best-performing setting reported by Lu et al. (2025). Concretely, for each subset, we fine-tune the pruned model for 1,000 steps, using 50 epochs for CIFAR-20, 50 epochs for ArtBench (Post-Impressionism), and 16 epochs for Fashion-Product. We estimate Shapley values with KernelSHAP (Lundberg & Lee, 2017) implementation from Lu et al. (2025).

---

[10] https://github.com/suinleelab/An-Efficient-Framework-for-Crediting-Data-Contributors-of-Diffusion-Models

# B. Proofs

## B.1. Proof of Proposition 4.2

*Proof of Proposition 4.2.* Fix a coalition $S \subseteq N$ and the realized retrained parameters $\theta_S^*$. Let $\varphi_u : \mathcal{X} \to \mathbb{R}^{m_u}$ be the utility-aligned feature extractor. Define the $\pi$-marginal pushforward feature distributions:

$$\nu_S := \mathbb{E}_{\mathbf{c} \sim \pi}\left[\varphi_{u\#} p_{\theta_S^*}(\cdot \mid \mathbf{c})\right], \qquad \hat{\nu}_S := \mathbb{E}_{\mathbf{c} \sim \pi}\left[\varphi_{u\#} p_\theta(\cdot \mid \mathbf{c})\right].$$

By the definition of $\mathcal{F}$ in equation 9, $v(S) = \mathcal{V}(\nu_S)$ and $\hat{v}_\theta(S) = \mathcal{V}(\hat{\nu}_S)$. Since $\mathcal{V}$ is $L_u$-Lipschitz with respect to $W_p$ for $p \in \{1, 2\}$, we have

$$|v(S) - \hat{v}_\theta(S)| \le L_u \, W_p(\nu_S, \hat{\nu}_S). \tag{31}$$

To bound $W_p(\nu_S, \hat{\nu}_S)$, consider the shared-noise coupling $\gamma_{\mathbf{c}}$ from Assumption 4.1. For $\mathbf{c} \sim \pi$ and $\mathbf{z} \sim \mathcal{N}(0, I)$, let

$$Z := \varphi_u\big(\mathbf{x}_0^{\theta_S^*}(\mathbf{z}, \mathbf{c})\big), \qquad \hat{Z} := \varphi_u\big(\mathbf{x}_0^\theta(\mathbf{z}, \mathbf{c})\big).$$

Under $(\mathbf{c}, \mathbf{z}) \sim \pi \times \mathcal{N}(0, I)$, the marginals of $Z$ and $\hat{Z}$ are $\nu_S$ and $\hat{\nu}_S$, respectively; hence $(Z, \hat{Z})$ defines a coupling of $(\nu_S, \hat{\nu}_S)$. Using the definition of the Wasserstein distance as the infimum over all couplings (Peyré et al., 2019), and specializing to $p = 2$, we obtain

$$W_2(\nu_S, \hat{\nu}_S) \le \left(\mathbb{E}_{\mathbf{c}, \mathbf{z}}\Big[\|Z - \hat{Z}\|_2^2\Big]\right)^{1/2}$$
$$= \left(\mathbb{E}_{\mathbf{c} \sim \pi}\left[\mathbb{E}_{\mathbf{z}}\left[\left\|\varphi_u(\mathbf{x}_0^{\theta_S^*}(\mathbf{z}, \mathbf{c})) - \varphi_u(\mathbf{x}_0^\theta(\mathbf{z}, \mathbf{c}))\right\|_2^2\right]\right]\right)^{1/2}.$$

By Assumption 4.1, the inner expectation is bounded by $(\varepsilon^{\varphi_u})^2$ for all $\mathbf{c} \in \mathcal{C}$, hence $W_2(\nu_S, \hat{\nu}_S) \le \varepsilon^{\varphi_u}$. Since $W_1(\nu_S, \hat{\nu}_S) \le W_2(\nu_S, \hat{\nu}_S)$, we obtain $W_p(\nu_S, \hat{\nu}_S) \le \varepsilon^{\varphi_u}$ for $p \in \{1, 2\}$. Substituting into equation 31 yields

$$|v(S) - \hat{v}_\theta(S)| \le L_u \, \varepsilon^{\varphi_u}.$$

which concludes the proof. $\qquad \square$

# C. Experiment Setup & Implementation Details

## C.1. Datasets and Model Properties

**CIFAR-20 ($32 \times 32$).** We derive CIFAR-20 from CIFAR-100[11], which contains 50,000 training images across 100 fine-grained classes grouped into 20 superclasses (Krizhevsky et al., 2009). Following Lu et al. (2025) for computational feasibility, we restrict CIFAR-100 to four superclasses (`large carnivores`, `flowers`, `household electrical devices`, `vehicles 1`). The resulting dataset contains 10,000 training images from 20 fine-grained classes (5 per superclass). Unless otherwise stated, we treat each fine-grained class as a *player* (corresponding to a class-specific labeling unit) and define a player's data as all images belonging to that class.

*Model properties.* To quantify the quality of generated images, we report Inception Score (IS) (Salimans et al., 2016) and Fréchet Inception Distance (FID) (Heusel et al., 2017), which capture complementary aspects of sample quality. For both metrics, we generate 10,240 samples per evaluation and compute scores using standard PyTorch implementations: Inception Score[12] and FID[13].

**ArtBench Post-Impressionism ($256 \times 256$).** ArtBench-10 is a dataset for benchmarking artwork generation with 10 art styles, consisting of 5,000 training images per style (Liao et al., 2022). For our experiments, we consider the 5,000 training images corresponding to the art style "post-impressionism" from 258 artists.

*Model property.* We consider a downstream use case in which many images are generated, and only the most aesthetically pleasing outputs are retained. Following (Lu et al., 2025), we evaluate each model using an aesthetic predictor[14] by scoring 150 generated images and reporting the mean predicted score as the model property. The predictor is a linear head trained on top of CLIP embeddings and outputs a scalar proxy for perceived aesthetic quality.

**Fashion-Product-100** ($256 \times 256$). We use the Fashion Product Images dataset (Aggarwal, 2019), which provides product images with textual descriptions, categories, and brand names. We remove images containing human faces to avoid identity-related artifacts, then select the 100 brands with the largest remaining product counts. To enable efficient LDS evaluation, we uniformly subsample within each selected brand by retaining $25\%$ of its products, yielding a final dataset of 4,468 images across 100 brands. In this dataset, each *brand* is treated as a *player*, and a player's data consists of all retained products from that brand.

*Model properties.* Following Moosaei et al. (2022), we evaluate generated fashion-product images using two complementary metrics. First, we report LPIPS (Zhang et al., 2018), using training images as the reference, as an image-quality (perceptual-fidelity) metric, where lower values indicate higher perceptual similarity. Second, we measure output diversity using an MS-SSIM-based score: we randomly sample 300 pairs of generated images and report Diversity $:= 1 - \text{MS-SSIM}$, averaged over the sampled pairs (higher indicates greater diversity). Together, these metrics capture brand-level effects on both perceptual quality and sample diversity.

## C.2. Additional Details on Diffusion Model Training and Inference

**CIFAR-20.** For CIFAR-20, we implement a DDPM-CFG (Ho & Salimans, 2022) with a U-Net architecture containing 10.4M parameters. The maximum diffusion timestep is set to $T = 1000$ during training, with a linear variance schedule for the forward diffusion process ranging from $\beta_1 = 10^{-4}$ to $\beta_T = 0.02$. We use the AdamW optimizer (Loshchilov & Hutter, 2017) with a learning rate of $3 \times 10^{-4}$ and weight decay of $10^{-4}$. The model is trained for 30,000 steps with a batch size of 256, using a cosine learning rate schedule with 3,000 warm-up steps. Random horizontal flipping is applied for data augmentation. During inference, images are generated using the 50-step DDIM solver (Song et al., 2020) with class label as conditioning.

**ArtBench (Post-Impressionism).** A Stable Diffusion model[15] (Rombach et al., 2022) is fully fine-tuned. The prompt is set to ``a Post-Impressionist painting by {artist}'' for each image. The model is trained using the AdamW optimizer (Loshchilov & Hutter, 2017) with a weight decay of $10^{-6}$, for 200 epochs and a batch size of 64. Cosine

---

[11] https://www.cs.toronto.edu/~kriz/cifar.html
[12] https://github.com/sbarratt/inception-score-pytorch
[13] https://github.com/mseitzer/pytorch-fid
[14] https://github.com/LAION-AI/aesthetic-predictor
[15] https://huggingface.co/lambdalabs/miniSD-diffusers

learning rate annealing is used, with 500 warm-up steps and an initial learning rate of $3 \times 10^{-4}$. At inference time, images are generated using the PNDM scheduler with 100 steps (Karras et al., 2022).

**Fashion-Product.** We fine-tune FLUX.1-DEV[16], which is a 12 billion parameter rectified flow transformer (Labs et al., 2025) capable of generating images from text descriptions, using LoRA with rank $r = 128$, corresponding to 358M trainable LoRA parameters. Prompts are constructed from the product display name using the template ``{brand} {category}'' (e.g., *Highlander Men Black Checked Shirt*). LoRA parameters are optimized with 8-bit AdamW (Loshchilov & Hutter, 2017) with weight decay $10^{-4}$ for 60 epochs and an effective batch size of 32 (batch size 16 per device with 2 gradient-accumulation steps). We adopt a constant learning-rate schedule with 500 warm-up steps and an initial learning rate of $3 \times 10^{-4}$. At inference time, we generate 256×256 images using the Euler Discrete Scheduler (Esser et al., 2024) with 50 denoising steps.

**Training details** For training diffusion models, we utilize NVIDIA GPUs: A40 for CIFAR-20 and ArtBench and H200 for Fashion-product, equipped with 48GB and 192GB of memory, respectively. This study employs the Diffusers package version 0.35.0 [17] to train models across. Per-sample gradients are computed using the techniques outlined in the PyTorch package tutorial (version 2.8.0) [18]. Furthermore, we apply the TRAK package[19] to project gradients with a random projection matrix. All experiments are conducted on systems equipped with 64 CPU cores and the specified NVIDIA GPUs.

To fine-tune FLUX.1-DEV efficiently, we apply two memory-saving strategies. First, because the VQ-VAE remains frozen, we pre-compute and cache its latent embeddings for all training samples. Likewise, we pre-compute and store text embeddings from both T5 and the CLIP text encoder. This eliminates the need to load the VQ-VAE and text-encoder modules into GPU memory during training. Second, we reduce optimizer and activation memory by using the 8-bit Adam optimizer from the `bitsandbytes`[20] package (Dettmers et al., 2022) together with fp16 mixed-precision fine-tuning.

### C.3. Training details for Surrogate XGBoost Model

This section details the implementation and optimization of the surrogate model introduced in Section 4.2. We employ a gradient-boosted decision tree (GBDT) regressor using the `XGBoost` framework.[21] To train the model, each sampled coalition is represented as a binary mask $z_m \in \{0, 1\}^n$, paired with its respective target value. The surrogate $\hat{f}_\omega$ is optimized by minimizing the Mean Squared Error (MSE) using the squared-loss objective. To ensure robust generalization and prevent overfitting, we perform a grid search over the hyperparameter space defined in Table S.1, selecting the optimal configuration via 5-fold cross-validation.

*Table S.1.* XGBoost hyperparameter search space for training the surrogate model $f_\omega$.

| Hyperparameter | Search Space / Values |
|---|---|
| Max. Tree Depth | $\{3, 5, 10\}$ |
| Number of Trees | $\{100, 300, 900\}$ |
| Learning Rate | $\{0.01, 0.05\}$ |
| L1 Regularization $\lambda$ | $\{0, 10\}$ |

---

[16]https://huggingface.co/black-forest-labs/FLUX.1-dev
[17]https://pypi.org/project/diffusers/
[18]https://pytorch.org/
[19]https://trak.csail.mit.edu/quickstart
[20]https://github.com/bitsandbytes-foundation/bitsandbytes
[21]https://xgboost.readthedocs.io/en/latest/index.html

# D. Synthetic Validation and Axiomatic Checks

In this section, we examine whether our proposed pipeline for Shapley computation in Section 4.2 satisfies the fundamental axioms of Shapley values and evaluate its sample efficiency across three distinct utility functions in synthetic environments.

## D.1. Experimental Environment Setup

We first define three synthetic coalition value functions $v : \{0, 1\}^N \to \mathbb{R}$ spanning increasing levels of nonlinearity and interaction structure. We use $N \in \{10, 11\}$ players so that the ground-truth Shapley values $\phi(v)$ can be computed exactly by enumerating all coalitions. Unless otherwise stated, we sample feature weights $\mathbf{w} \sim \mathcal{N}(0, 1)$ and fix the base utility to $b = 0.25$. For any coalition indicator $\mathbf{z} \in \{0, 1\}^N$, $z_i = 1$ denotes inclusion of player $i$.

**Linear (additive).** This setting is purely additive and serves as a baseline where contributions are independent.

$$v(\mathbf{z}) = b + \mathbf{w}^\top \mathbf{z}. \tag{32}$$

**Nonlinear (additive with saturation).** To introduce diminishing returns without explicit feature interactions, we apply a global nonlinearity to the additive score:

$$v(\mathbf{z}) = 3 \tanh\left(\frac{b + \mathbf{w}^\top \mathbf{z}}{3}\right). \tag{33}$$

This tests whether the estimator remains accurate under output saturation near the extremes of the coalition hypercube.

**Interaction-heavy.** To emphasize interaction effects, we add pairwise and third-order terms before applying the same saturating nonlinearity:

$$v(\mathbf{z}) = 3 \tanh\left(\frac{b + \mathbf{w}^\top \mathbf{z} + \sum_{(i,j,\alpha)\in\mathcal{P}} \alpha\, z_i z_j + \alpha_{\text{tri}}\, z_0 z_2 z_4}{3}\right), \tag{34}$$

where $\mathcal{P} = \{(0, 1, 1.5), (2, 3, -1.0), (4, 5, 0.8)\}$ and $\alpha_{\text{tri}} = 2.0$. This regime requires resolving synergistic effects that cannot be captured by linear surrogates alone.

## D.2. Comparative Analysis of Sample Efficiency

**Evaluation setup & baselines** To evaluate the convergence properties of our proposed estimator, we compare SurrogateSHAP against baselines on synthetic coalition utilities where the proxy gap is set to $\mathcal{E}_{\text{gap}} = 0$ (Equation (12)). This controlled environment allows us to isolate the surrogate error $\mathcal{E}_{\text{sur}}$ and measure the accuracy of the estimated Shapley values using the relative $L_2$ error, defined as $\|\hat{\phi} - \phi(v)\|_2 / \|\phi(v)\|_2$..

For a fixed budget of $M$ function evaluations, we enforce the inclusion of the boundary cases, the empty coalition $\mathbf{0}$ and the grand coalition $\mathbf{1}$, to ensure foundational consistency with the efficiency and dummy axioms. The remaining $M - 2$ coalitions are drawn from the interior of the hypercube, $\{0, 1\}^n \setminus \{\mathbf{0}, \mathbf{1}\}$, according to the specific sampling distribution of each method (e.g., the Shapley kernel or a uniform distribution) sampled with replacement. We report the mean relative $L_2$ error aggregated over 60 independent trials.

Our baselines include KernelSHAP, a weighted linear regression estimator (Lundberg & Lee, 2017), and ProxySPEX with varying interaction orders ($k \in \{1, 2, 3\}$) (Butler et al., 2025). Both SurrogateSHAP and ProxySPEX use the same ensemble XGBoost.

**Results** The results across three synthetic environments illustrate the trade-off between estimator inductive bias and model capacity. In the purely linear regime, KernelSHAP leverages its inherent linear bias to achieve $L_2$ error $\approx 10^{-14}$, significantly outperforming surrogate-based methods (Figure S.1-a). However, as utility complexity increases, KernelSHAP's performance stagnates; in contrast, in non-additive settings, tree-based surrogates—including SurrogateSHAP and ProxySPEX ($k = 3$)—converge more slowly but eventually surpass KernelSHAP, reaching an error of $\approx 10^{-2}$ at $M = 512$, (Figure S.1-b).

This advantage is most pronounced in the interaction-heavy regime (Figure S.1-c), where KernelSHAP's linear assumption causes the error to plateau near $10^{-1}$. While ProxySPEX (Butler et al., 2025) improves upon the linear model, it remains sensitive to the specified interaction order $k$: low-order configurations ($k = 1, 2$) underfit, while higher orders eventually

saturate. Conversely, by avoiding explicit sparsity constraints, SurrogateSHAP exceeds ProxySPEX performance at $M = 500$ and achieves errors several orders of magnitude lower than the linear estimator. Finally, we investigate the bias-variance decomposition of these estimators; our results show that in interaction-heavy settings, SurrogateSHAP achieves both lower bias and lower variance than competing methods (Figure S.2).

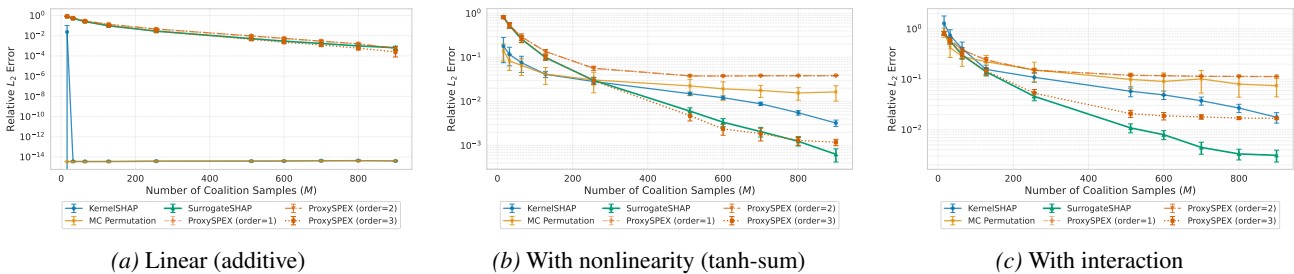

*(a)* Linear (additive)     *(b)* With nonlinearity (tanh-sum)     *(c)* With interaction

*Figure S.1.* Sample-efficiency across utility landscapes. The x-axis shows the number of sampled coalitions $M$, and the y-axis shows the $\ell_2$ relative to the oracle. (a) Simple linear additive utilities; (b) nonlinear utilities; (c) interaction-dominated utilities.

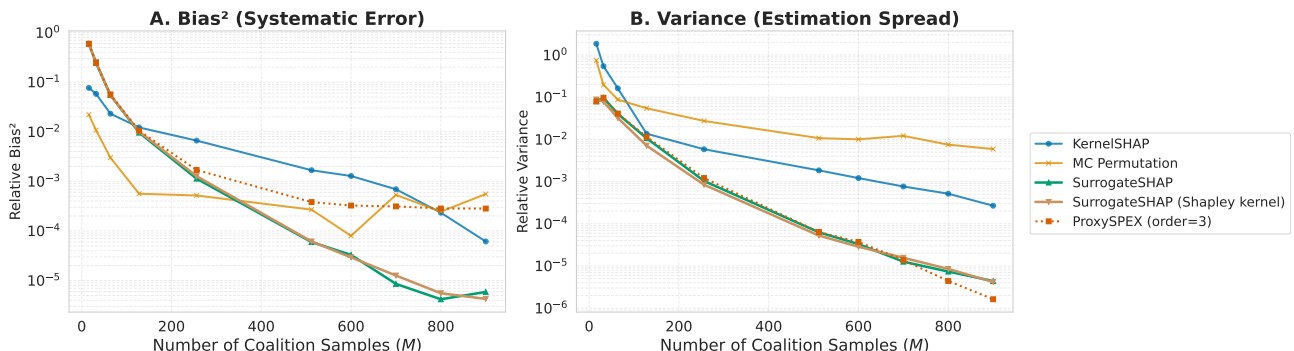

*Figure S.2.* Error decomposition (bias (left) vs. variance (right)) for the interaction setting.

### D.3. Axiomatic Sanity Checks

**Evaluation setup**   In addition to evaluating convergence, we validate that SurrogateSHAP adheres to the core game-theoretic axioms across all three synthetic environments, specifically focusing on the null player and efficiency properties. To assess the null player axiom, we injected a player with a zero marginal contribution into the utility function.

**Results**   Our results demonstrate that the attribution magnitude $|\phi_{\text{null}}|$ consistently converges toward zero in all regimes as the number of sampled coalitions $M$ increases, reaching its minimum at the exhaustive limit of $M = 2^{10}$ as illustrated in panels B of Figure S.3. This convergence indicates that the surrogate model correctly identifies and discounts features that do not contribute to the overall model behavior.

The efficiency axiom is similarly satisfied, as the absolute efficiency gap $|\sum \hat{\phi} - \Delta v|$ remains negligible across all utility complexities. Throughout our experiments, this gap consistently stays below a threshold of $0.006$, as shown in panels C of Figure S.3. Because the sum of the attributions closely matches the total change in utility, we confirm that the exact TreeSHAP decomposition applied to our learned surrogate maintains high theoretical fidelity. This stability ensures that the resulting Shapley values are a mathematically sound representation of the global utility gain, regardless of the underlying interaction landscape.

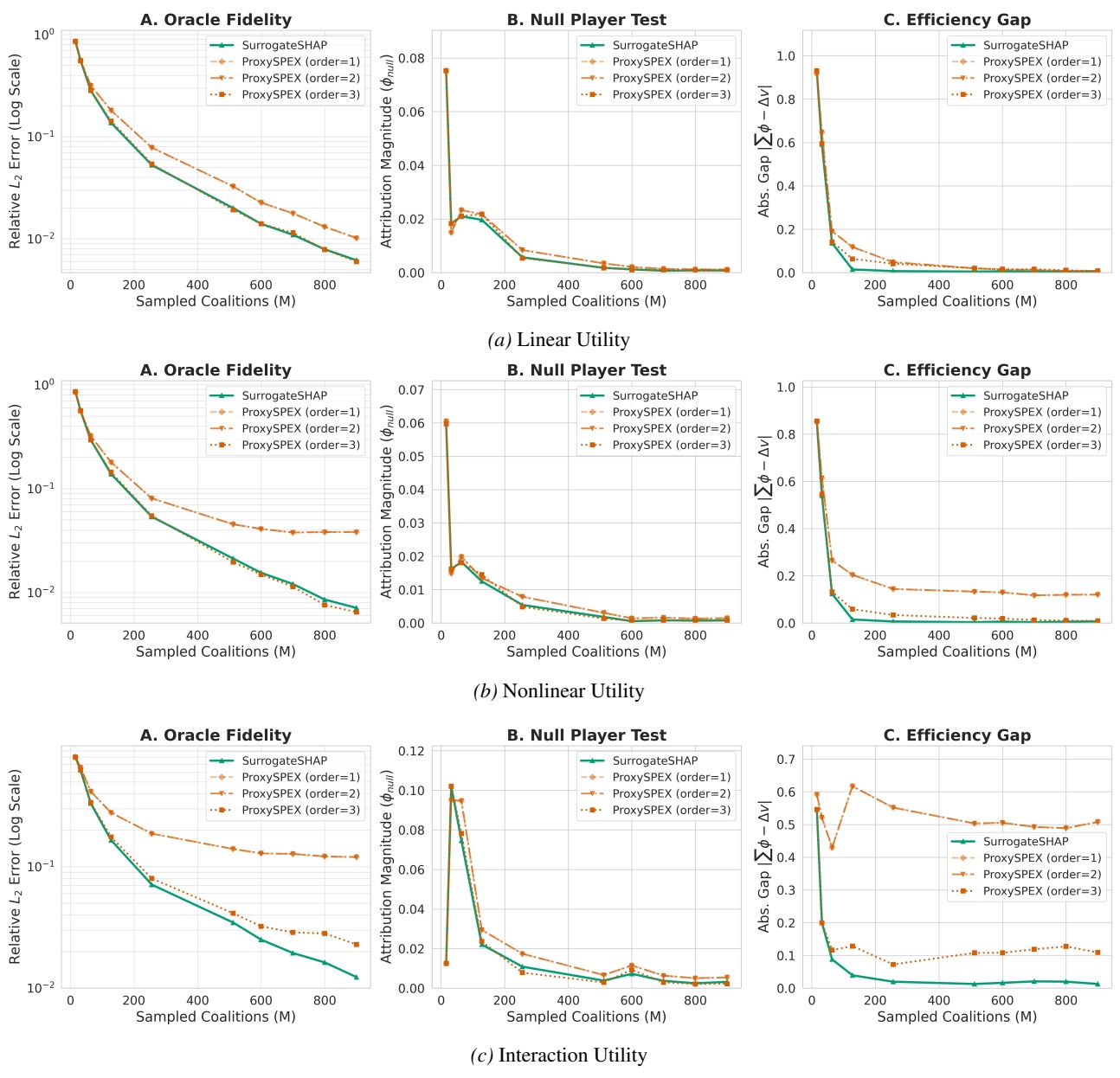

*(a)* Linear Utility

*(b)* Nonlinear Utility

*(c)* Interaction Utility

*Figure S.3.* Axiomatic Validation across three environments. In all cases, the relative $L_2$ error (A) and Null Player attribution (B) diminish with increased sampling, while the Efficiency Gap (C) remains consistently near zero.

# E. Additional Results

## E.1. Approximation Gap Analysis

In this section, we provide an extensive validation of the relationship between global generative quality and local similarity metrics. We compare our proposed proxy game (Section 4.1) against full retraining instances across subsets sampled from the Shapley distribution to quantify the fidelity of our approximation. Specifically, we sample $N = 100$ subsets and generate 100 samples for both our proxy games and the retraining instances to compute local distances and global metrics.

### E.1.1. CIFAR-20

**Fréchet Inception Distance (FID)**   As shown in Figure S.4-left, Proxy FID exhibits high alignment with retrained FID ($R^2 = 0.935$, Spearman $\rho = 0.984$, $N = 100$), confirming that the proxy effectively preserves relative performance rankings. Correlation analysis reveals that $\Delta_{\text{FID}}$ is strongly associated with local reconstruction errors, particularly pixel-wise similarity (MAE: $r = 0.745$, SSIM: $r = -0.782$) and representation-level drift (RMSE-CLIP: $r = 0.689$, RMSE-LPIPS: $r = 0.667$). These results empirically validate the stability assumption ($\varepsilon, \varphi_u$), specifically, that the utility error is bounded by the representation-space drift as predicted in Theorem 4.2.

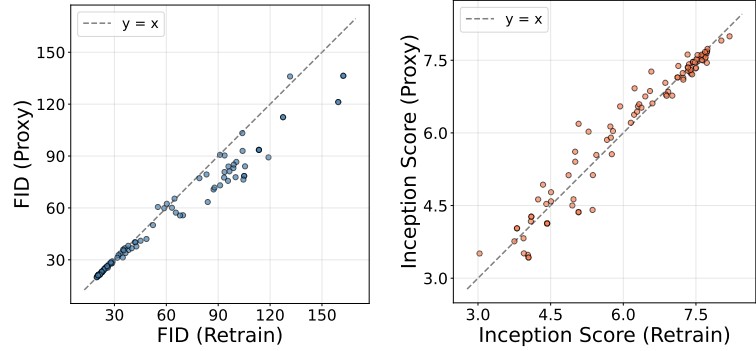

*Figure S.4.* Fidelity of the proxy utility: Alignment between the proxy game ($y$-axis) and ground-truth retraining ($x$-axis).

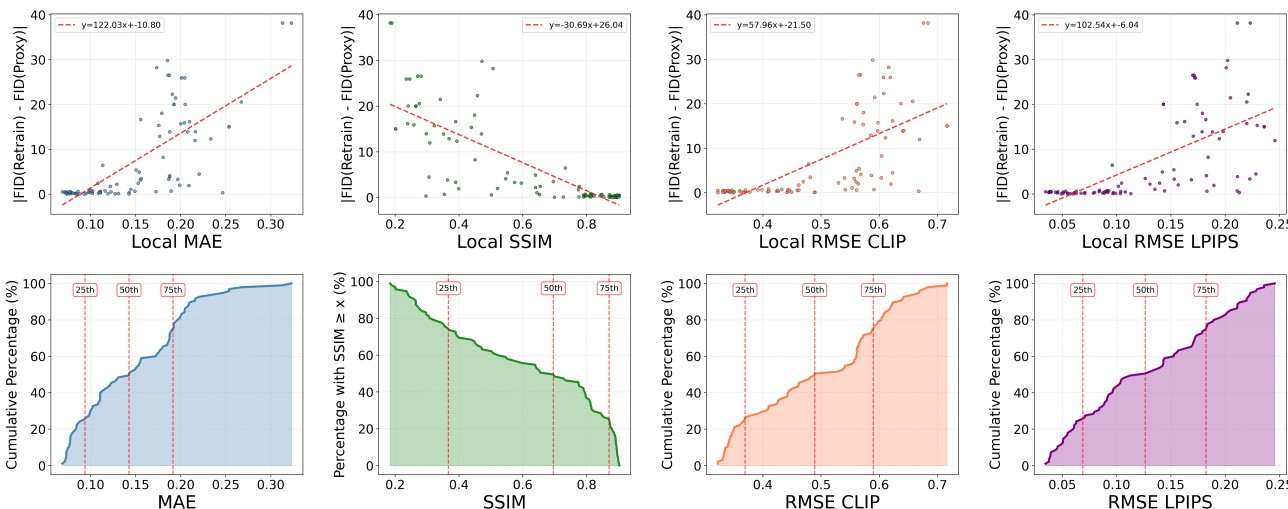

*Figure S.5.* **Empirical Validation of Proxy FID. (Top)** The approximation gap $\delta_{\text{FID}}$ scales linearly with local reconstruction metrics (MAE, SSIM) and representation-space drift (RMSE-CLIP, RMSE-LPIPS). **(Bottom)** Cumulative Distribution Functions (CDFs) of local fidelity metrics across $N = 100$ subsets. Dashed red lines denote the $25^{\text{th}}$, $50^{\text{th}}$, and $75^{\text{th}}$ percentiles.

**Empirical Validation of Proxy IS.**   We further validate the Inception Score (IS) of proxy coalitions across the same subset distribution. As shown in Figure S.4-right, the Proxy IS demonstrates a high degree of fidelity to the retraining ground truth ($R^2 = 0.920$, Spearman $\rho = 0.942$, $N = 100$). Consistent with the FID findings, the IS approximation gap,

$\Delta_{\text{IS}} = |\text{IS}_{\text{retrain}} - \text{IS}_{\text{proxy}}|$, is strongly conditioned on local reconstruction quality. We observe a significant relationship between $\delta_{\text{IS}}$ and our fidelity metrics: MAE ($r = 0.791$), SSIM ($r = -0.745$), RMSE-CLIP ($r = 0.580$), and RMSE-LPIPS ($r = 0.575$). These results reinforce the conclusion that the proxy effectively preserves the utility landscape, particularly when representation drift $\varepsilon^{\varphi_u}$ is small.

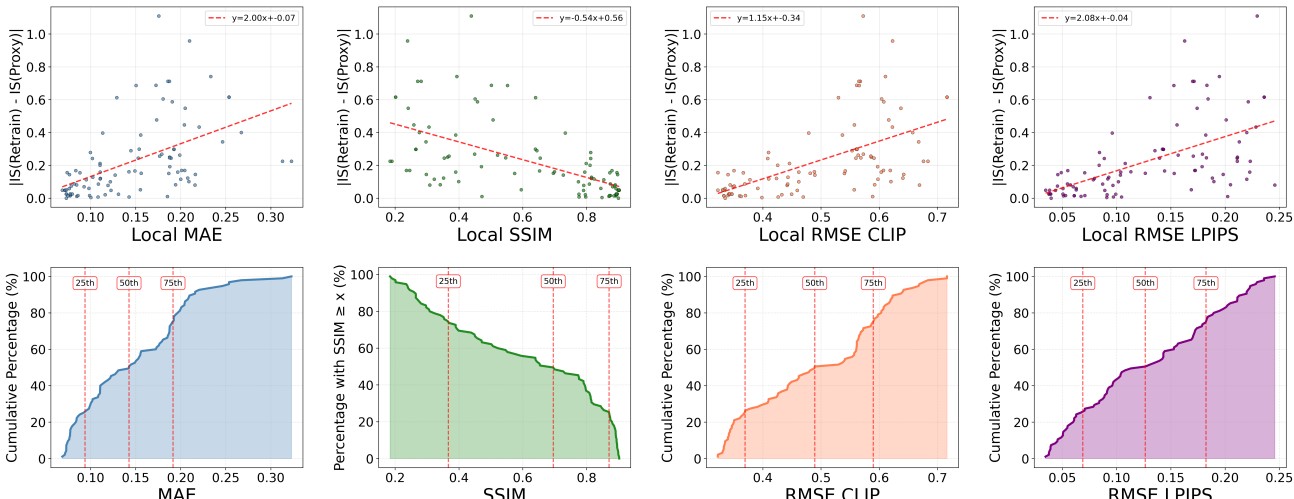

*Figure S.6.* **Empirical Validation of Proxy Inception Score. (Top)** The approximation gap $\delta_{\text{IS}}$ scales linearly with local reconstruction metrics (MAE, SSIM) and representation-space drift (RMSE-CLIP, RMSE-LPIPS). **(Bottom)** Cumulative Distribution Functions (CDFs) of local fidelity metrics across $N = 100$ subsets. Dashed red lines denote the $25^{\text{th}}$, $50^{\text{th}}$, and $75^{\text{th}}$ percentiles.

**Distributional Thresholds** To provide practical reference points for proxy reliability, Table S.4 summarizes the empirical distributions of local metrics. Our results suggest that the proxy remains a robust estimator of generative performance provided local structural similarity is maintained (median SSIM $= 0.730$). For instance, MAE exhibits a median of $0.142$ (90th percentile: $0.214$), while SSIM reaches $0.894$ at the 90th percentile, establishing empirical bounds where the approximation gap is minimized.

*Table S.2.* Empirical distributions of local reconstruction metrics across 100 samples of 100 evaluation subsets from the Shapley distribution for CIFAR-20

| Metric | Mean | Median | 25th % | 75th % | 90th % | 95th % |
|---|---|---|---|---|---|---|
| MAE | 0.105 | 0.132 | 0.085 | 0.188 | 0.214 | 0.235 |
| SSIM | 0.779 | 0.785 | 0.390 | 0.874 | 0.894 | 0.900 |
| RMSE (CLIP dist.) | 0.452 | 0.442 | 0.369 | 0.589 | 0.631 | 0.667 |
| RMSE (LPIPS) | 0.108 | 0.109 | 0.068 | 0.182 | 0.217 | 0.224 |

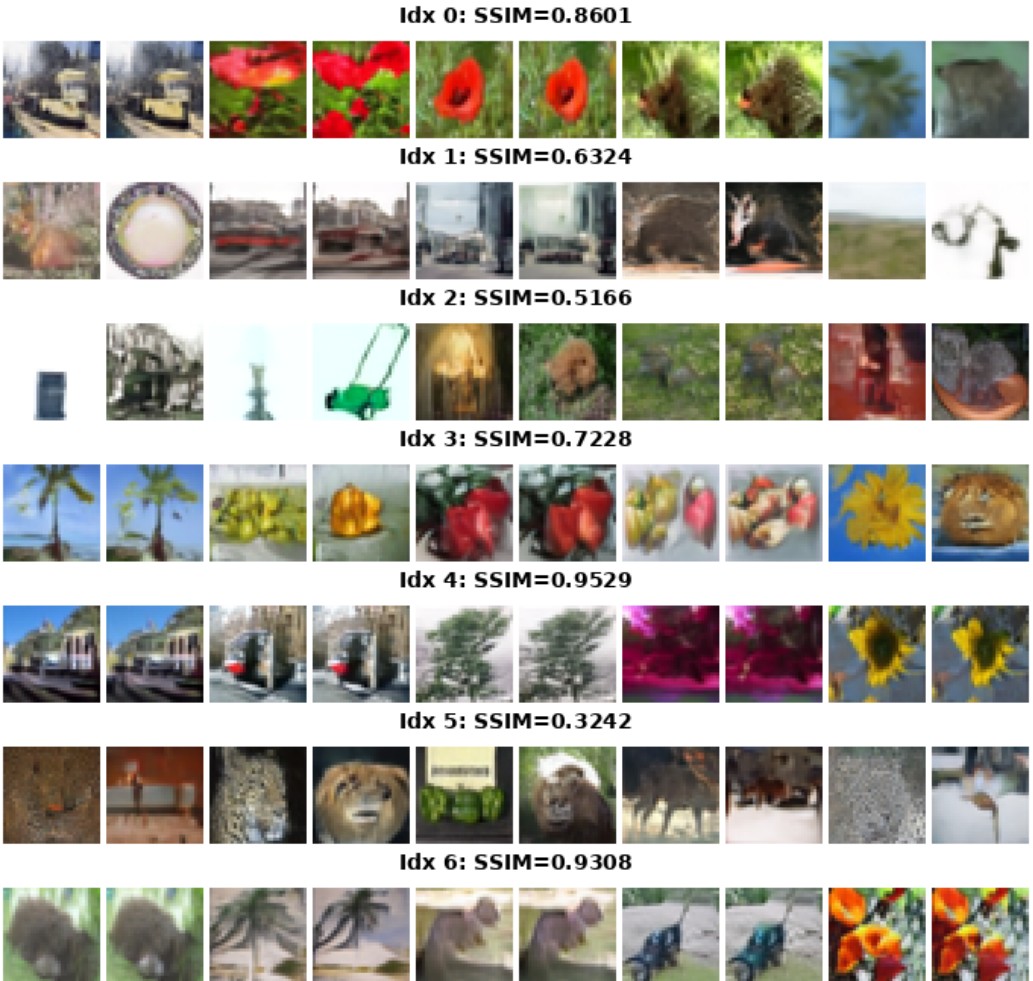

*Figure S.7.* **Qualitative Comparison of Retrained vs. Target Model Outputs (CIFAR-20)** Each row (Idx 0–6) displays five pairs of images generated using a shared noise coupling $z$, with the retrained model $\theta_S^*$ on the left and the target model $\theta$ output on the right of each pair. The labels indicate the Structural Similarity Index (SSIM) for each coalition $S$.

### E.1.2. ARTBENCH POST-IMPRESSION VALIDATION

**Average Aesthetic Score.** As illustrated in Figure S.8, the proxy aesthetic scores exhibit moderate alignment with the retraining ground truth ($R^2 = 0.330$, Spearman $\rho = 0.589$, $N = 100$). We define the approximation gap as $\Delta_{\text{Aesthetic Score}} = |\text{Aesthetic Score}_{\text{retrain}} - \text{Aesthetic Score}_{\text{proxy}}|$. This gap demonstrates a stronger correlation with representation-level drift (RMSE-CLIP: $r = 0.445$; RMSE-LPIPS: $r = 0.578$) than with pixel-level metrics (MAE: $r = 0.224$; SSIM: $r = -0.315$). This suggests that the proxy's fidelity for aesthetic utility is primarily sensitive to semantic shifts in the feature space rather than low-level pixel residuals, consistent with the perceptual nature of aesthetic scoring models.

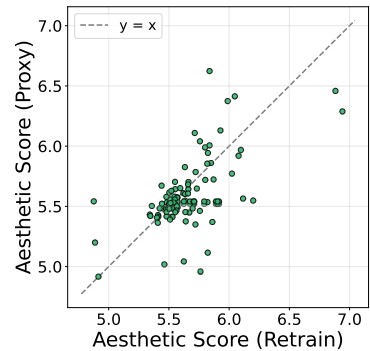

*Figure S.8.* Fidelity of the proxy utility: Alignment between the proxy game ($y$-axis) and ground-truth retraining ($x$-axis).

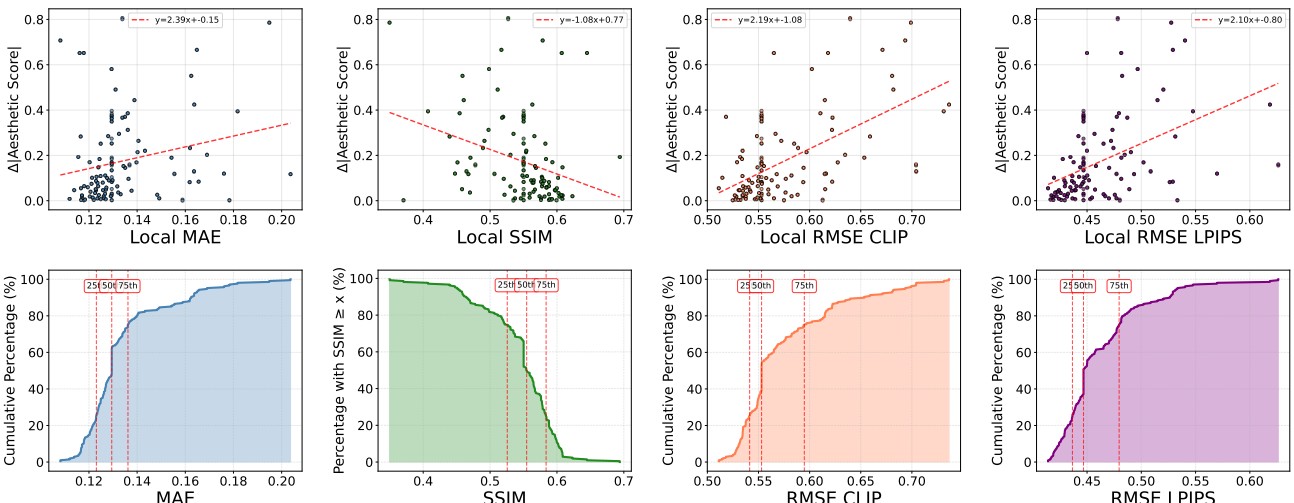

*Figure S.9.* **Empirical Validation of Proxy Average Aesthetic Score. (Top)** The approximation gap $\delta_{\text{avg. aesthetic score}}$ scales linearly with local reconstruction metrics (MAE, SSIM) and representation-space drift (RMSE-CLIP, RMSE-LPIPS). **(Bottom)** Cumulative Distribution Functions (CDFs) of local fidelity metrics across $N = 100$ subsets. Dashed red lines denote the $25^{\text{th}}$, $50^{\text{th}}$, and $75^{\text{th}}$ percentiles.

*Table S.3.* Empirical distributions of local reconstruction metrics across 100 samples of 100 evaluation subsets from Shapley distribution for ArtBench (post-impressionism)

| Metric | Mean | Median | 25th % | 75th % | 90th % | 95th % |
| --- | --- | --- | --- | --- | --- | --- |
| MAE | 0.147 | 0.129 | 0.128 | 0.170 | 0.201 | 0.211 |
| SSIM | 0.531 | 0.553 | 0.447 | 0.603 | 0.634 | 0.642 |
| RMSE (CLIP dist.) | 0.529 | 0.510 | 0.494 | 0.594 | 0.653 | 0.693 |
| RMSE (LPIPS) | 0.458 | 0.446 | 0.436 | 0.479 | 0.526 | 0.533 |

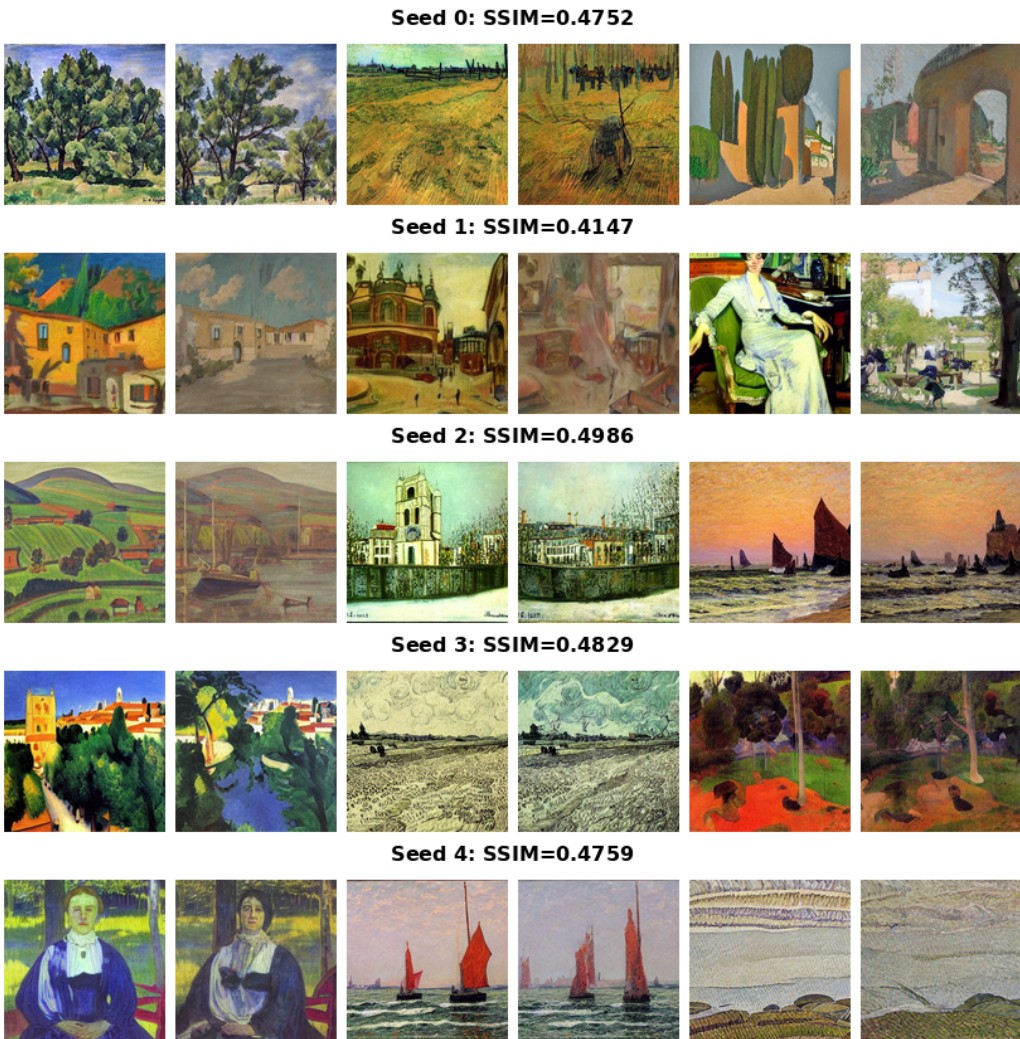

*Figure S.10.* **Qualitative Comparison of Retrained vs. Target Model Outputs (ArtBench (Post-Impressionism))** Each row (Idx 0–4) displays three pairs of images generated using a shared noise coupling $z$, with the retrained model $\theta_S^*$ on the left and the target model $\theta$ output on the right of each pair. The labels indicate the Structural Similarity Index (SSIM) for each coalition $S$.

### E.1.3. FASHION-PRODUCT VALIDATION

**Average LPIPS.** We evaluate the proxy's fidelity in approximating the mean LPIPS utility on the Fashion-Product dataset. In this setup, the LPIPS utility measures the perceptual distance between generated samples and a reference training point, whereas RMSE-LPIPS quantifies the drift between samples generated by the proxy and the retrained ground truth. As shown in Figure S.11-left, the proxy utility exhibits moderate linear alignment with the retraining ground truth ($R^2 = 0.150$, Spearman $\rho = 0.443$, $N = 100$). However, the utility approximation gap, $\Delta_{\text{LPIPS}} = |\text{LPIPS}_{\text{retrain}} - \text{LPIPS}_{\text{proxy}}|$, is highly structured. Specifically, $\Delta_{\text{LPIPS}}$ is exceptionally well-correlated with cross-model representation drift (RMSE-CLIP: $r = 0.817$; RMSE-LPIPS: $r = 0.900$). This strong positive correlation indicates that the utility error vanishes predictably as the generated outputs of the proxy and retrained models converge in representation space. These results validate that the proxy's limitations are strictly a function of reconstruction quality, empirically supporting the stability bound in Proposition 4.2.

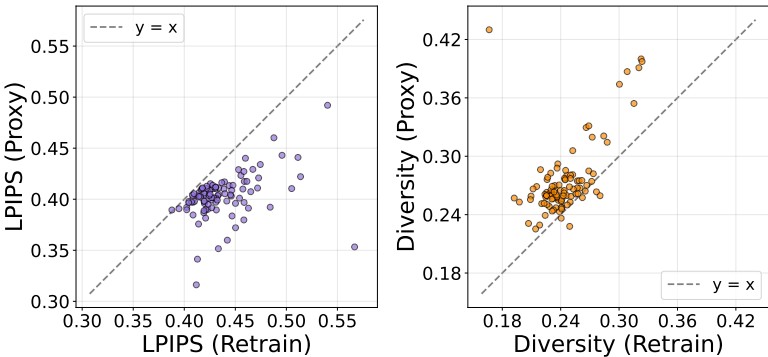

*Figure S.11.* Fidelity of the proxy utility: Alignment between the proxy game ($y$-axis) and ground-truth retraining ($x$-axis).

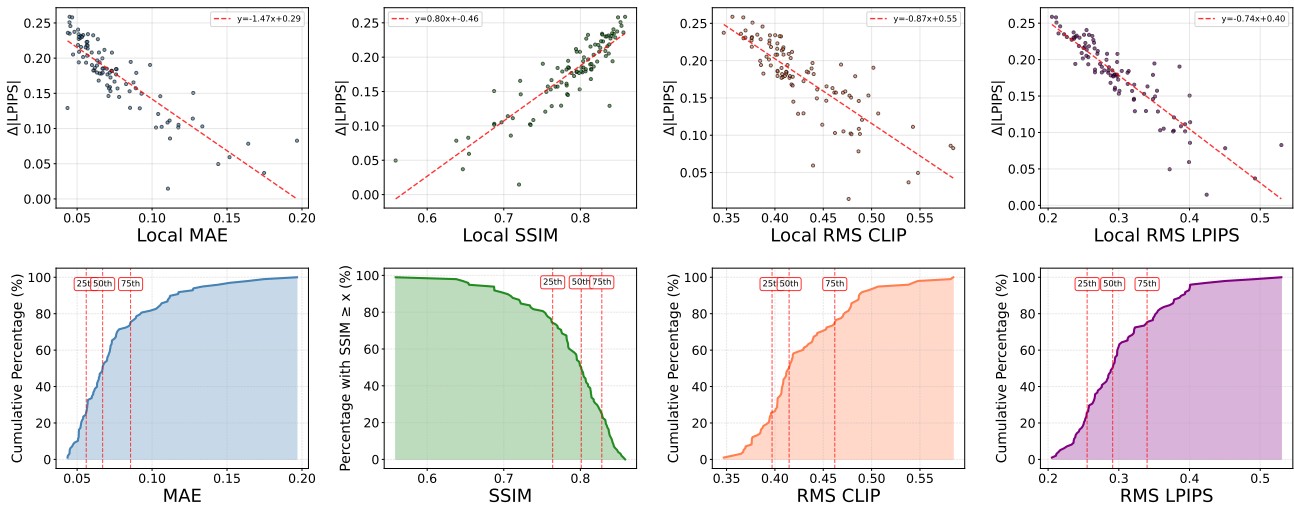

*Figure S.12.* **Empirical Validation of Proxy Average LPIPS Score. (Top)** The approximation gap $\delta_{\text{LPIPS}}$ scales linearly with local reconstruction metrics (MAE, SSIM) and representation-space drift (RMSE-CLIP, RMSE-LPIPS). **(Bottom)** Cumulative Distribution Functions (CDFs) of local fidelity metrics across $N = 100$ subsets. Dashed red lines denote the $25^{\text{th}}$, $50^{\text{th}}$, and $75^{\text{th}}$ percentiles.

**Diversity.** We evaluate the proxy's fidelity in approximating the diversity score on the Fashion-Product dataset. As illustrated in Figure S.11-right, the proxy diversity scores exhibit moderate alignment with the retraining ground truth ($R^2 = 0.313$, Spearman $\rho = 0.495$, $N = 100$), capturing the general distributional trends despite some dispersion. We define the approximation gap as $\Delta_{\text{Diversity}} = |\text{Diversity}_{\text{retrain}} - \text{Diversity}_{\text{proxy}}|$. This gap demonstrates a stronger correlation with representation-level drift (RMSE-CLIP: $r = 0.585$; RMSE-LPIPS: $r = 0.679$) and pixel-level metrics (MAE: $r = 0.694$; SSIM: $r = -0.581$).

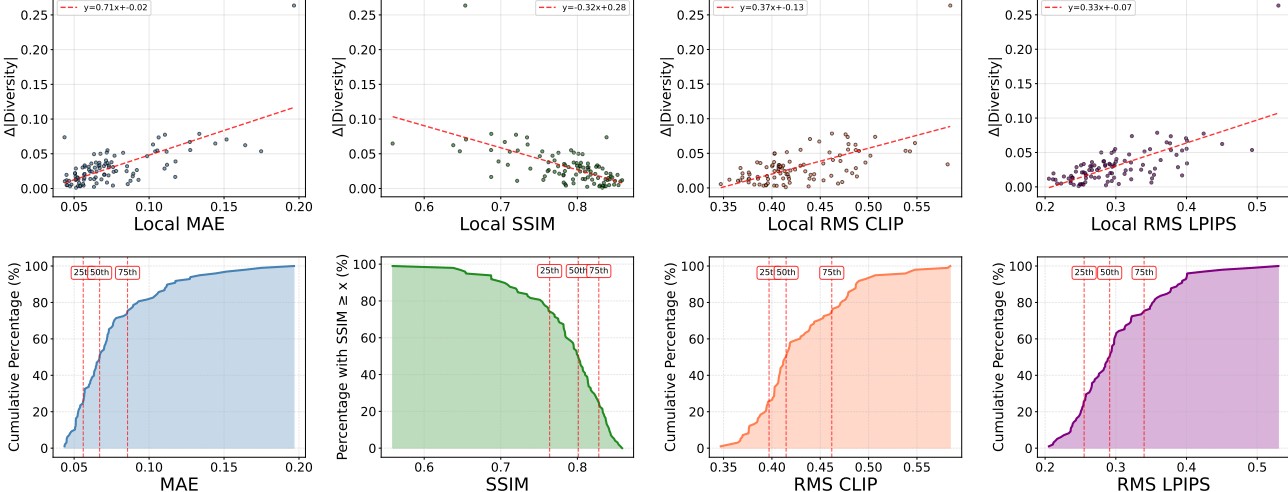

*Figure S.13.* **Empirical Validation of Proxy Diversity Score.** **(Top)** The approximation gap $\delta_{\text{Diversity}}$ scales linearly with local reconstruction metrics (MAE, SSIM) and representation-space drift (RMSE-CLIP, RMSE-LPIPS). **(Bottom)** Cumulative Distribution Functions (CDFs) of local fidelity metrics across $N = 100$ subsets. Dashed red lines denote the $25^{\text{th}}$, $50^{\text{th}}$, and $75^{\text{th}}$ percentiles.

*Table S.4.* Empirical distributions of local reconstruction metrics across 100 samples of 100 evaluation subsets from Shapley distribution for Fashion-Product

| Metric | Mean | Median | 25th % | 75th % | 90th % | 95th % |
|---|---|---|---|---|---|---|
| MAE | 0.177 | 0.166 | 0.147 | 0.195 | 0.235 | 0.263 |
| SSIM | 0.786 | 0.801 | 0.763 | 0.827 | 0.841 | 0.848 |
| RMS CLIP | 0.429 | 0.414 | 0.397 | 0.461 | 0.487 | 0.511 |
| RMS LPIPS | 0.301 | 0.291 | 0.255 | 0.340 | 0.389 | 0.400 |

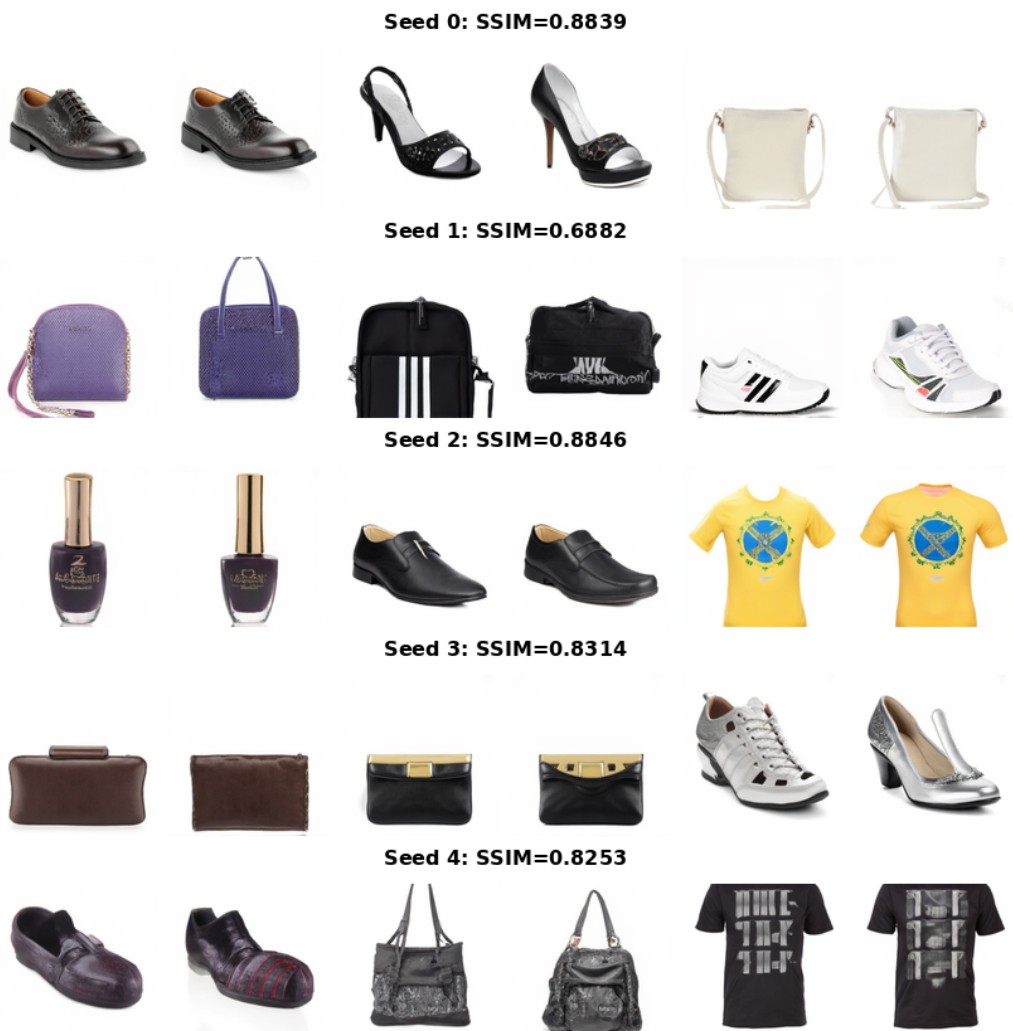

*Figure S.14.* **Qualitative Comparison of Retrained vs. Target Model Outputs (Fashion-Product).** Each row (Idx 0–4) displays three pairs of images generated using a shared noise coupling $z$, with the retrained model $\theta_S^*$ on the left and the target model $\theta$ output on the right of each pair. The labels indicate the Structural Similarity Index (SSIM) for each coalition $S$.

## E.2. Additional Results for Linear Datamodel Score (LDS)

We provide supplemental results for the Linear Datamodel Score (LDS) experiments using varying ratio $\alpha \in \{0.25, 0.75\}$. These evaluations follow the experimental protocol detailed in Section 5.4 and demonstrate the consistency of our findings across different datamodel $\alpha$.

*Table S.5.* LDS (%) results with $\alpha = 0.25$. Means and 95% confidence intervals across five random initializations are reported.

| Method | CIFAR-20 | | ArtBench (Post-Impressionism) | Fashion Product | |
|---|---|---|---|---|---|
| | Inception score | FID | Aesthetic score | Diversity | LPIPS |
| Pixel similarity (avg.) | $-4.83 \pm 2.02$ | $0.77 \pm 1.24$ | $-1.03 \pm 2.30$ | $16.64 \pm 16.12$ | $-6.48 \pm 12.43$ |
| Pixel similarity (max) | $-20.25 \pm 3.48$ | $-1.34 \pm 1.46$ | $3.52 \pm 3.12$ | $-16.71 \pm 10.18$ | $1.67 \pm 7.65$ |
| CLIP similarity (avg.) | $18.68 \pm 1.36$ | $-5.22 \pm 3.90$ | $0.37 \pm 0.97$ | $9.81 \pm 14.31$ | $-11.46 \pm 18.77$ |
| CLIP similarity (max) | $-8.12 \pm 1.44$ | $1.38 \pm 0.56$ | $-2.46 \pm 1.21$ | $0.35 \pm 17.99$ | $-8.42 \pm 8.21$ |
| Gradient similarity (avg.) | $18.69 \pm 0.85$ | $-38.49 \pm 0.18$ | $6.92 \pm 5.00$ | $18.64 \pm 5.66$ | $-7.12 \pm 11.39$ |
| Gradient similarity (max) | $7.03 \pm 3.46$ | $-6.18 \pm 1.05$ | $-3.00 \pm 3.40$ | $9.04 \pm 9.54$ | $-3.39 \pm 4.19$ |
| Gmvaluator (Yang et al., 2025) | $30.42 \pm 0.27$ | $-15.31 \pm 0.98$ | $-5.06 \pm 2.67$ | $6.37 \pm 13.66$ | $-13.58 \pm 11.60$ |
| Aesthetic score (avg.) | – | – | $14.76 \pm 1.98$ | – | – |
| Aesthetic score (max) | – | – | $8.84 \pm 1.24$ | – | – |
| Relative IF (Barshan et al., 2020) | $22.79 \pm 1.99$ | $0.43 \pm 1.36$ | $9.57 \pm 3.12$ | $8.51 \pm 4.52$ | $-2.20 \pm 4.13$ |
| Renormalized IF (Hammoudeh & Lowd, 2022) | $22.84 \pm 1.97$ | $-0.35 \pm 1.35$ | $4.95 \pm 3.78$ | $5.85 \pm 4.66$ | $-2.33 \pm 4.08$ |
| TRAK (Park et al., 2023) | $23.53 \pm 3.98$ | $-0.79 \pm 0.35$ | $8.31 \pm 3.21$ | $8.31 \pm 4.15$ | $-3.12 \pm 4.42$ |
| Journey-TRAK (Georgiev et al., 2023) | $8.82 \pm 0.20$ | $-15.87 \pm 2.63$ | $-19.86 \pm 1.73$ | $-1.24 \pm 9.18$ | $2.51 \pm 11.20$ |
| D-TRAK (Zheng et al., 2023) | $3.39 \pm 2.48$ | $-15.73 \pm 2.27$ | $-20.75 \pm 2.52$ | $12.31 \pm 8.44$ | $-3.30 \pm 16.89$ |
| DAS (Lin et al., 2024) | $24.40 \pm 1.96$ | $-0.45 \pm 2.34$ | $8.45 \pm 3.28$ | $8.06 \pm 4.23$ | $-3.09 \pm 4.22$ |
| Leave-one-out (LOO) | $68.48 \pm 0.35$ | $-13.40 \pm 1.52$ | $9.76 \pm 1.93$ | $-14.41 \pm 6.19$ | $-2.42 \pm 4.61$ |
| Sparsified-FT Shapley (Lu et al., 2025) | $57.73 \pm 2.99$ | $-2.03 \pm 0.73$ | $15.27 \pm 3.11$ | $4.31 \pm 4.15$ | $18.57 \pm 6.21$ |
| SurrogateSHAP (**Ours**) | $\textbf{78.85} \pm \textbf{1.20}$ | $\textbf{43.11} \pm \textbf{0.14}$ | $\textbf{52.29} \pm \textbf{1.62}$ | $\textbf{27.97} \pm \textbf{9.80}$ | $\textbf{17.35} \pm \textbf{7.78}$ |

Sparsified-FT and SurrogateSHAP use an identical number of coalitions.

*Table S.6.* LDS (%) results with $\alpha = 0.75$. Means and 95% confidence intervals across five random initializations are reported.

| Method | CIFAR-20 | | ArtBench (Post-Impressionism) | Fashion Product | |
|---|---|---|---|---|---|
| | Inception score | FID | Aesthetic score | Diversity | LPIPS |
| Pixel similarity (avg.) | $-5.84 \pm 1.24$ | $-1.82 \pm 1.21$ | $14.63 \pm 9.07$ | $2.88 \pm 15.71$ | $-6.76 \pm 8.34$ |
| Pixel similarity (max) | $-12.11 \pm 0.21$ | $-9.87 \pm 0.55$ | $14.02 \pm 0.91$ | $-15.60 \pm 9.54$ | $17.67 \pm 2.92$ |
| CLIP similarity (avg.) | $-18.77 \pm 0.86$ | $19.98 \pm 0.17$ | $14.57 \pm 6.68$ | $-8.72 \pm 6.90$ | $5.72 \pm 5.35$ |
| CLIP similarity (max) | $16.78 \pm 0.93$ | $-7.39 \pm 0.38$ | $13.89 \pm 5.45$ | $-4.88 \pm 4.47$ | $13.21 \pm 5.28$ |
| Gradient similarity (avg.) | $29.44 \pm 1.85$ | $-17.19 \pm 0.60$ | $0.34 \pm 2.26$ | $3.71 \pm 15.03$ | $-10.19 \pm 12.45$ |
| Gradient similarity (max) | $-0.96 \pm 1.56$ | $-2.44 \pm 0.72$ | $-3.89 \pm 3.56$ | $0.41 \pm 10.93$ | $-1.28 \pm 11.47$ |
| Gmvaluator (Yang et al., 2025) | $-5.89 \pm 0.517$ | $30.07 \pm 0.94$ | $15.12 \pm 3.32$ | $8.99 \pm 12.67$ | $-9.26 \pm 2.71$ |
| Aesthetic score (avg.) | – | – | $3.25 \pm 2.29$ | – | – |
| Aesthetic score (max) | – | – | $-9.47 \pm 6.97$ | – | – |
| Relative IF (Barshan et al., 2020) | $1.09 \pm 1.14$ | $16.62 \pm 1.21$ | $3.80 \pm 5.15$ | $2.07 \pm 10.82$ | $7.42 \pm 4.24$ |
| Renormalized IF (Hammoudeh & Lowd, 2022) | $1.15 \pm 1.10$ | $16.09 \pm 1.51$ | $0.51 \pm 4.89$ | $-1.49 \pm 10.48$ | $9.42 \pm 4.81$ |
| TRAK (Park et al., 2023) | $1.68 \pm 1.13$ | $15.91 \pm 2.52$ | $3.16 \pm 4.91$ | $2.17 \pm 9.46$ | $5.77 \pm 2.77$ |
| Journey-TRAK (Georgiev et al., 2023) | $-1.67 \pm 0.70$ | $-17.65 \pm 2.43$ | $-4.81 \pm 2.63$ | $13.58 \pm 15.80$ | $-10.45 \pm 10.29$ |
| D-TRAK (Zheng et al., 2023) | $8.34 \pm 1.35$ | $-11.83 \pm 1.59$ | $-22.87 \pm 4.56$ | $0.26 \pm 15.68$ | $11.86 \pm 9.45$ |
| DAS (Lin et al., 2024) | $1.97 \pm 1.14$ | $16.49 \pm 1.53$ | $3.52 \pm 5.00$ | $2.15 \pm 9.25$ | $5.76 \pm 2.73$ |
| Leave-one-out (LOO) | $50.88 \pm 0.71$ | $\textbf{34.33} \pm \textbf{1.22}$ | $4.40 \pm 3.31$ | $14.59 \pm 5.91$ | $-5.57 \pm 11.28$ |
| Sparsified-FT Shapley (Lu et al., 2025) | $44.79 \pm 0.74$ | $5.38 \pm 1.10$ | $19.35 \pm 2.55$ | $-2.19 \pm 8.77$ | $-10.38 \pm 3.14$ |
| SurrogateSHAP (**Ours**) | $\textbf{57.37} \pm \textbf{0.62}$ | $23.58 \pm 0.68$ | $\textbf{33.07} \pm \textbf{3.82}$ | $\textbf{31.39} \pm \textbf{11.66}$ | $\textbf{18.48} \pm \textbf{9.20}$ |

Sparsified-FT and SurrogateSHAP use an identical number of coalitions.

## E.3. Surrogate Architecture Analysis

We validated our surrogate model choice by comparing XGBoost against linear, 2-layer MLP, and random forest using 5-fold cross-validation. We utilized the coalition sampling budgets $M$ specified in Table 2. As shown in Table S.7, XGBoost consistently achieved the lowest MSE across four of the five utility metrics; despite the MLP's performance on CIFAR-20 FID, XGBoost proved to be the most robust architecture overall. These results support the use of gradient-boosted trees for capturing non-linear feature interactions, similar to findings in (Butler et al., 2025).

*Table S.7.* 5-fold cross-validation MSE of different surrogate architectures. Values in parentheses denote standard deviations. The best result in each column is shown in bold.

| Surrogate | CIFAR-20 (FID) | CIFAR-20 (IS) | ArtBench (Aesthetic) | Fashion (LPIPS, $\times 10^{-3}$) | Fashion (Div., $\times 10^{-3}$) |
|---|---|---|---|---|---|
| Linear | 595.4 (70.1) | 0.517 (0.068) | 0.236 (0.052) | 1.4 (0.1) | 1.0 (0.1) |
| MLP | **95.0 (43.0)** | 0.117 (0.015) | 0.356 (0.041) | 8.7 (1.5) | 8.2 (1.2) |
| RandomForest | 345.9 (63.1) | 0.359 (0.039) | 0.115 (0.017) | 1.2 (0.2) | 0.7 (0.1) |
| XGBoost | 139.9 (30.1) | **0.114 (0.050)** | **0.085 (0.023)** | **1.1 (0.2)** | **0.6 (0.1)** |

## E.4. Hyperparameter Sensitivity: D-TRAK & TRAK

We evaluate the performance of TRAK and D-TRAK across a range of regularization strengths ($\lambda$) and projection dimensions ($k$). Our results indicate that performance is relatively invariant to the choice of projection dimension $k$, whereas the regularization strength $\lambda$ serves as a critical hyperparameter (Figure S.15).

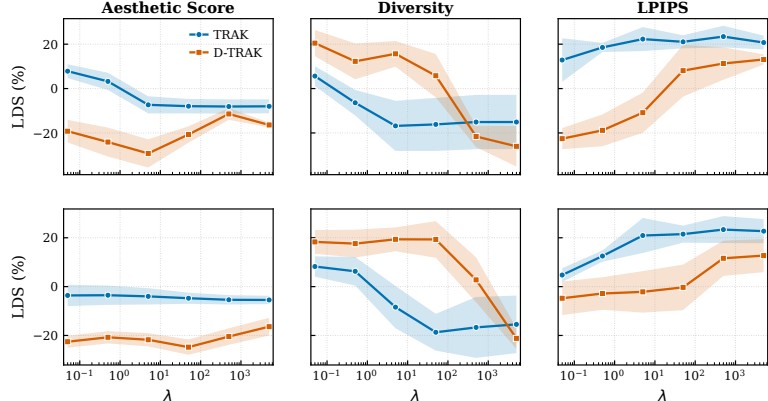

*Figure S.15.* LDS (%) with $\alpha = 0.5$ of TRAK and D-TRAK as a function of the regularization strength $\lambda$, evaluated on ArtBench Post-Impressionism (Aesthetic Score) and Fashion-Product (Diversity and LPIPS), for $k = 4,096$ (top) and $k = 32,768$ (bottom). Shaded bands denote 95% CIs across seeds.

### E.5. Counterfactual results

We report full counterfactual retraining results across all evaluated percentiles. For each dataset/metric, we include both *bottom-percentile removal* Table 3 and *top-percentile removal* Table S.8, which together assess whether an attribution method ranks contributors correctly. Table S.11 presents the complete results on ArtBench (Post-Impressionism), and Table S.10, S.9, S.12, and S.13 provide the corresponding results on CIFAR-20 (Inception Score/FID) and Fashion Product (LPIPS/Diversity).

*Table S.8.* **Counterfactual performance after top removal.** We report the relative change (%) in $\Delta\mathcal{F}$ after removing the top $k\%$ most influential groups identified by each method. Results are mean $\pm$ std over five random seeds.

| Method | Top 5% | Top 10% | Top 15% | Top 20% |
|---|---|---|---|---|
| **CIFAR-20 ( $\Delta$ FID $\uparrow$ )** | | | | |
| CLIP Score | $13.42 \pm 3.57$ | $17.53 \pm 1.58$ | $17.99 \pm 2.11$ | $18.09 \pm 3.51$ |
| LOO | $18.19 \pm 3.42$ | $17.48 \pm 2.68$ | $22.37 \pm 3.14$ | $25.07 \pm 3.60$ |
| SurrogateSHAP (Ours) | $\mathbf{27.76 \pm 4.60}$ | $\mathbf{31.70 \pm 2.47}$ | $\mathbf{32.20 \pm 3.34}$ | $\mathbf{32.73 \pm 2.14}$ |
| **ArtBench ( $\Delta$ Aesthetic Score $\downarrow$ )** | | | | |
| Avg. Aesthetic Score | $\mathbf{-4.30 \pm 1.10}$ | $-4.99 \pm 1.13$ | $-3.69 \pm 0.29$ | $-2.36 \pm 1.69$ |
| Avg. Pixel Similarity | $-0.23 \pm 2.03$ | $-2.03 \pm 0.57$ | $-3.98 \pm 1.19$ | $-3.26 \pm 3.21$ |
| Relative Influence | $-0.76 \pm 2.06$ | $-4.09 \pm 1.03$ | $3.46 \pm 0.18$ | $-2.08 \pm 1.01$ |
| SurrogateSHAP (Ours) | $-3.30 \pm 1.58$ | $\mathbf{-5.76 \pm 0.49}$ | $\mathbf{-5.75 \pm 0.63}$ | $\mathbf{-7.71 \pm 1.38}$ |
| **Fashion Product ( $\Delta$ Diversity $\downarrow$ )** | | | | |
| Avg. GradSim | $-6.52 \pm 5.82$ | $-6.09 \pm 9.24$ | $-5.44 \pm 3.33$ | $-3.59 \pm 2.03$ |
| D-TRAK | $2.21 \pm 2.24$ | $-7.18 \pm 6.23$ | $-5.05 \pm 9.52$ | $\mathbf{-4.14 \pm 2.01}$ |
| SurrogateSHAP (Ours) | $\mathbf{-16.11 \pm 1.71}$ | $\mathbf{-17.49 \pm 8.34}$ | $-7.05 \pm 7.48$ | $-0.60 \pm 7.41$ |

### E.5.1. CIFAR-20

Figure S.17 shows the top six contributors to the FID score on CIFAR-20, as identified by SurrogateSHAP.

*Table S.9.* **Full counterfactual retraining results on CIFAR-20 (FID).** Top: *top-percentile removal* (higher is better). Bottom: *bottom-percentile removal* (lower is better). Results are reported over five seeds.

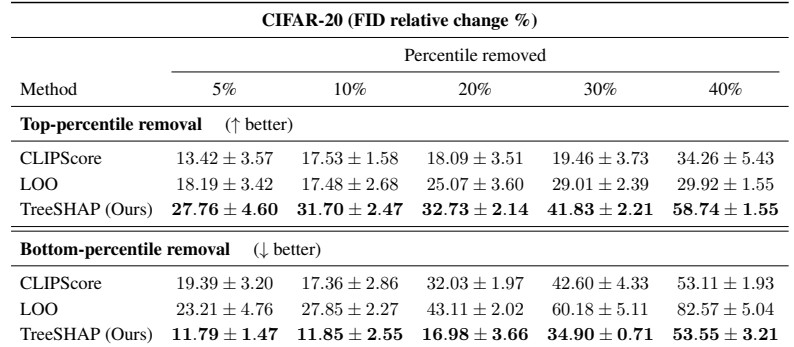

| | **CIFAR-20 (FID relative change %)** | | | | |
|---|---|---|---|---|---|
| | | | Percentile removed | | |
| Method | 5% | 10% | 20% | 30% | 40% |
| **Top-percentile removal** ( $\uparrow$ better) | | | | | |
| CLIPScore | $13.42 \pm 3.57$ | $17.53 \pm 1.58$ | $18.09 \pm 3.51$ | $19.46 \pm 3.73$ | $34.26 \pm 5.43$ |
| LOO | $18.19 \pm 3.42$ | $17.48 \pm 2.68$ | $25.07 \pm 3.60$ | $29.01 \pm 2.39$ | $29.92 \pm 1.55$ |
| TreeSHAP (Ours) | $\mathbf{27.76 \pm 4.60}$ | $\mathbf{31.70 \pm 2.47}$ | $\mathbf{32.73 \pm 2.14}$ | $\mathbf{41.83 \pm 2.21}$ | $\mathbf{58.74 \pm 1.55}$ |
| **Bottom-percentile removal** ( $\downarrow$ better) | | | | | |
| CLIPScore | $19.39 \pm 3.20$ | $17.36 \pm 2.86$ | $32.03 \pm 1.97$ | $42.60 \pm 4.33$ | $53.11 \pm 1.93$ |
| LOO | $23.21 \pm 4.76$ | $27.85 \pm 2.27$ | $43.11 \pm 2.02$ | $60.18 \pm 5.11$ | $82.57 \pm 5.04$ |
| TreeSHAP (Ours) | $\mathbf{11.79 \pm 1.47}$ | $\mathbf{11.85 \pm 2.55}$ | $\mathbf{16.98 \pm 3.66}$ | $\mathbf{34.90 \pm 0.71}$ | $\mathbf{53.55 \pm 3.21}$ |

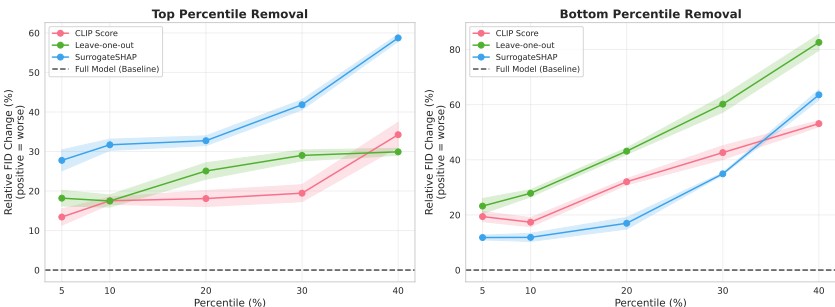

*Figure S.16.* Counterfactual evaluation on CIFAR-20. We report the relative change in FID ($\Delta\mathcal{F}\%$) after removing the top (left) and bottom (right) percentiles of contributors. The $x$-axis denotes the percentage of contributors removed.

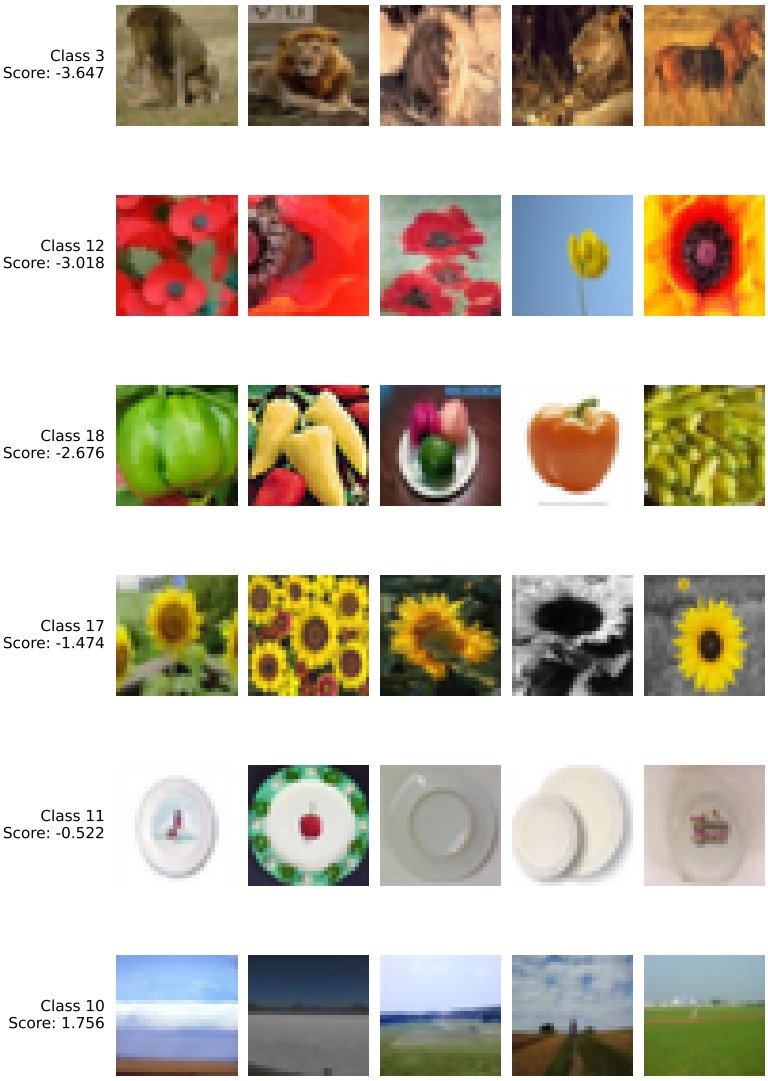

*Figure S.17.* **Influential Data Contributors for FID.** This figure visualizes the top six classes identified by SurrogateSHAP as having the highest marginal contribution to FID on the CIFAR-20. Each row displays representative training samples from the identified class alongside their computed importance score.

*Table S.10.* **Full counterfactual retraining results on CIFAR-20 (Inception Score).** Top: *top-percentile removal* (more negative is better). Bottom: *bottom-percentile removal* (larger/less negative is better). Results are reported over five seeds.

| | CIFAR-20 (Inception Score relative change %) | | | | |
|---|---|---|---|---|---|
| | Percentile removed | | | | |
| Method | 5% | 10% | 20% | 30% | 40% |
| **Top-percentile removal** ($\downarrow$ better) | | | | | |
| GMV | $-20.17 \pm 0.53$ | $-19.52 \pm 0.30$ | $-19.71 \pm 0.47$ | $-24.40 \pm 0.76$ | $-29.49 \pm 1.00$ |
| LOO | $-\mathbf{21.97} \pm \mathbf{0.39}$ | $-22.45 \pm 0.64$ | $-24.34 \pm 1.53$ | $-27.84 \pm 0.50$ | $-29.94 \pm 0.80$ |
| TreeSHAP (Ours) | $-20.87 \pm 1.25$ | $-\mathbf{26.55} \pm \mathbf{1.43}$ | $-\mathbf{31.05} \pm \mathbf{1.33}$ | $-\mathbf{35.50} \pm \mathbf{0.01}$ | $-\mathbf{40.03} \pm \mathbf{1.05}$ |
| **Bottom-percentile removal** ($\uparrow$ better) | | | | | |
| GMV | $-20.57 \pm 0.86$ | $-18.22 \pm 0.91$ | $-18.44 \pm 1.20$ | $-16.66 \pm 1.74$ | $-15.38 \pm 0.96$ |
| LOO | $-\mathbf{19.40} \pm \mathbf{0.55}$ | $-\mathbf{16.87} \pm \mathbf{0.56}$ | $-\mathbf{12.93} \pm \mathbf{0.11}$ | $-\mathbf{10.81} \pm \mathbf{1.07}$ | $-10.22 \pm 1.12$ |
| TreeSHAP (Ours) | $-19.82 \pm 0.59$ | $-17.20 \pm 1.25$ | $-\mathbf{12.93} \pm \mathbf{0.11}$ | $-10.97 \pm 0.82$ | $-\mathbf{9.36} \pm \mathbf{1.06}$ |

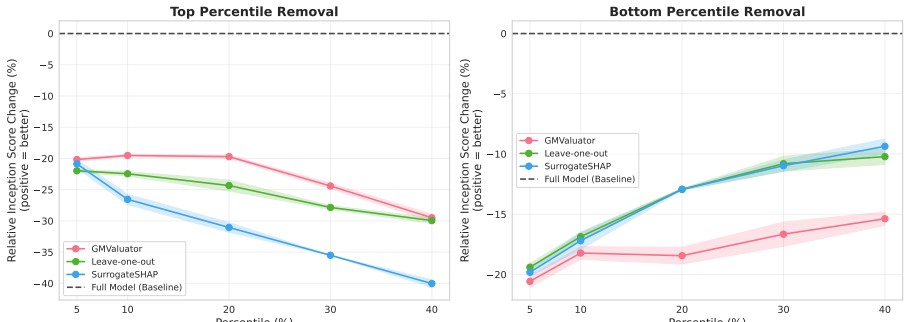

*Figure S.18.* Counterfactual evaluation on CIFAR-20. We report the relative change in Inception Score ($\Delta\mathcal{F}\%$) after removing the top (left) and bottom (right) percentiles of contributors. The $x$-axis denotes the percentage of contributors removed.

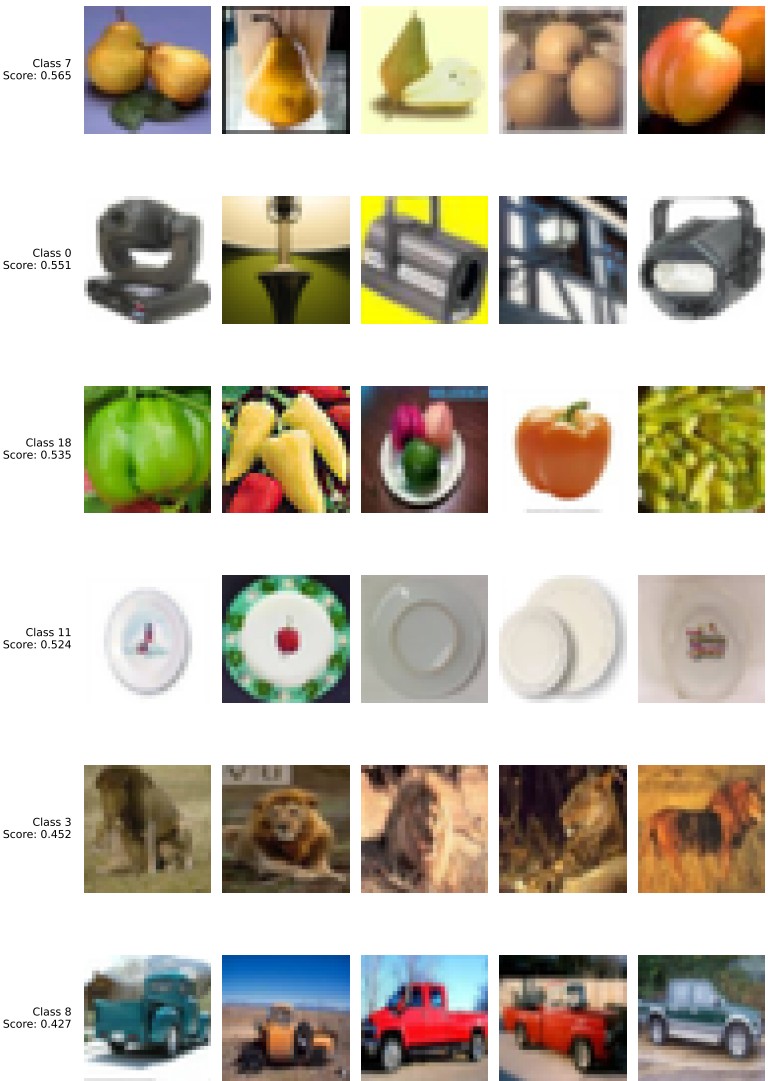

*Figure S.19.* **Influential Data Contributors for Inception Score.** This figure visualizes the top six classes identified by SurrogateSHAP as having the highest marginal contribution to Inception Score on the CIFAR-20. Each row displays representative training samples from the identified class alongside their computed importance score.

### E.5.2. ARTBENCH POST-IMPRESSIONISM

*Table S.11.* **Full counterfactual retraining results on ArtBench (Post-Impressionism).** Top: *top-percentile removal* (more negative is better). Bottom: *bottom-percentile removal* (larger is better). Results are reported over five seeds.

| | Avg. Aesthetic Score (relative change %) | | | | | | | | |
|---|---|---|---|---|---|---|---|---|---|
| | Percentile removed | | | | | | | | |
| Method | 1% | 2% | 3% | 4% | 5% | 10% | 15% | 20% | 30% |
| **Top-percentile removal** ($\downarrow$ better) | | | | | | | | | |
| Aesthetic | $-2.26 \pm 0.67$ | $-1.85 \pm 2.29$ | $-3.60 \pm 1.24$ | $-2.53 \pm 1.99$ | $\mathbf{-4.30 \pm 1.10}$ | $-4.99 \pm 1.13$ | $-3.69 \pm 0.29$ | $-2.36 \pm 1.69$ | $-2.40 \pm 0.70$ |
| Pixel Sim. | $-1.62 \pm 1.03$ | $\mathbf{-3.58 \pm 2.07}$ | $-0.46 \pm 0.68$ | $-3.19 \pm 0.68$ | $-0.23 \pm 2.03$ | $-2.03 \pm 0.57$ | $-3.98 \pm 1.19$ | $-3.26 \pm 3.21$ | $-1.63 \pm 0.58$ |
| Rel. Infl. | $\mathbf{-3.04 \pm 2.07}$ | $-2.17 \pm 2.07$ | $-2.21 \pm 0.67$ | $-2.73 \pm 2.59$ | $-0.76 \pm 2.06$ | $-4.09 \pm 1.03$ | $-3.46 \pm 0.18$ | $-2.08 \pm 1.01$ | $-2.18 \pm 2.53$ |
| TreeSHAP (Ours) | $-2.60 \pm 0.90$ | $-3.27 \pm 1.50$ | $\mathbf{-3.97 \pm 0.47}$ | $\mathbf{-4.29 \pm 1.58}$ | $-3.30 \pm 1.69$ | $\mathbf{-5.76 \pm 0.49}$ | $\mathbf{-5.75 \pm 0.63}$ | $\mathbf{-7.71 \pm 1.38}$ | $\mathbf{-5.73 \pm 0.58}$ |
| **Bottom-percentile removal** ($\uparrow$ better) | | | | | | | | | |
| Aesthetic | $-2.08 \pm 0.70$ | $-2.84 \pm 0.35$ | $-2.64 \pm 1.52$ | $-2.57 \pm 1.51$ | $-2.59 \pm 1.44$ | $-1.38 \pm 1.16$ | $-2.22 \pm 1.06$ | $-2.40 \pm 0.42$ | $+3.19 \pm 1.02$ |
| Pixel Sim. | $-3.63 \pm 2.10$ | $-1.88 \pm 0.72$ | $-2.63 \pm 1.09$ | $-1.80 \pm 0.86$ | $-1.32 \pm 1.05$ | $-1.68 \pm 1.78$ | $-0.80 \pm 1.07$ | $-2.20 \pm 1.13$ | $-1.64 \pm 1.76$ |
| Rel. Infl. | $-0.95 \pm 1.21$ | $\mathbf{-0.79 \pm 0.62}$ | $-1.79 \pm 1.11$ | $-1.23 \pm 1.02$ | $-2.93 \pm 2.72$ | $-1.25 \pm 2.28$ | $+1.08 \pm 0.94$ | $+0.17 \pm 0.65$ | $-0.06 \pm 2.24$ |
| TreeSHAP (Ours) | $\mathbf{-0.02 \pm 1.44}$ | $-1.09 \pm 0.53$ | $\mathbf{+0.65 \pm 1.06}$ | $-0.06 \pm 1.04$ | $\mathbf{+0.43 \pm 1.67}$ | $\mathbf{+2.54 \pm 2.38}$ | $\mathbf{+3.06 \pm 0.38}$ | $\mathbf{+3.62 \pm 2.37}$ | $\mathbf{+5.03 \pm 1.59}$ |

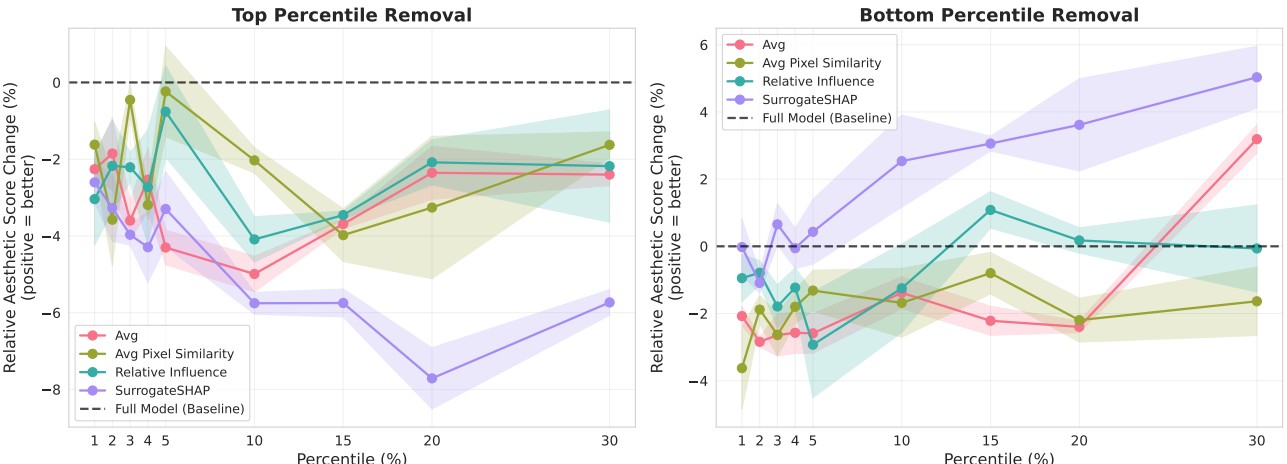

*Figure S.20.* Counterfactual evaluation on ArtBench (Post-impressionism). We report the relative change in average aesthetic score ($\Delta\mathcal{F}\%$) after removing the top (left) and bottom (right) percentiles of contributors. The $x$-axis denotes the percentage of contributors removed.

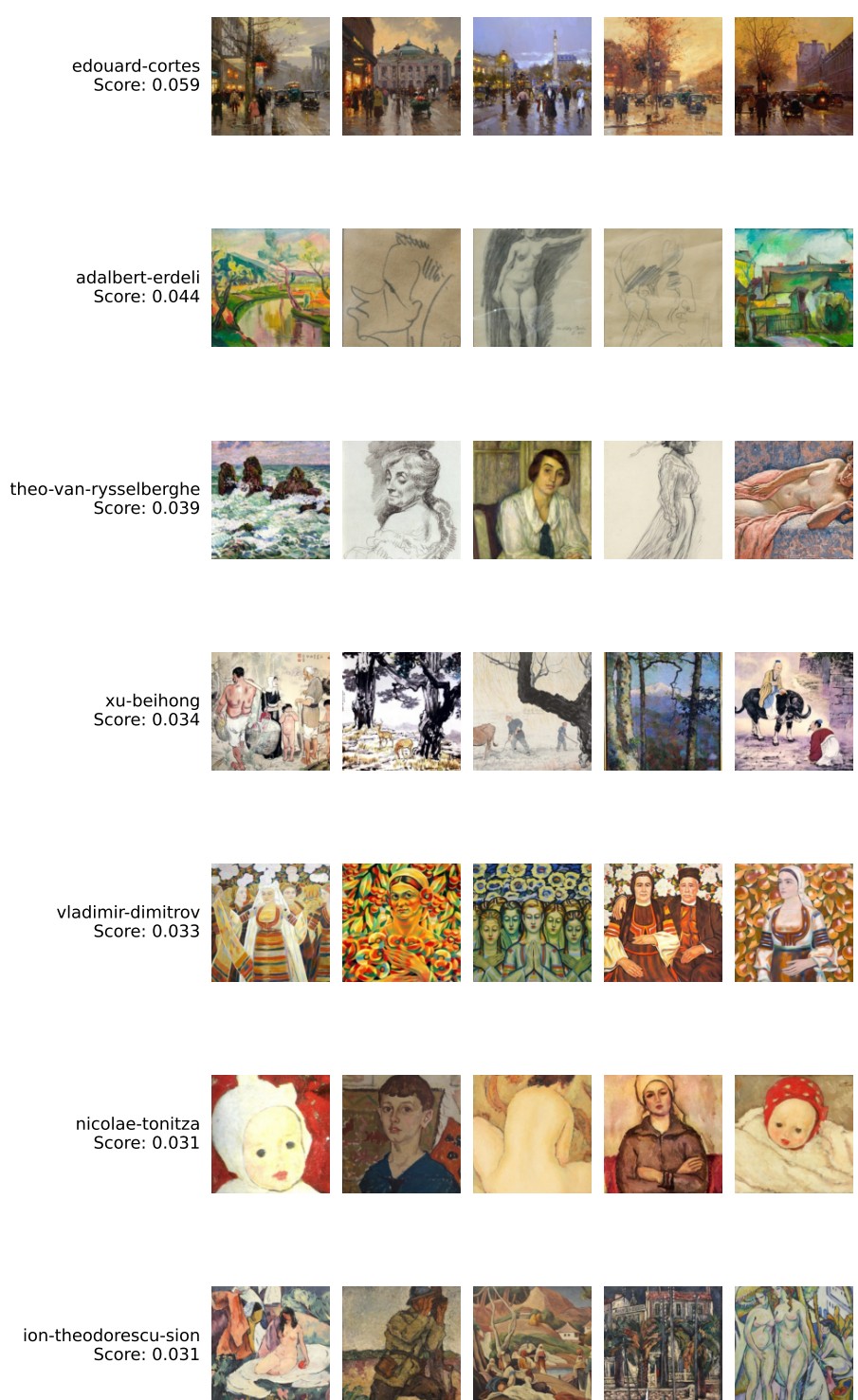

*Figure S.21.* **Influential Data Contributors for Artistic Aesthetics.** This figure visualizes the top seven artists identified by Surrogate-SHAP as having the highest marginal contribution to the mean aesthetic score on the ArtBench (Post-Impressionism). Each row displays representative training samples from the identified artist alongside their computed importance score.

### E.5.3. FASHION PRODUCT

*Table S.12.* **Full counterfactual retraining results on Fashion Product (LPIPS).** Top: *top-percentile removal* (higher is better). Bottom: *bottom-percentile removal* (lower is better). Results are reported over five seeds.

| | | | | | | | |
|---|---|---|---|---|---|---|---|
| **Fashion Product (LPIPS relative change %)** | | | | | | | |
| | | | | Percentile removed | | | |
| Method | 1% | 2% | 5% | 10% | 15% | 20% | 30% |
| **Top-percentile removal** (↑ better) | | | | | | | |
| Avg. PixelSim | $5.15 \pm 0.49$ | $\mathbf{7.76 \pm 3.95}$ | $3.13 \pm 1.90$ | $3.09 \pm 2.59$ | $3.09 \pm 1.85$ | $1.32 \pm 3.58$ | $-3.93 \pm 4.00$ |
| Rel. Infl. | $\mathbf{7.33 \pm 1.10}$ | $3.16 \pm 0.90$ | $\mathbf{4.64 \pm 1.97}$ | $3.39 \pm 0.16$ | $5.89 \pm 2.96$ | $\mathbf{8.74 \pm 3.98}$ | $\mathbf{8.90 \pm 1.45}$ |
| SurrogateSHAP | $6.07 \pm 1.99$ | $1.27 \pm 1.51$ | $3.33 \pm 3.51$ | $\mathbf{8.37 \pm 1.94}$ | $\mathbf{8.477 \pm 2.83}$ | $3.57 \pm 2.30$ | $5.82 \pm 0.33$ |
| **Bottom-percentile removal** (↓ better) | | | | | | | |
| Avg. PixelSim | $\mathbf{3.29 \pm 2.15}$ | $9.30 \pm 3.43$ | $2.45 \pm 2.09$ | $7.78 \pm 0.86$ | $11.29 \pm 2.93$ | $6.83 \pm 2.81$ | $7.23 \pm 3.07$ |
| Rel. Infl. | $3.58 \pm 0.20$ | $\mathbf{5.20 \pm 1.98}$ | $3.87 \pm 2.45$ | $3.73 \pm 1.71$ | $4.83 \pm 0.91$ | $2.73 \pm 4.83$ | $2.98 \pm 2.51$ |
| SurrogateSHAP | $5.86 \pm 1.69$ | $5.73 \pm 3.95$ | $\mathbf{1.43 \pm 3.32}$ | $\mathbf{0.17 \pm 2.54}$ | $\mathbf{-0.62 \pm 3.45}$ | $\mathbf{-1.83 \pm 2.85}$ | $\mathbf{-0.96 \pm 1.15}$ |

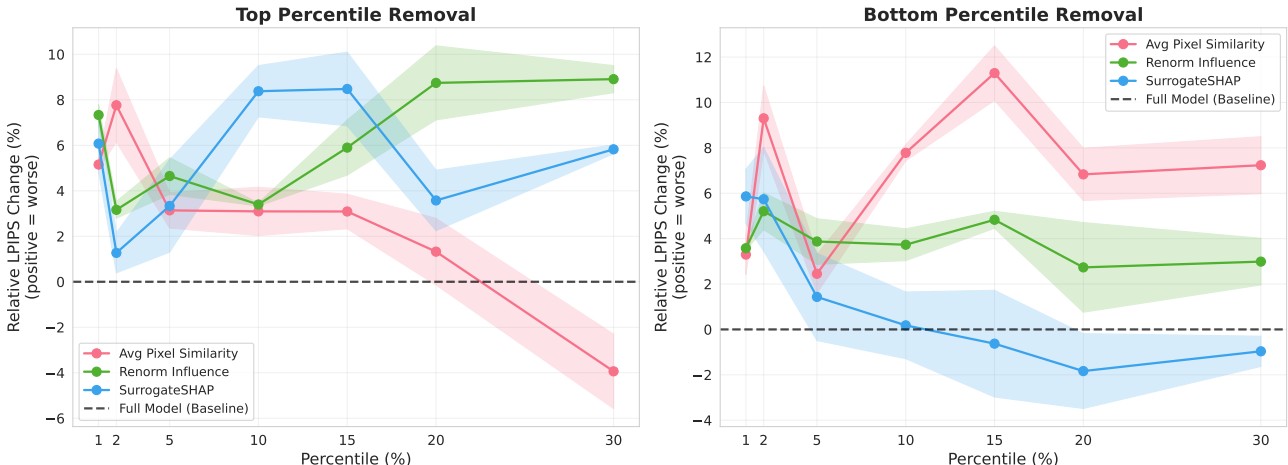

*Figure S.22.* Counterfactual evaluation on Fashion-Product. We report the relative change in mean LPIPS ($\Delta\mathcal{F}\%$) after removing the top (left) and bottom (right) percentiles of contributors. The $x$-axis denotes the percentage of contributors removed.

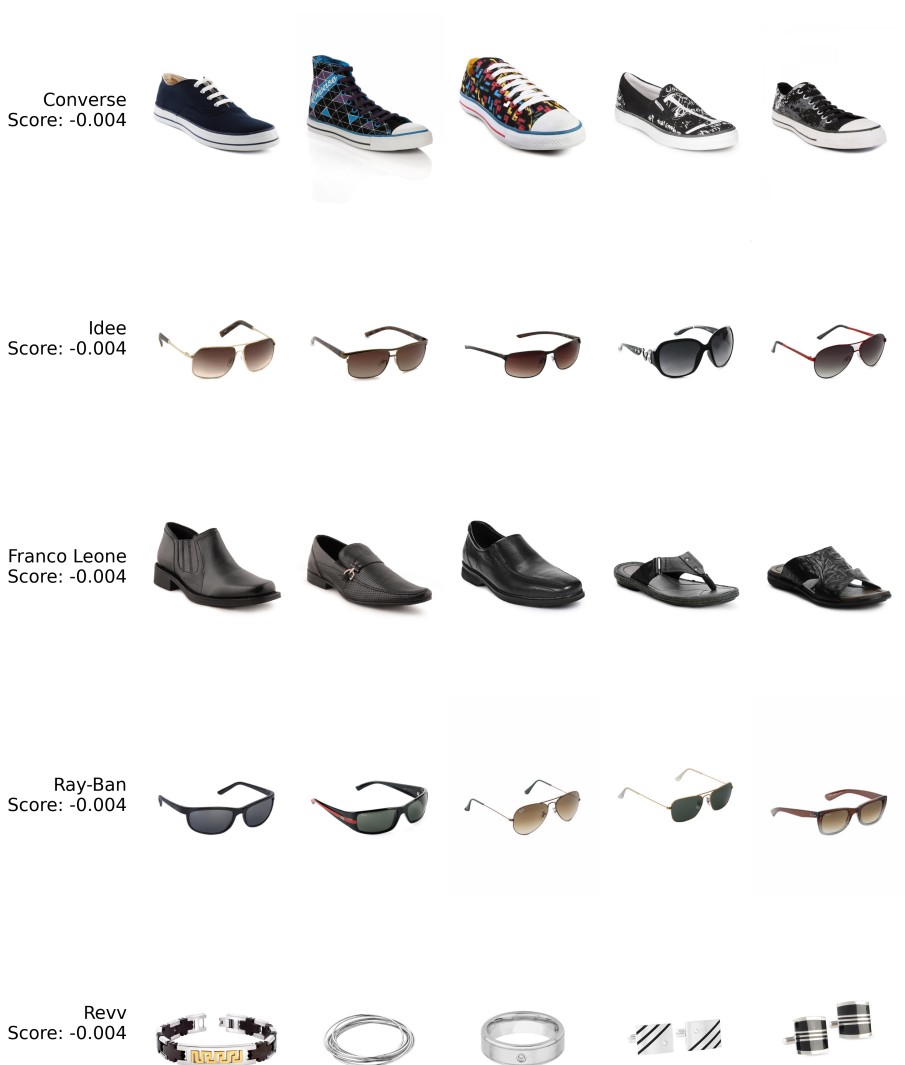

*Figure S.23.* **Influential Data Contributors for LPIPS.** This figure visualizes the top five brands identified by SurrogateSHAP as having the highest marginal contribution to LPIPS on the Fashion-Product. Each row displays representative training samples from the identified brand alongside their computed importance score.

*Table S.13.* **Full counterfactual retraining results on Fashion Product (Diversity).** Top: *top-percentile removal* (lower/more negative is better). Bottom: *bottom-percentile removal* (higher is better). Results are reported over five seeds.

| | Fashion Product (Diversity relative change %) | | | | | | |
|---|---|---|---|---|---|---|---|
| | Percentile removed | | | | | | |
| Method | 1% | 2% | 5% | 10% | 15% | 20% | 30% |
| **Top-percentile removal** ($\downarrow$ better) | | | | | | | |
| Avg. GradSim | $-6.09 \pm 2.83$ | $-3.78 \pm 2.70$ | $-6.52 \pm 5.82$ | $-6.09 \pm 9.24$ | $-5.44 \pm 3.33$ | $-3.59 \pm 2.03$ | $-4.32 \pm 4.16$ |
| D-TRAK | $-\mathbf{8.36} \pm \mathbf{4.46}$ | $-\mathbf{12.58} \pm \mathbf{6.49}$ | $+2.21 \pm 2.24$ | $-7.18 \pm 6.33$ | $+5.05 \pm 9.52$ | $-\mathbf{4.14} \pm \mathbf{2.01}$ | $-3.20 \pm 7.20$ |
| SurrogateSHAP | $-1.20 \pm 5.16$ | $-8.96 \pm 0.07$ | $-\mathbf{16.11} \pm \mathbf{1.71}$ | $-\mathbf{17.49} \pm \mathbf{8.34}$ | $-\mathbf{7.05} \pm \mathbf{7.48}$ | $-0.60 \pm 7.41$ | $-\mathbf{11.15} \pm \mathbf{6.19}$ |
| **Bottom-percentile removal** ($\uparrow$ better) | | | | | | | |
| Avg. GradSim | $-3.63 \pm 2.99$ | $-8.13 \pm 2.74$ | $+3.19 \pm 4.98$ | $-6.53 \pm 2.44$ | $+4.83 \pm 3.54$ | $+6.70 \pm 6.17$ | $+2.94 \pm 2.16$ |
| D-TRAK | $-\mathbf{1.56} \pm \mathbf{3.65}$ | $-2.73 \pm 6.16$ | $-1.12 \pm 4.25$ | $-4.49 \pm 4.98$ | $-0.50 \pm 2.14$ | $-7.24 \pm 5.95$ | $-10.57 \pm 3.40$ |
| SurrogateSHAP | $-2.24 \pm 5.44$ | $-\mathbf{2.06} \pm \mathbf{5.43}$ | $+\mathbf{9.36} \pm \mathbf{3.46}$ | $+\mathbf{5.22} \pm \mathbf{5.01}$ | $+\mathbf{12.75} \pm \mathbf{3.34}$ | $+\mathbf{7.46} \pm \mathbf{4.44}$ | $+\mathbf{7.19} \pm \mathbf{3.55}$ |

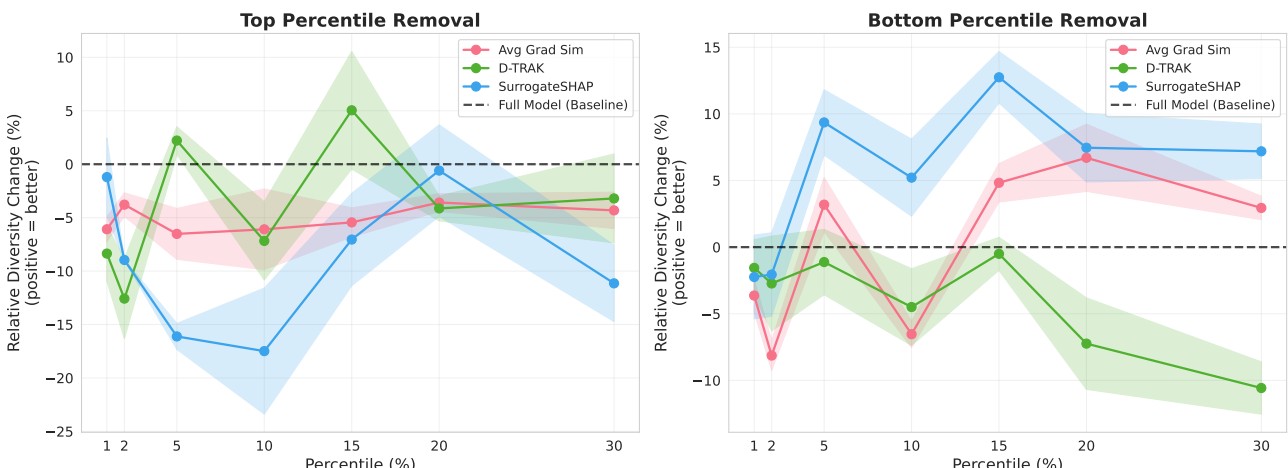

*Figure S.24.* Counterfactual evaluation on Fashion-Product. We report the relative change in diversity ($\Delta \mathcal{F}\%$) after removing the top (left) and bottom (right) percentiles of contributors. The $x$-axis denotes the percentage of contributors removed.

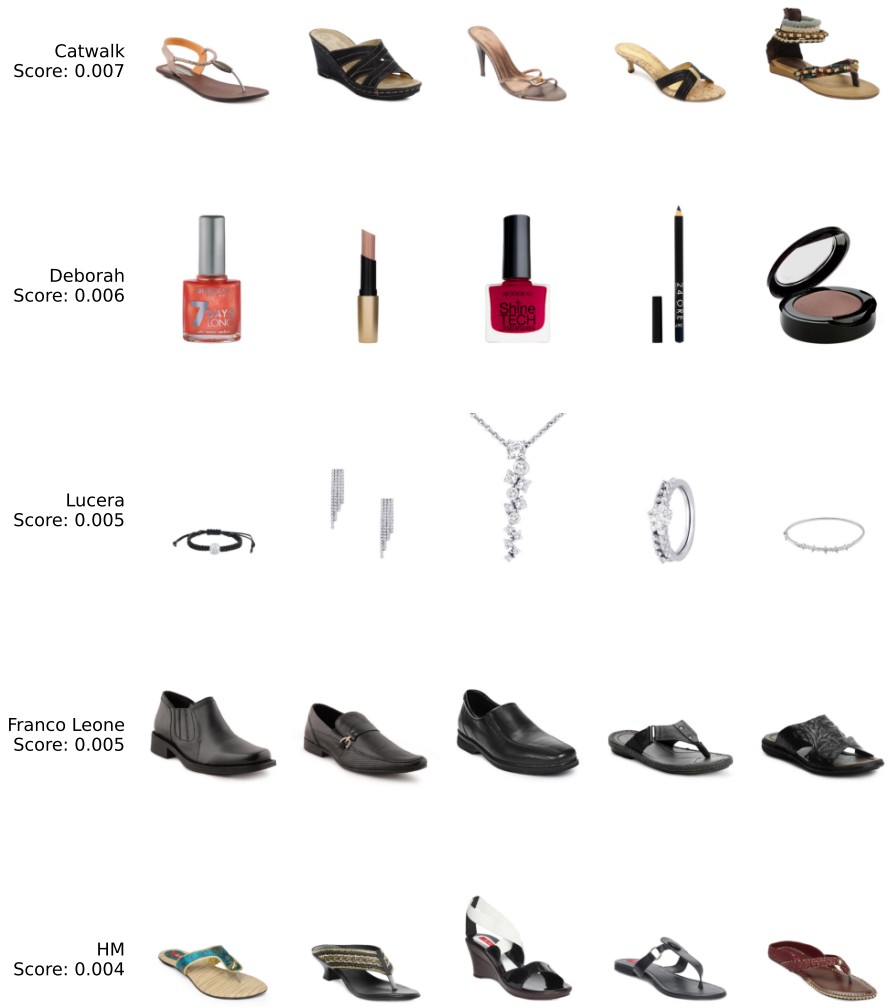

*Figure S.25.* **Influential Data Contributors for Diversity.** This figure visualizes the top five brands identified by SurrogateSHAP as having the highest marginal contribution to Diversity on the Fashion-Product. Each row displays representative training samples from the identified brand alongside their computed importance score.

# F. Additional details for the dermatology data auditing case study

**ISIC Dataset**  We use dermoscopic lesions (which offer a magnified view of the patient's skin) from the ISIC archive (Codella et al., 2018; Tschandl et al., 2018; Combalia et al., 2019; Rotemberg et al., 2021), specifically the 2019 and 2020 ISIC challenge datasets, for this analysis. We exclude non-dermoscopic images and images lacking sex labels. After partitioning, we are left with 76295 images.

We then fine-tune a Stable Diffusion model[22] on dermoscopic images using hospital-conditioned captions. To create the finetuning dataset, we sample a maximum of 1200 images from each of the 10 hospital sites, while balancing it in terms of sex and melanoma. For hospitals with less than 1200 images, we sample all the images. After the preprocessing step, we are left with 9922 images with the distribution shown in Figure S.26. The diffusion model is fine-tuned using LoRA with rank r=32, corresponding to 3.2M parameters. The prompt is set to ``Image of a dermoscopic lesion from Hospital {hospital site}'' for each image. The LoRA parameters are trained using the AdamW optimizer for 100 epochs with a learning rate of $1 \times 10^{-5}$. The images are generated at inference time using the PNDM scheduler for 100 steps.

*Model property* Let $\mathcal{H} = \{1, ..., H\}$ denote the hospital sites, treated as the data contributors in our setup, with $H = 10$. We first train independent sex and melanoma classifiers on the ISIC dataset, yielding probabilistic predictors $f_{\text{sex}}(x) \in [0, 1]$ and $f_{\text{mel}}(x) \in [0, 1]$. We use vision transformer-based models (ViT-Base architectures) (Dosovitskiy et al., 2021) for both the prediction tasks. We start with classifiers pre-trained on ImageNet (Deng et al., 2009) and then replace the 1000-class linear classification head with a new linear head suited for binary prediction. We optimize the network using an Adam optimizer with a cosine learning rate scheduler with an initial learning rate of $1 \times 10^{-4}$. We train for 10 epochs with a batch size of 256. The sex classifier achieves a ROC-AUC of 0.752, and the melanoma classifier achieves a ROC-AUC of 0.937.

To quantify the spurious correlation, we sample coalitions $S \subseteq \mathcal{H}$ from the Shapley distribution. For each coalition, we generate a set of images $\{x_j\}_{j=1}^K \sim p_\theta(\cdot \mid S)$ by restricting the sampled prompt to the hospital sites in $S$. The coalition utility is defined as the dependence between the two probes on the generated images:

$$v(S) = \rho\left(\{f_{\text{sex}}(x_j)\}_{j=1}^K, \{f_{\text{mel}}(x_j)\}_{j=1}^K\right),$$

where $\rho$ is the Spearman correlation and $K = 256$. Intuitively, a large $|v(S)|$ indicates a potentially spurious association between sex and melanoma predictions induced by the coalition's generated distribution. Finally, we evaluate $v(S)$ for 100 sampled coalitions and use SurrogateSHAP to compute the attributions for each hospital site.

---

[22]https://huggingface.co/stable-diffusion-v1-5/stable-diffusion-v1-5

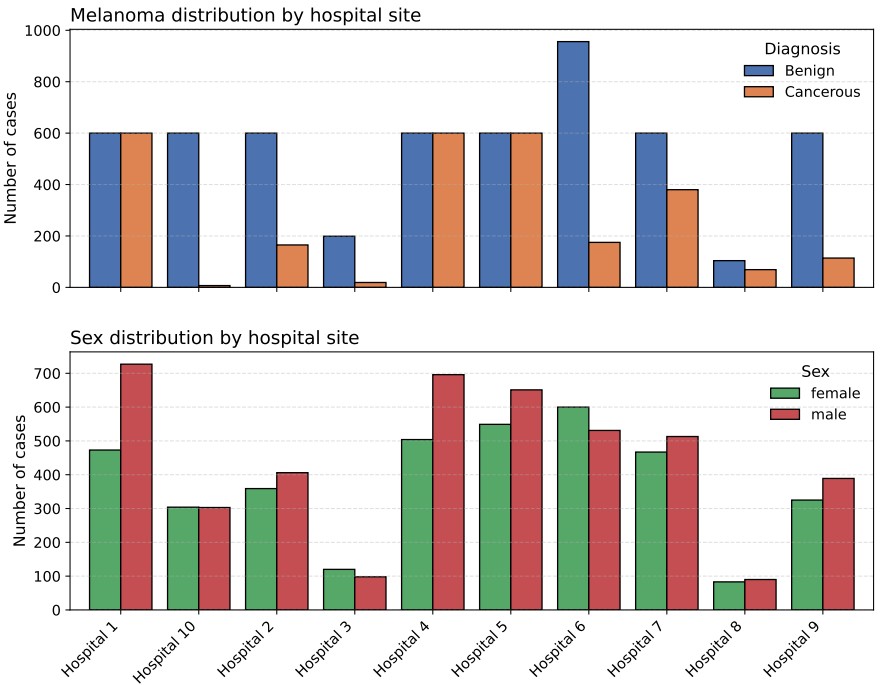

*Figure S.26.* Distribution of the fine-tuning data for the Stable Diffusion model across different hospital sites in terms of the diagnosis and patient sex.

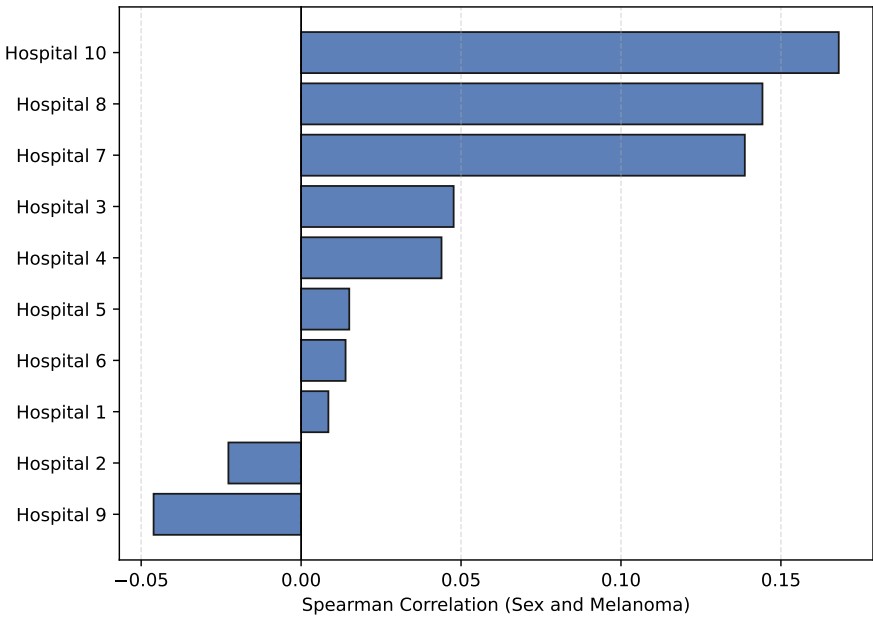

*Figure S.27.* Spearman correlation between sex and melanoma binary labels in the training data across different hospital sites. Hospital 10 has the maximum correlation $r = 0.17, p = 6.3 \times 10^{-6}$.

