# OpenReview forum: "SurrogateSHAP: Training-Free Contributor Attribution for Text-to-Image (T2I) Models"
_ICML.cc/2026/Conference — ICML 2026 regular_

### Official Review · Reviewer_tE6D · 2026-02-24

**Soundness:** 3
**Presentation:** 3
**Significance:** 3
**Originality:** 3
**Overall Recommendation:** 5
**Confidence:** 3

**Summary:**

This paper proposes a training-free framework named SurrogateSHAP to address the attribution problem of data contributors in T2I diffusion models. Due to the double-counting bottleneck in evaluating data value using Shapley Values, the authors introduce two core innovations:

- Training-Free Proxy Game: Reinterprets the target model as a hybrid of conditional generators, directly approximating the utility of retrained models through inference.

- Tree-Based Proxy and TreeSHAP: Utilizes gradient-boosted trees to approximate complex utility functions and employs TreeSHAP for parsing Shapley values, significantly enhancing sampling efficiency.

**Compliance With Llm Reviewing Policy:**

Affirmed.

**Final Justification:**

My concerns are all resolved. My recommendation is to accept.

**Key Questions For Authors:**

1. If the removed subset of contributors is extremely large or contains highly unique data distributions, leading to catastrophic forgetting or severe degradation in the retrained model, does this theoretical bound remain sufficiently tight?

**Limitations:**

Yes. Although the authors do not have a separate section called "Limitations," they do discuss the limitations and potential risks of their approach in various sections of the text, particularly the "Impact Statements" section and the experimental setup section.

**Strengths And Weaknesses:**

Strengths

1. The authors not only propose a heuristic approach but also provide theoretical bounds on approximation error between agent games and re-training games based on representation space stability.

2. By completely bypassing the subset re-training process, this method achieves significant computational efficiency advantages.

3. The paper evaluates performance using multiple utility metrics across diverse datasets, including CIFAR-20, ArtBench, and Fashion-Product.

4. In an audit case involving dermatopathology data (ISIC dataset), SurrogateSHAP successfully pinpointed the specific data source (hair artifacts from Hospital 7) responsible for generating spurious correlations between gender and melanoma, demonstrating strong practical guidance value.

Weakness

1. Fundamentally, LoRA is merely a Low-Rank Approximation and cannot be considered fully equivalent to full-parameter retraining. Using an *approximate method* as the Ground Truth to validate the accuracy of *another approximate method* (the SurrogateSHAP proxy) compromises the rigor of the evaluation. It raises a critical question: is the proposed method actually approximating the true Shapley Value, or is it merely approximating the Shapley Value of the LoRA fine-tuning process?

2. Taking a massive model like FLUX as an example, even a single inference pass to generate high-quality images is highly time-consuming. Assuming a sampling budget of $M=512$ or higher, where each subset requires generating hundreds of images to evaluate its diversity or aesthetics, this immense computational overhead for inference remains an exceptionally heavy burden in practice.

---

> ### Author Rebuttal · Authors · 2026-03-29
>
> Dear reviewer tE6D,
>
> Thank you for the kind words and for voting for acceptance. We’re thrilled to hear that you recognized both the theoretical and practical contributions of our work, and that the reviewer found the clinical use case meaningful. In the space below, we respond to the reviewer’s specific concerns.
>
> ### **Re: Proxy target & definition of retraining game.**
> We thank the reviewers for highlighting this point. We agree that the term retraining game is imprecise in the current draft and may be read as referring specifically to full end-to-end retraining from scratch, which is not our intent. Due to space limitations, we refer the reviewer to our response to **Reviewer wV9C — Proxy target & definition of retraining game** for a detailed clarification.
>
> ### **Re: Computational practicality.**
> We thank the reviewer for raising this important practical concern. We agree that for a model like FLUX (12B), evaluating coalition utilities remains computationally expensive even when only inference is required. However, our claim is not that our method makes this cost negligible, but rather that it reduces an otherwise intractable problem into a practically feasible one.
>
> Furthermore, SurrogateSHAP remains effective under limited budgets: with $M=400$ (about one day on 4$\times$ H200 GPUs), it still achieves 24.24% LDS on FLUX. The computation cost can be reduced further by using fewer generated samples per subset. Thus, while large-scale model evaluation inevitably incurs substantial cost, our method remains practical and improves with additional compute. Crucially, under these same computational constraints, competing baselines perform worse. For instance, methods requiring gradient computation (such as TRAK, where computing a single set of gradients takes $\sim$16 hours on a single H200) yield near-zero or even negative LDS.
>
> ### **Re: Proxy behavior under large removals**
> We thank the reviewer for this important question. To probe this regime, we evaluated the average difference in coalition utility between the proxy and original games after removing 75% of the players. As shown below, the error is larger than reported across all coalitions (Table 1), but it does not increase catastrophically, suggesting that the proxy remains empirically informative.
>
> | Metric | CIFAR-20 (FID) | CIFAR-20 (IS) | ArtBench Post-impressionism (Aesthetic score) | Fashion Product (LPIPS) | Fashion Product (Diversity) |
> |---|---:|---:|---:|---:|---:|
> | NRMSE |0.122|0.087|0.164|0.348 |0.373|
> | NMAE | 0.100|0.072|0.124|0.293| 0.272|
> ------------------------
> If there is any other information we can provide during the discussion period that would allow you to further raise your score, please let us know.
>
> Thanks again,
>
> **Due to space constraints, we have dedicated the remainder of the space to responding to Reviewer wV9C's questions/concerns.**
>
> ---
>
> ## **Questions & concerns from Reviewer wV9C:**
> ### **Re: Hyperparameter sweeps.**
> Following the reviewer’s suggestion, we provide results of hyperparameter sweeps for both TRAK and D-TRAK, varying the projection dimension $k \in $ {4096,32768} and the regularization parameter $\lambda$  from $5e^{-2}$ to $5e^{2}$. **Due to space constraints, we only show TRAK's result.**
>
> |  | $5e^{-2}$ | $5e^{-1}$ | $5e$ | $5e^1$ | $5e^2$ |
> |---|---:|---:|---:|---:|---:|
> | **Artbench (aesthetic score)** | -3.61 (4.36) | -3.52 (3.96) | -3.99 (3.34) | -4.75 (2.28) | -5.40 (1.93) |
> | **Fashion-product (Diversity)**| 8.23 (4.18) | 6.27 (5.86) | -8.40 (8.68) | -18.69 (7.54) | -16.70 (12.48) |
> | **Fashion-product (LPIPS)**  | 4.75 (2.82) | 12.52 (2.31) | 20.92 (7.17) | 21.49 (3.49) | 23.36 (5.53) |
>
> Similar to findings in [1], our results show that changing $k$ does not significantly affect performance, while tuning $\lambda$ can improve TRAK-based methods in some settings. We promise to update the revised manuscript with tuned results. At the same time, this highlights a practical limitation: hyperparameters tied to gradient computation are extremely expensive to tune (e.g., a single 100-timestep gradient run on FLUX can take up to 16 hours on one H200 GPU). By contrast, although SurrogateSHAP requires computation for sample coalitions, its hyperparameter tuning is confined to the XGBoost surrogate, and 5-fold CV completes in seconds (see below). This makes it more efficient to allocate additional compute to sampling more coalitions rather than to expensive hyperparameter sweeps.
>
> ### **Re: GBT training time.**
> Following the reviewer’s suggestion, we measured the wall-clock time of GBT training with a 5-fold CV for the settings in Table 2. The average runtime is 0.86s for CIFAR-20, 1.65s for ArtBench Post-impressionism, and 26.62s for Fashion-Product. Compared with coalition utility evaluation, which takes at least 10 minutes per coalition for CIFAR-20 and ArtBench (Table 1), this cost is negligible. We will include these numbers in the revision.
>
> Reference
>
> 1: https://arxiv.org/pdf/2311.00500

---

> > ### Author Rebuttal · Reviewer_tE6D · 2026-04-04
> >
> > Thanks for your responses. My previous concerns are all resolved, and the positive rating remains.

---

> > > ### Author Response · Authors · 2026-04-05
> > >
> > > Thank you for your thoughtful follow-up. We are very glad that our responses resolved your concerns, and we sincerely appreciate your continued positive rating.

---

### Official Review · Reviewer_h3h7 · 2026-03-13

**Soundness:** 3
**Presentation:** 2
**Significance:** 3
**Originality:** 3
**Overall Recommendation:** 5
**Confidence:** 4

**Summary:**

This paper introduce a training-free method for TDA on Text-to-Image model to resolve the prohibitive computational cost of Data Shapley methods. The method relies on training-free proxy and a surrogate model-based procedure. The paper also evaluates the method on several standard benchmarks.

**Compliance With Llm Reviewing Policy:**

Affirmed.

**Final Justification:**

I will maintain my positive score on this paper.

**Key Questions For Authors:**

What are the insights (qualitatively) of the "theoretical fidelity of the proxy" (section 4.1.2) section?

**Limitations:**

yes

**Strengths And Weaknesses:**

**Strength**

Training-free TDA method is highly appreciated and practical for modern AI models. Though gradient-based methods are proposed for Text-to-Image models, the difficulties in the implementation and caveats in the target function (noise distribution) block the adoption of these methods. While TDA is valuable for some downstream applications, such as pricing.

The clever training-free coalition evaluation stated in Section 4.1 is reasonable to me. The evaluation in Section 5 is also solid.

**Weakness**

The notation in Section 3.2 is confusing and hard to follow. The authors may want to rewrite the “Contributor-label correspondence” paragraph.

---

> ### Author Rebuttal · Authors · 2026-03-29
>
> Dear reviewer h3h7,
>
> Thank you for the thoughtful feedback and for supporting acceptance. We are glad that you found our training-free approach compelling and the evaluation solid. We will revise the *Contributor-label correspondence* section to improve clarity and presentation. Below, we address the reviewer’s questions.
>
> ### **Re: Qualitative insights of theoretical fidelity of the proxy.**
> We thank the reviewer for this question. This Section provides a local robustness justification for the proxy: it proves that if training on a subset $S$ only mildly perturbs the model’s class-conditional output distribution in a utility-aligned feature space, then any score that is Lipschitz (e.g., CLIP-score) or locally Lipschitz (e.g., FID) will also change only mildly. Qualitatively, this means that the proxy is reliable precisely when the training game model remains close to the target model in the semantic/perceptual representation relevant to the evaluation metric. Thus, the proxy can serve as a good approximation to the training game utility in stable regimes. In turn, the contributor ranking induced by Shapley values is preserved whenever the differences between contributors’ Shapley values are sufficiently larger than the approximation error.
>
> If there is any other information we can provide during the discussion period that would allow you to further raise your score, please let us know.
>
> Thanks again,
>
> #### **Due to space constraints, we have dedicated the remainder of the space to responding to Reviewer QKA6’s questions/minor concerns.**
>
> ---
> ### **Re: Scalability and surrogate fit**
> We thank the reviewer for these important questions. We agree that reporting these results will improve clarity and will include them in the revision. Below, we report the number of sampled coalitions ($M$), computation time, and the surrogate’s 5-fold CV performance on unseen coalitions drawn from the Shapley distribution, as measured by $R^2$ and Spearman’s $\rho$.
> |Dataset|# Players|$M$|Compute|$R^2$ |$\rho$|
> |---|---:|---:|---|---|---|
> |CIFAR-20|20|600|1 day (4×A40)|FID: 0.920; IS: 0.967|FID: 0.902; IS: 0.965|
> |ArtBench Post-impressionism|256|700 |1 day (4×A40)|Aesthetics: 0.142|Aesthetics: 0.46|
> |Fashion-Product|100|800|2 days (4×H200)|Diversity: 0.103; LPIPS: 0.132|Diversity: 0.434; LPIPS: 0.516|
>
> These results show that the surrogate models the coalition space accurately for CIFAR-20, where the coalition space is small ( $2^{20}$). While $R^2$ is lower for ArtBench and Fashion-Product, this is expected given the sparse sampling budget ($M \le 800$) relative to the massive coalition spaces ($2^{100}$ and $2^{256}$). While a low $R^2$ suggests that the estimated Shapley values may deviate from the ground truth, the surrogate still preserves relative rankings, as reflected by $\rho$. This ranking fidelity provides a useful signal for Shapley estimation and is further supported by our LDS and counterfactual evaluations, where the resulting Shapley values outperform. We will revise the paper to state our claim more precisely: **our method enables practical Shapley attribution for large-scale T2I models, but does not guarantee uniformly accurate approximation over unseen coalition spaces**.
>
> ### **Re: Counterfactual evaluation - Diversity.**
> We thank the reviewer for raising this point. We observe that the Shapley values for diversity are highly skewed: only a small fraction of contributors (~10%) have high positive values, while many others are near-zero or negative. As a result, once the top positive contributors are removed, removing additional contributors need not produce further degradation and can even offset it. Furthermore, diversity is computed from pairwise similarities across generated samples, making it more variable than pointwise metrics,e.g., LPIPS. The moderate alignment between original game instances ($R^2 = 0.48, \rho = 0.53$) further indicates that this is a challenging setting.
>
> ### **Re: Dermatology audit experiment.**
> We thank the reviewer for raising these points. Our training data is balanced for sex and melanoma (Appendix G); therefore, any observed correlation between classifiers' outputs and sex is inherently driven by spurious features. Following your suggestion, we also replaced the ViT with EfficientNet-B2 and confirmed that Hospital 7 remains the top contributor. **This demonstrates that our method is robust to the choice of classifier architecture.** Regarding the hair artifact, we agree that it is a post hoc interpretation rather than a causal claim, and we will clarify this in the revision. Verifying this relationship is an interesting direction, but it is beyond our current scope.
>
> ### **Re: Notation and clarity.**
> We thank the reviewer for highlighting these presentation issues. We will revise $\mathcal{F}(p_\theta)$ to make its dependence on the subset-conditioned mixture $S$ explicit, and include the limitations of the contributor–label correspondence setting.

---

> > ### Author Rebuttal · Reviewer_h3h7 · 2026-04-02
> >
> > Thanks for the reply, and my concerns have been fully addressed. I will maintain my positive score.

---

> > > ### Author Response · Authors · 2026-04-02
> > >
> > > We appreciate the reviewer’s positive feedback and are glad that our responses resolved your question & concern!

---

### Official Review · Reviewer_wV9C · 2026-03-13

**Soundness:** 2
**Presentation:** 3
**Significance:** 2
**Originality:** 2
**Overall Recommendation:** 4
**Confidence:** 5

**Summary:**

This paper proposes SurrogateSHAP, a training-free contributor attribution framework for text-to-image diffusion models.
The main idea is to replace the expensive retraining game with a proxy game that evaluates coalitions by sampling from the full model restricted to coalition-specific prompts, and then fit a gradient-boosted tree surrogate over coalition utilities so that TreeSHAP can be used for more sample-efficient contributor valuation.
The method is evaluated on CIFAR-20, ArtBench, and a Fashion-Product dataset, with utilities including FID, Inception Score, aesthetic score, LPIPS, and diversity.
The paper reports that the proxy tracks retraining reasonably well, that TreeSHAP improves query efficiency over KernelSHAP on synthetic games, and that SurrogateSHAP outperforms baselines such as LOO, TRAK variants, D-TRAK, DAS, and sparsified-FT Shapley in LDS and counterfactual removal experiments.

**Compliance With Llm Reviewing Policy:**

Affirmed.

**Final Justification:**

My concerns have been adequately addressed.

**Key Questions For Authors:**

1. The paper reports very poor and even negative LDS for TRAK and related gradient-based baselines on some tasks. Can the authors provide stronger validation that these baselines were implemented and tuned correctly for each setting, for example with hyperparameter sweeps, reproductions of known results, or sanity checks on utilities more closely aligned with the diffusion training objective? A convincing answer here would substantially increase my confidence in the empirical claims.

2. For ArtBench and Fashion-Product, “retraining” is replaced by LoRA fine-tuning rather than full retraining. How sensitive are the conclusions to that choice, and in what sense should the proposed proxy still be interpreted as approximating the original retraining game rather than a LoRA-specific game? This matters for the paper’s core claim.

3. The budget comparison omits GBT training time on the grounds that it is negligible. Can the authors report the actual wall-clock numbers and include them in the main comparison for completeness?

**Limitations:**

yes

**Strengths And Weaknesses:**

1. The paper addresses an important problem and the overall direction is interesting. Replacing repeated retraining with a proxy game, then fitting a surrogate for Shapley estimation, is a sensible systems-oriented idea. The empirical results are also broad in the sense that they cover multiple datasets and utilities, and the paper reports meaningful speedups over retraining-based alternatives.

2. My main concern is that the evidence for the central empirical claim is not yet strong enough. The theoretical discussion relies on a stability assumption that is not directly verifiable in realistic settings, and the key large-scale experiments do not compare against true retraining for Stable Diffusion/FLUX, but instead use LoRA fine-tuning as the practical stand-in for retraining. That makes the target being approximated less clear, especially when the paper’s main claim is about fidelity to the retraining game.

3. I am also not fully convinced by the treatment of baselines, especially the very poor D-TRAK results. The paper reports strongly negative LDS for D-TRAK on ArtBench and weak or inconsistent results elsewhere, but the setup is also intentionally focused on downstream utilities such as aesthetics and diversity rather than training-loss-aligned objectives. Since the paper itself argues that gradient-based methods can be misaligned with such utilities, these negative numbers do not by themselves show that D-TRAK is flawed or incorrectly implemented; they may simply reflect a mismatch between method and target metric. The baseline section says TRAK-based methods compute gradients at the target model and then aggregate datum-level attributions by summation, but I did not find enough ablation or sanity checks to rule out implementation sensitivity, hyperparameter sensitivity, or unfair disadvantage relative to the proposed method. The omission of GBT training time from the budget comparison is also favorable to the proposed approach, even if the authors claim it is negligible.

---

> ### Author Rebuttal · Authors · 2026-03-29
>
> Dear reviewer wV9C,
>
> We thank the reviewer for your thoughtful feedback. We are pleased that the reviewer recognizes the significance of our research problem and our method in achieving meaningful speedups. Below, we respond to the reviewer’s concerns.
>
> ### **Re: Empirical validation of the stability assumption.**
> While $\epsilon^{\phi_u}$ in Eq. (6) is intractable, we approximate it empirically by measuring the feature-space gap (LPIPS, CLIP) between samples generated by the proxy and original games for the same coalition, across 100 sampled Shapley coalitions. For reference, we also report the results between two independently trained instances of the original game with different seeds, which provides an empirical minimum. As shown below and in Table 1, representation drift, i.e., RMS CLIP/LPIPS, remains reasonably controlled and tracks the downstream utility difference, i.e., NMAE, NRMSE, between the original and proxy games.
> ||Method|RMS CLIP|RMS LPIPS|NMAE|NRMSE|
> |---|---|---:|---:|---:|---:|
> |ArtBench Post-impressionism|LoRA-ft vs LoRA-ft|0.446|0.339|0.029|0.099|
> || sFT|0.571|0.479|0.101|0.166|
> || Ours|0.529|0.458|0.068|0.124|
> |Fashion-Product |LoRA-ft vs LoRA-ft |0.399|0.267|0.087|0.116|
> || sFT|0.458|0.337|0.198 |0.296|
> || Ours|0.429|0.301|0.165|0.221|
> ### **Re: Proxy target & definition of retraining game.**
> We thank the reviewers for highlighting this point. We agree that the term *retraining game* is imprecise in the current draft and may suggest fidelity specifically to full end-to-end retraining from scratch. This was not our intent. **In our definition, the retraining game (page 3, line 133) is defined by a specified training algorithm, which may range from full retraining to different fine-tuning procedures. Our goal is therefore to attribute each contributor’s effect on the final model under that training procedure, rather than under full retraining alone.** To avoid conflating these settings and to better reflect the generality of the definition, we will revise the terminology throughout the paper from *retraining game* to *training game*.
>
> We also agree that results under LoRA do not automatically generalize to full fine-tuning. We therefore provide additional full fine-tuning results (without LoRA) for Stable Diffusion on ArtBench Post-Impressionism. The LDS results below show that SurrogateSHAP continues to provide a reliable ranking of held-out utility in this setting.
> |Metric|Ours|TRAK|D-TRAK|Journey-TRAK|Relative IF|Renorm IF|DAS|Gradsim|Pixel Sim|CLIP Sim|Aesthetic score|
> |---|---:|---:|---:|---:|---:|---:|---:|---:|---:|---:|---:|
> |LDS (%)|39.0|-5.34|-8.75|6.64|-3.01|-5.84|-5.66|-5.43|3.62|5.32|6.91|
>
> Similarly, we evaluate how well the proxy game approximates the full fine-tuning game, both in terms of representation drift (RMS CLIP/LPIPS) and downstream model behavior (NRMSE/NMAE). For reference, we also report the results from two full fine-tuning instances as empirical estimates of the minimum (shown in parentheses), and the results show that our proxy provides reasonable approximations. **Together with CIFAR-20 (retraining) and Fashion-Product (LoRA ft), these results show that SurrogateSHAP is effective across various training games.**
> ||RMS CLIP|RMS LPIPS|NRMSE| NMAE|
> |---|---:|---:|---:|---:|
> |Proxy vs full ft|0.636 (0.427)|0.554 (0.337)|0.136 (0.082)|0.098 (0.057)|
>
> ### **Re: Mismatch of gradient-based methods.**
> We thank the reviewer for raising this point. We agree that TRAK-based methods are not flawed; rather, as the reviewer noted, they are not equally suited to all attribution settings. For example, artist crediting may require evaluating influence across a broader body of work using loss-independent utilities, such as stylistic or aesthetic measures. Since gradient-based methods typically attribute influence through individual datapoints and specified training objectives, e.g., diffusion loss, they are less suited to capturing group-level influence under such utilities [1,2]. As such settings become increasingly important, these limitations directly motivate our utility-agnostic, group-level framework.
>
> ### **Re: Sanity check with diffusion loss as utility.**
> Following the reviewer’s suggestion, we validated our implementation of TRAK-based methods on ArtBench Post-Impressionism in an individual data attribution setting, using diffusion loss as the target utility (similar to ArtBench-2 in [3]). Our result shows that both methods achieve positive LDS, and D-TRAK outperforms TRAK ($21.60 \pm 4.88$ vs. $11.8 \pm 4.93$). These results align with those in [3] and show that our implementation recovers the expected behavior.
>
> ### **Due to space constraints, our response to the reviewer's remaining questions/concerns can be found in the response box for tE6D. Please let us know if we can further clarify any of these points.**
>
> **References**
> 1. https://arxiv.org/abs/1905.13289
> 2. https://arxiv.org/abs/2407.03153
> 2. https://arxiv.org/pdf/2311.00500

---

> > ### Author Rebuttal · Reviewer_wV9C · 2026-04-03
> >
> > Thanks, I raised my score.

---

> > > ### Author Response · Authors · 2026-04-03
> > >
> > > We appreciate your thoughtful feedback and are glad that our responses have addressed your concerns. We sincerely thank you for your updated assessment of our work.

---

### Official Review · Reviewer_QKA6 · 2026-03-13

**Soundness:** 3
**Presentation:** 3
**Significance:** 3
**Originality:** 2
**Overall Recommendation:** 4
**Confidence:** 4

**Summary:**

This paper proposes SurrogateSHAP, a framework for attributing the influence of data contributors (groups of training samples) to text-to-image diffusion models using Shapley values. The method addresses two bottlenecks in Shapley-based attribution: (i) the cost of retraining models for each coalition subset, and (ii) the number of coalition queries needed to estimate Shapley values accurately. The first bottleneck is tackled by defining a training-free proxy game that treats the pretrained model's conditional distributions as stand-ins for retrained subset models, reducing coalition evaluation to mixture sampling. The second is addressed by fitting a gradient-boosted tree (GBT) surrogate to the proxy utility and extracting exact Shapley values via TreeSHAP. The authors evaluate on three settings (CIFAR-20 with DDPM-CFG, ArtBench Post-Impressionism with Stable Diffusion, Fashion-Product with FLUX.1) using LDS, counterfactual removal, and proxy fidelity metrics, and include a clinical case study on dermatology bias auditing using ISIC data.

**Compliance With Llm Reviewing Policy:**

Affirmed.

**Key Questions For Authors:**

1. What is the actual number of coalition samples M used for each of the three real-data experiments? How does the surrogate's held-out MSE (or R²) vary with M on the Fashion-Product benchmark (n=100), and at what M does performance saturate?
2. The proxy fidelity for ArtBench aesthetics (R² = 0.330) and Fashion-Product LPIPS (R² = 0.150) is low. Can you provide an analysis of which coalitions exhibit the largest proxy-retraining gap, and whether these correspond to coalitions where a dominant contributor is removed (violating the stability assumption)?
3. In the Fashion-Product diversity counterfactual (Table 3), the effect at top-20% removal (−0.60%) is dramatically smaller than at top-5% (−16.11%). What explains this non-monotonicity? Is the diversity metric itself unreliable at larger removal fractions, or does this indicate a failure mode of the attribution?
4. For the dermatology case study, how sensitive is Hospital 7's attribution to the choice of sex and melanoma classifiers? If you swap in a different classifier architecture, does Hospital 7 remain the top contributor?

**Limitations:**

The authors provide a reasonable impact statement acknowledging risks of metric-specific pruning leading to representation bias. However, several technical limitations are underaddressed: (a) the contributor-label one-to-one mapping assumption excludes settings where contributors share overlapping prompt spaces (e.g., two brands producing similar product categories); (b) the proxy game's fidelity degrades substantially for certain utility functions (aesthetics, LPIPS), yet no guidance is given on when the proxy should not be trusted; (c) computational cost of the proxy itself, generating mixture samples and evaluating distributional metrics like FID, still requires thousands of forward passes, which is not negligible for FLUX-scale models; (d) the dermatology study uses the ISIC dataset which includes patient sex metadata without discussing IRB status or data use agreements, though this data is publicly available with established research use protocols.

**Strengths And Weaknesses:**

Strengths:
1. The two-stage decomposition—proxy game followed by tree surrogate—is well-motivated and cleanly separates the approximation into two independently analyzable error terms (Equation 11). The error decomposition into E_sur and E_gap is a useful conceptual contribution that clarifies where fidelity is lost.
2. The experimental coverage is broad: three distinct model architectures (DDPM, Stable Diffusion, FLUX), five utility functions (FID, IS, aesthetics, LPIPS, diversity), and a compelling clinical case study. The dermatology auditing experiment (Section 6) is the strongest applied contribution, demonstrating that SurrogateSHAP can localize spurious correlations that naive label-level analysis (Figure S.26) cannot detect.
3. The synthetic validation in Section E is thorough. The axiomatic checks (null player, efficiency) and the bias-variance decomposition (Figure S.2) add credibility to the TreeSHAP-based estimator, particularly in the interaction-heavy regime where KernelSHAP clearly plateaus.

Weaknesses:

1. The proxy game assumption is strong and inadequately validated for the harder settings. Assumption 4.1 posits uniform bounded representation drift across all coalitions and all conditions. The empirical evidence for ArtBench shows only moderate proxy-retraining alignment (R² = 0.330 for aesthetic score, Spearman ρ = 0.589). The Fashion-Product LPIPS proxy achieves R² = 0.150, ρ = 0.443. These are weak correlations that undermine the theoretical narrative. The paper treats these as acceptable without discussing the regime where the assumption breaks down—e.g., when a contributor is removed whose data is semantically distant from other contributors and the pretrained conditional cannot compensate. The bound in Proposition 4.2 is vacuous if ε^φ_u is large, and the authors never estimate ε^φ_u empirically to show it is meaningfully small.
2. "Retraining" in two of three benchmarks means LoRA fine-tuning, not full retraining. The paper is upfront about this (Section 5.1), but it conflates the two throughout. The proxy game approximation is likely to be tightest when "retraining" is parameter-efficient fine-tuning (since the model backbone is frozen and conditionals are more stable). This means the favorable proxy fidelity numbers in Table 1 for ArtBench and Fashion-Product may not generalize to settings where contributors actually retrain the base model. The CIFAR-20 experiment does use full retraining, but at 32×32 resolution with a 10.4M parameter U-Net—a toy setting that is not representative of modern T2I workloads.
3. The LDS metric is itself an assumption-laden evaluation. LDS measures how well additive aggregation of per-contributor Shapley values predicts coalition utilities. But the entire motivation for Shapley values over LOO is that interactions matter and additive aggregation of individual scores fails. High LDS thus rewards methods whose scores happen to be linearly predictive of held-out utilities—a somewhat circular property for a method that explicitly models non-linear interactions. The authors do not discuss this tension.
4. Missing ablation on the GBT surrogate. The paper uses XGBoost with a grid search over a modest hyperparameter space (Table S.1). There is no ablation on surrogate model class (e.g., random forest, MLP, linear model) to demonstrate that the GBT architecture specifically is needed. The claim that TreeSHAP provides exact Shapley values on the surrogate is true by construction, but the surrogate generalization error on unseen coalitions (especially in high-n regimes like ArtBench with 258 artists) is never reported as a standalone metric. The 5-fold CV MSE of the surrogate should be reported.
5. Scalability claims are not stress-tested. The largest contributor count is 258 (ArtBench), and the sampling budget is not stated clearly for the real experiments (only implied through the "identical number of coalitions" footnote in Table 2). For 100 brands in Fashion-Product, the Boolean hypercube has 2^100 coalitions. How many coalitions M were actually sampled? What is the surrogate's test-set R² at this scale? The paper omits these numbers despite them being the most informative measure of practical scalability.
6. Counterfactual evaluation on Fashion-Product (Diversity) is noisy and inconsistent. In Table 3, SurrogateSHAP achieves −16.11% at top-5% removal but only −0.60% at top-20% removal. This non-monotonicity is suspicious and unexplained. For a method that claims to rank contributors correctly, removing more high-ranked contributors should generally produce larger degradation. The baselines also show erratic behavior (D-TRAK: +2.21% at 5%, meaning diversity increased when supposedly influential contributors were removed). The high variance across seeds (±8.34 at 10%) suggests the diversity metric itself is unstable, which casts doubt on whether it is a valid utility for this evaluation.
7. The dermatology case study lacks important controls. The correlation between sex-classifier and melanoma-classifier outputs (r = 0.21) is used as the utility function, but the authors do not verify that this correlation is actually spurious vs. reflecting a genuine epidemiological signal (melanoma incidence does differ by sex). The claim that Hospital 7's hair artifacts drive the correlation is plausible but post-hoc; no ablation confirms that the hair-artifact mechanism is causal rather than coincidental. A controlled experiment injecting known confounders would strengthen this substantially.
8. Notation and writing quality. The paper is generally well-written but has inconsistencies: v̂_θ(S) is defined in Equation 5 with the mixture over p_θ, but in Proposition 4.2 it is redefined as F(p_θ) without the mixture notation (Equation 8 partially reconciles this, but the transition is confusing). The contributor-label correspondence (Definition 3.1 and surrounding text) assumes a one-to-one mapping between contributors and prompt tags, which is restrictive and not discussed as a limitation.

---

> ### Author Rebuttal · Authors · 2026-03-29
>
> Dear reviewer QKA6,
>
> We thank the reviewer for carefully examining our work and providing your feedback. We’re happy to hear that you acknowledge our motivation and method clarity, and that our dermatology use case is “the strongest applied contribution”. In the space below, we respond to the reviewer’s specific concerns.
>
> ### **Re: Regarding the moderate correlations for ArtBench and Fashion-product**
> We thank the reviewer for raising this point. The moderate correlations on ArtBench and Fashion-Product reflect the inherent difficulty of these settings: complex conditioning and substantially larger player counts (256 and 100) than CIFAR-20. To establish a realistic maximum, we compared two independently trained instances from the original game (LoRA ft). This yields $R^2 = 0.61, \rho = 0.71$ for ArtBench, and $R^2 = 0.56, \rho = 0.59$ for Fashion-Product. Relative to this baseline, the proxy-target correlations remain informative, and as shown in our LDS and counterfactual evaluations, our proxy captures useful information for identifying important contributors.
>
> ### **Re: Empirical validation of the stability assumption**
> We thank the reviewer for these important questions. Due to space constraints, we refer the reviewer to our responses to **Reviewer wV9C — Empirical validation of the stability assumption**.
>
> ### **Re: Proxy target & definition of retraining game**
> We agree that *retraining game* is imprecise in the current draft and may suggest full end-to-end retraining from scratch. Due to space constraints, we refer the reviewer to our response to **Reviewer wV9C — Proxy target and definition of the retraining game**.
>
> ### **Re: Additive aggregation evaluation (LDS)**
> Our motivation for choosing Shapley over LOO is indeed that player interactions matter. However, this does not mean that additive aggregation evaluation, e.g., LDS, is inappropriate. Rather, it means that a good attribution method should account for the players' interactions within each coalition when assigning individual scores. Shapley values satisfy this property by incorporating each player’s average marginal contribution across all possible coalitions. As a result, the sum of individual Shapley values can still serve as a meaningful additive surrogate for coalition utility, yielding high LDS as shown in our evaluation. Such additive aggregation evaluation is commonly used in feature attribution [1].
>
> We acknowledge that, as LDS is measured under the datamodel distribution, it may favor methods estimated from more diverse coalitions than LOO. However, this alone does not explain our gains compared to Shapley-based baselines under the same coalition budget (Table 2).
>
> ### **Re: Surrogate comparison and XGBoost performance with varying $M$**
> Following the reviewer’s feedback, we performed a 5-fold cross-validation to evaluate the surrogate’s MSE using the same number of sampled coalitions ($M$) as in Table 2: $M=600$ for CIFAR-20, $M=700$ for ArtBench, and $M=800$ for Fashion-Product. We compared XGBoost against several alternative surrogate architectures, including a linear model, a 2-layer MLP, and a random forest. The results demonstrate that XGBoost achieves the lowest MSE across 4 of the 5 evaluated utility metrics; while the MLP performed best on CIFAR-20 FID, XGBoost remained the most robust architecture overall. These findings are consistent with prior work [2] and validate our choice of the surrogate model.
>
> | Surrogate | CIFAR-20 (FID) | CIFAR-20 (IS) | ArtBench (Aesthetic score) | Fashion product (LPIPS, $\times 10^{-3}$) | Fashion product (Diversity, $\times 10^{-3}$) |
> |---|---:|---:|---:|---:|---:|
> | Linear | 595.4 (70.1) | 0.517 (0.068) | 0.236 (0.052) | 1.4 (0.1) | 1.0 (0.1) |
> | MLP | **95.0 (43.0)** | 0.117 (0.015) | 0.356 (0.041) | 8.7 (1.5) | 8.2 (1.2) |
> | RandomForest | 345.9 (63.1) | 0.359 (0.039) | 0.115 (0.017) | 1.2 (0.2) | 0.7 (0.1) |
> | XGBoost | 139.9 (30.1) | **0.114 (0.050)** | **0.085 (0.023)** | **1.1 (0.2)** | **0.6 (0.1)** |
>
> Below, we report 5-fold CV MSE for Fashion-Product as $M$ varies. Performance generally improves with larger, though not monotonically, which is expected given the enormous and sparsely sampled coalition space. We do not observe a clear saturation point within our compute budget.
>
> | $M$ | Diversity ($\times 10^{-4}$) | LPIPS ($\times 10^{-4}$) |
> |---|---:|---:|
> | 100 | 13.7 (3.22) | 9.15 (3.02) |
> | 200 | 15.0 (2.47) | 7.24 (1.36) |
> | 400 | 13.0 (3.91) | 6.59 (1.92) |
> | 600 | 11.4 (3.93) | 7.05 (2.67) |
> | 800 | 11.8 (2.53) | 5.85 (0.73) |
>
> ### **Due to space constraints, our response to the reviewer's remaining questions/concerns can be found in the response box for h3h7. Please let us know if we can further clarify any of these points.**
>
> #### **References**
> 1. https://arxiv.org/abs/2005.00631
> 2. https://arxiv.org/pdf/2505.17495

---

> > ### Author Rebuttal · Reviewer_QKA6 · 2026-04-04
> >
> > The core issues: moderate proxy fidelity on the harder benchmarks, noisy counterfactual results for Fashion-Product diversity, and limited robustness checks for the clinical case study remain only partially resolved.

---

> > > ### Author Response · Authors · 2026-04-05
> > >
> > > We thank the reviewer for their continued engagement and for highlighting these remaining points. To address them directly, we provide additional experimental results below.
> > >
> > > ### **Re: Moderate proxy fidelity and noisy counterfactual results on the harder benchmarks.**
> > >
> > > We thank the reviewer for raising these points. The moderate fidelity observed was based on pairwise metrics, primarily Multi-Scale Structural Similarity (MS-SSIM)-based diversity. To assess whether this behavior is metric-specific in the more challenging Fashion-Product setting, we additionally evaluate two alternative metrics used for fashion items: Inception Score (IS) and entropy-based diversity [1,2]. Below, we report the alignment of our proxy and the training game under these two metrics, and compare against the existing sparsified fine-tuning (sFT) proxy [2]. Results are reported as Spearman's rank correlation/R-squared ($\rho$/$R^2$).
> > >
> > > | Metrics | Training Game | sFT | Ours |
> > > | :--- | :---: | :---: | :---: |
> > > | MS-SSIM diversity | 0.560 / 0.590|0.435 / 0.100 | 0.443 / 0.150 |
> > > | Entropy-based diversity|0.979 / 0.997|0.842 / 0.801 | 0.970 / 0.930 |
> > > | IS | 0.952 / 0.976|0.832 / 0.810 |0.950 / 0.920 |
> > >
> > > As shown in the table, our method demonstrates strong proxy fidelity on the two additional metrics, closely tracking the Training Game oracle. While alignment remains moderate for MS-SSIM diversity, our method still outperforms the sFT baseline. Similarly, we evaluated LDS on these two new metrics. As shown in the table below, our method achieves LDS scores of 59.37% and 55.46%, respectively (also using 800 sampled coalitions, consistent with Table 1).
> > >
> > > | | Ours | TRAK | D-TRAK | Journey TRAK | Relative IF | Renorm IF | DAS | Grad Sim | Pixel Sim | CLIP Sim |
> > > | :--- | :---: | :---: | :---: | :---: | :---: | :---: | :---: | :---: | :---: | :---: |
> > > | Entropy-based diversity | **59.37 (5.15)** | -21.40 (5.98) | -17.30 (2.43) | 2.52 (5.32) | -22.62 (6.00) | -18.27 (6.13) | -21.12 (5.72) | 5.00 (5.56) | 9.07 (3.34) | -9.78 (2.03) |
> > > | IS | **55.46 (2.10)** | -36.15 (1.67) | -33.85 (3.43) | 4.78 (1.38) | -35.24 (1.33) | -32.26 (1.64) | -35.87 (1.76) | -15.82 (3.25) | -15.84 (5.74) | -32.40 (3.99) |
> > >
> > > Lastly, the counterfactual evaluations below demonstrate that removing top contributors identified by our method yields a consistent drop in both entropy-based diversity and IS. Conversely, the two strongest baselines above produce weaker or inconsistent trends.
> > >
> > > | Metric | Method | 5% | 10% | 15% | 20% |
> > > | :--- | :--- | ---: | ---: | ---: | ---: |
> > > | Entropy-based diversity | **Ours** | **-2.39 (0.97)** | **-5.45 (0.50)** | **-6.13 (0.53)** | **-9.04 (0.28)** |
> > > || Grad Sim | -1.21 (0.34) | -2.28 (0.89) | -0.82 (0.27) | -2.60 (0.71) |
> > > || Pixel Sim | -0.96 (0.33) | -1.56 (0.44) | -2.50 (0.66) | -1.01 (0.06) |
> > > | IS | **Ours** | **-5.37 (5.43)** | **-16.36 (1.58)** | **-16.25 (1.93)** |**-19.92 (1.18)** |
> > >  || Grad Sim | +1.38 (1.28) | -3.21 (3.80) | +2.56 (2.72) | -2.32 (1.59) |
> > >  || Journey TRAK | +9.13 (1.50) | +5.05 (1.74) | +2.25 (2.13) | -2.79 (2.00) |
> > >
> > > Together, these results suggest that the weaker fidelity previously observed on harder benchmarks is metric-dependent. Specifically, pairwise metrics like MS-SSIM diversity appear noisier in this setting than point-wise metrics (such as IS and entropy-based diversity). When evaluated on more stable metrics, our method tracks the training game oracle very closely. We will explicitly discuss this as a limitation in the revised paper.
> > >
> > > ### **Re: Limited robustness checks for the clinical case study.**
> > > To test the robustness of our method against the choice of classifier, we replaced the ViT architecture with EfficientNet-B2. While the overall correlation between the sex and melanoma classifier outputs shifted from $r=0.21$ ($p=3.2 \times 10^{-6}$) with ViT to $r=0.106$ ($p=7.5 \times 10^{-6}$) with EfficientNet-B2, the coalition outputs of the two classifiers remained positively correlated ($r=0.68$, $p=0.031$). Most importantly, both settings consistently identified Hospital 7 as the top contributor, with the resulting values of 0.28 (ViT) and 0.08 (EfficientNet-B2). These findings demonstrate that our method remains robust across variations in the underlying classifier architecture. **Please let us know if you would like further information or if we have misunderstood your question.**
> > >
> > > **Reference**
> > >
> > > [1] Moosaei et al. "OutfitGAN: Learning Compatible Items for Generative Fashion Outfits, CVFAD," CVPR 2022.
> > >
> > > [2] Lin et al. "An Efficient Framework for Crediting Data Contributors of Diffusion Models," ICLR 2025.

---

### Decision · Program_Chairs · 2026-04-30

**Decision:**

Accept (regular)

**Comment:**

After reading all the materials of this paper, my recommendation is *accept*. No objection was raised in the internal discussion.

**Research Question**

This paper considers the data attribution in the setting of training-free text-to-image models.

**Challenge**

The authors argue that the existing Shapley value-based data attribution methods suffer from an expensive computational cost.

**Philosophy**

The authors aim to approximate the expensive retraining gram through inference.

**Solution**

The authors proposed SuurogateShap, which has two complementary components. The first one is a training-free proxy game that reinterprets the target model as a mixture of conditional generators, enabling subset evaluation without retraining. The second further boosts the efficiency via a tree-ensemble surrogate.

**Experiments**

The experimental evaluation is extensive.

1. Figure 1 is good to verify the correctness of the proposed method.

2. Table 3 is necessary to validate the performance in the downstream task. In addition, it is preferred to see the performance improve via removing detrimental samples, rather than beneficial ones, i.e., putting Table S7-11 in the main paper.

Based on the comments from all reviewers and my meta-review, I would like to recommend this paper for acceptance to ICML.